

# 1 Picturing and modelling catchments by representative
# 2 hillslopes

Ralf Loritz[1], Sibylle K. Hassler[1], Conrad Jackisch[1], Niklas Allrogen[2], Loes van Schaik[3],
Jan Wienhöfer[1], and Erwin Zehe[1]
[1] Karlsruhe Institute of Technology (KIT), Institute of water and river basin management,
Karlsruhe, Germany
[2] University of Potsdam, Institute of Earth and Environmental Science, Potsdam,
Germany
[3] Technische Universität Braunschweig, Enviromental Systems Analysis, Institute of
Geoecology, Braunschweig, Germany
*Correspondence to*: Ralf Loritz ([Ralf.Loritz@kit.edu](mailto:Ralf.Loritz@kit.edu))
**Abstract:** Despite the numerous hydrological models existing in hydrology we are
limited to a few forms of conceptualization when abstracting hydrological systems into
different model frameworks. Speaking in black and white terms, in most cases
hydrological systems are either represented spatially lumped with conceptual models or
spatially explicit with physically-based models. Physically-based models are often
parameter-rich, making the parametrization challenging, while conceptual models are
parsimonious, with only a few parameters needing to be identified. But this simplistic
mathematical expression is often also their drawback since their model states and
parameters are difficult to translate to the physical properties of a catchment. It is
interesting to note that both hydrological modeling approaches often start with the
drawing of a perceptual model. This follows the hydrologist's philosophy to separate
dominant patterns and processes from idiosyncratic system details. Due to the importance
of hillslopes as key landscape elements perceptual models are often displayed as 2D
cross-sections. In this study we examine whether we can step beyond the qualitative
character of perceptual models by using them as blueprint for setting up representative
hillslope models. Thereby we test the hypothesis if a single hillslope can represent the
functioning of an entire lower mesoscale catchment in a spatially aggregated way. We do
this by setting up and testing two hillslope models in catchments located in two different
geological settings, Schist and Marl, using a two-dimensional physically-based model.
Both models are parametrized based on intensive field data and literature values without



automatic calibration. Remarkably we are able to not only match the water balance of
both catchments but further have some success in simulating runoff generation as well as
soil moisture and sap flow dynamics. Particularly, our findings demonstrate that both
models performed well during the winter season and clearly worse during the summer
period. Virtual experiments revealed that this was most likely either due to a poor
representation of the onset of vegetation in the Schist catchment or due to emergence of
soil cracks in the Marl area. Both findings underpin that a static parameterization of
hydrological models might be problematical in case of emergent behavior. Additional
virtual experiments indicate that the storage of water in the bedrock and not so much the
topographic gradient is a first order control on the hydrological functioning of the Schist
catchment. We conclude that the representative hillslope concept is a feasible approach in
data rich regions and that this form of abstraction provides an added value to the
established conceptualization frameworks in hydrology.





## 1 Introduction

### 1.1 Representative hillslope models as catchment pictures

According to Heidegger (1977), "the modern is characterized as the age of the world picture". Science conceives and grasps the world as a picture rather than simply picturing the world. In hydrology such a picture is for instance reflected by a perceptual model. Perceptual models are abstractions of a hydrological system displaying how dominant properties and processes jointly control its functioning. It is conspicuous that these models are frequently drawn in the form of a hillslope-like cross-section. This reflect that hillslopes are often regarded as the key landscape elements controlling transformation of precipitation and radiation inputs into terrestrial fluxes and stocks of water (e.g. Bronstert and Plate, 1997), energy (Zehe et al., 2010a, 2013) and sediments (Mueller et al., 2010). Although any perceptional model is a simplification and is pre-determined by the designer's background and experience (Beven, 2012) it provides a useful means to facilitate communication between researchers with different backgrounds by integrating their knowledge (Seibert and McDonnell, 2002). Further, perceptual models are the natural starting point for any process-oriented model exercise.

Due to their simplistic way of abstracting the landscape, the rationale of this study is to explore whether perceptual models of two distinctly different catchments, derived from comprehensive and diverse field observations, may serve as blueprints for setting up representative hillslope models. Recently, Wrede et al. (2015) followed the same idea by deriving conceptual model structures from perceptual models of three headwaters located in the Attert experimental basin, which is also our study area. However, here we use a physically-based model because the underlying equations depend on observable parameters and state variables (Loague and VanderKwaak, 2004; Zehe et al., 2006). The representative hillslope model is hence a straightforward conceptualization of the underlying perceptual model (as illustrated in Figure 3, section 2) by drawing from available field data and expert knowledge. We regard this as the most parsimonious representation of the catchment in a physically-based model framework which provides a spatially explicit but yet effective picture about how gradients, spatial patterns of soil properties and rapid flow paths jointly control the dominant processes. Testing of such an approach offers several avenues for scientific learning as further specified below.





The usefulness of physically-based models as learning tools has been corroborated in
several studies, especially those working on the hillslope and lower catchment mesoscale.
For instance Hopp and McDonnell (2009) explored the role of bedrock topography on the
runoff formation. Complementary to this, Bishop et al. (2015), Wienhöfer and Zehe
(2014) and Klaus and Zehe (2011) focused on the influence of lateral and vertical
preferential flow networks on subsurface water flow and solute transport, including the
issue of equifinality and its reduction. These and other studies (e.g. Ebel et al., 2008)
corroborate that physically-based models may be benchmarked against a variety of
observations beyond stream flow – such as soil moisture observations, groundwater tables
or tracer break-through curves. Essentially, such a multi-response evaluation reveals
whether a model allows consistent predictions of dynamics within the catchment and for
its integral response behavior (Ebel and Loague, 2006). Last but not least, virtual
experiments with physically-based models are straightforward to implement and interpret,
because their parameters and their spatial setup are well connected to those observables
we use to characterize hydrological systems. Such virtual experiments may reveal first
order controls on hydrological system behaviors, and thereby sustain scientific learning
(Weiler and McDonnell, 2004), for instance to decide which variables shall be observed
within field campaigns in order to reduce equifinality (Zehe et al., 2014) as further
specified below.
**1.2 Challenges in modeling catchments by representative hillslopes**
Several physically-based and distributed model studies employ typical hillslope catenas
as building blocks (Bronstert and Plate, 1997; Zehe and Blöschl, 2004; Jackisch et al.,
2014). However, the challenge of how to identify a behavioral representative hillslope as
most parsimonious representation of a small catchment in a physically-based model has
rarely been addressed. This reflects the fact that the identifiability of representative
hillslopes has been strongly questioned since the idea was born. For example, Beven
(2006) argues that neither the hillslope form is uniquely defined nor is it clear whether it
is the form that matters, the pattern of saturated areas (Dunne and Black, 1970) or the
subsurface architecture. The enormous spatial variability of soil hydraulic properties and
preferential flow paths in conjunction with process non-linearity are additional arguments
against the identifiability of representative hillslope models (Beven and Germann, 2013).
However, hillslopes act as miniature catchments (Bachmair and Weiler, 2011), which
made Zehe et al. (2014) postulate that structurally similar hillslopes act as functional units



for the runoff generation and might thereby be a key unit for understanding functional
behavior of catchments of organized complexity (Dooge, 1986). Complementarily,
Robinson et al. (1995) showed that the behavior of catchments up to the lower mesoscale
are still strongly dominated by the hillslope behavior. Kirkby (1976) showed for
catchments extending up to 50 km$^2$ that random river networks had the same explanative
power for catchment behavior as the real river network. He concluded that as long as river
networks are not dominant the anomalous areas of the catchment bear the key to
understand its functioning. Unfortunately, topographic characteristics are not always
conclusive for explaining differences in hydrological behavior, as recently shown by
Jackisch (2015) for headwaters located in distinctly different geological parts of the Attert
experimental basin.

We hence suggest that it is worth investigating to what extent the behavior of these lower
mesoscale catchments can be explained using a single representative hillslope model. In
line with the arguments stated above such a representative hillslope can be neither a one-
to-one copy of a real hillslope in a catchment nor a simple average of several hillslopes
and their structural properties. A much more promising avenue might be to conceptualize
the representative hillslope as a functional analogue of the perceptual model, which is in
turn a generalized and simplified picture of the catchment structure and functioning. The
challenge in this model identification process is to balance necessary complexity with
greatest possible simplicity by preserving the typical patterns and structural catchment
characteristics and to neglect the unnecessary or idiosyncratic ones (Zehe et al., 2014).
Naturally, it has to be expected that a representative hillslope is, as a spatially aggregated
two-dimensional model of a catchment, not uniquely identifiable because non-uniqueness
and equifinality is inherent to all of the governing equations (Kirchner, 2006; Klaus and
Zehe, 2010; Zehe et al. 2014). But the degrees of freedom in the identification of
behavioral physically model structures can be reduced by using complementary
observations such as tracers (Klaus and Zehe 2011, Wienhöfer and Zehe, 2014) or
constraining parameters based on observations (Bárdossy, 2006).

Other reasons why representative hillslope studies are rarely realized are the challenges
and data needs that go along with applying physically-based models in larger
heterogeneous environmental systems (e.g. Or et al., 2015)  as well as the vital debate
about their limitations. Physically-based models typically rely on the Darcy-Richards



concept for simulating soil water dynamics, the Penman–Monteith equation for
simulating soil-vegetation-atmosphere exchange processes and 1-D or 2-D hydraulic
approaches for simulating overland and stream flow. Each of these concepts is naturally
subject to the limitations arising from our imperfect understanding of the related (bio-)
physical processes and the limited transferability of process descriptions which were
derived under idealized laboratory conditions to settings in natural systems (Grayson et
al., 1992; Gupta et al., 2012). For example, the Darcy-Richards equation assumes
dominance of capillarity controlled, mainly diffusive soil water fluxes and local
thermodynamic equilibrium conditions. While these assumption are well justified when
the radiation balance controls soil water dynamics it is violated during rainfall events
which trigger preferential flow (Hassanizadeh 2002; Simunek et al. 2003; Or 2008). The
variety of concepts that have been proposed to incorporate not-well mixed preferential
flow into hydrological models ranges from early stochastic convection (Simmons 1982),
double-domain approaches (Haws et al., 2005, Köhne et al., 2006, Bishop et al., 2015),
spatially explicit representations of macropores as connected flow paths (Sander and
Gerke 2009, Klaus and Zehe, 2010) to pore network models (Vogel and Roth, 2001;
Bastardie et al., 2003; Katuwal et al., 2015). As each of these approaches has specific
advantages and drawbacks we still lack an approach that is commonly agreed upon for
studies at the hillslope scale (Beven and Germann, 2013).

But besides the widely discussed limitations of hydrological models to represent
preferential flow, the simulation of soil-vegetation-atmosphere transfer is likely to be an
even weaker point. The large number of stomata conductance models that have been
proposed (Damour et al., 2010) reflects the uncertainty in the community on how to
represent plant physiological controls on transpiration in hydrological and land surface
models. Also a proper representation of the vegetation phenology is a challenge for
hydrological modelling. Most physically based models account for dynamic changes in
leaf area index, root depth and plant cover, however, often in the form of fixed annual
cycles that have been derived at a specific catchment or are taken from the literature. Soil
water is, however, a limiting factor for growth of agricultural crops (Abrahamsen and
Hansen, 2000; Kucharik and Brye, 2003) and the dynamics of functional vegetation,
particularly in semi-arid areas (e.g. Rodriguez-Iturbe, 1999; Tietjen and Jeltsch, 2007;
Tietjen et al., 2010). This implies that the plant phenology should be more of a dynamic
state of the model rather than a parameter set with a fixed annual cycle (Jackisch et al.,





2014). In those cases where locally observed time series of phenological data and crop
growth are not available, we often rely on parameterizations from the literature. While a
transfer of such parameter setups might not impair the simulation of direct runoff reaction
on the event scale, it may create serious biases in simulated soil moisture dynamics (Zehe
et al., 2010a) and long term simulations of the water balance of a catchment.
Because of the mentioned shortcomings, simulations with physically-based models will
essentially bear the fingerprints of the bio-physical limitations of the underlying equations
and of the limited transferability of parameter sets among places (Beven, 2002).
However, these limitations should not be misinterpreted as an argument against applying
imperfect physically-based models but rather as a challenge and an option for learning
(Loague and VanderKwaak, 2004). Particularly so, since the identification of these
limitations itself is of key importance to separate the predictable from the un-predictable
as well as for improving theoretical underpinnings of hydrological models (Clark et al.,

193  2016).

**1.3 Objectives and approach**
Despite - or in fact because of - all the shortcomings and possibilities discussed above, we
explore if and how we can picture and model two lower mesoscale catchments in the
Attert experimental basin by representative hillslopes. The rationale of our study is not to
"sell" a particular model, but a) to shed light on to what extent this most parsimonious
representation of the dominant catchment structural properties in a physically-based
model allows behavioral simulations of catchment functioning beyond reproduction of
stream flow and b) to identify limits in our theories and related physically-based models
by analyzing the model deficiencies as proposed by Ebel and Loague (2006). We hence
avoid automatic parameter calibration to optimize curve fitting since there are more
parsimonious and better suited model structures for this purpose. We instead rely on
various available observations, process-based reasoning, and appropriate literature data
for conceiving perceptual models and parameterizing representative hillslope models as
their functional analogues. More specifically, we use geophysical images to constrain
subsurface strata and bedrock topography (as suggested by e.g. Graeff et al., 2009),
representative soil-water retention curves derived from a large data set of undisturbed soil
samples, soil pits and dye staining experiments, and predefined phenological data (Zehe
et al., 2001) for model parametrization.





The key challenge in the hillslope identification process is to achieve the right balance of
complexity and simplicity by finding patterns and dominant structures in the
overwhelming heterogeneity of the surface and subsurface. By aggregating a catchment in
a two-dimensional hillslope and by choosing CATFLOW (described in detail in section
2.3) as the model framework we essentially assume:
• That the spatial organization of the catchment creates anisotropy in the fluxes
sustaining the water balance and stream flow production which allows an
aggregated description in two dimensions, representing downslope and vertical
flows as well as the dominant gradients, hydraulic properties and flow paths in
spatially explicit but yet effective manner.
• Timing of streamflow is mainly controlled by the hillslope properties, which
implies that the time scale of lateral flow concentration in the hillslope body is
larger than the time scale of flood routing in the river network at this scale.
• An effective representation of the topology of preferential flow paths is more
important than a representation of non-equilibrium between rapid flow and matrix
flow at this scale;
• An effective soil water retention curve derived from a large data set of
undisturbed soil samples is sufficient for simulation of the fluxes and average
storage dynamics controlling the water balance and stream flow production. The
relevant heterogeneity arises from spatial variability in the saturated hydraulic
conductivity and porosity, which may be accounted for in stochastic form.

Finally we benchmark the hillslope models, after successful reproduction of the water
balance, against the hydrograph as well as distributed soil moisture and sap flow
observations. This exercise will reveal the validity of the listed assumptions and of our
notion of the "perfect" balance of complexity and simplicity, as well as whether the
parameterization of a representative hillslope is transferable to a different catchment in
the same landscape and time. It can be furthermore seen as a first test of the concept of
hillslope scale functional units which constrain similarity of runoff production (Zehe et
al., 2014). We hence perform several virtual experiments to a) identify first order controls
on the functioning of these "functional units" and guide future field campaigns for their
identification and b) find key gaps in available process representations in order to sharpen
questions for future research.



**2. Study site and data basis**

**2.1 The Attert experimental basin**

This study is based on comprehensive laboratory and field data collected in the Attert basin within the CAOS (Catchments As Organized Systems) research unit (Figure 1, Zehe et al., 2014). The Attert basin is located in the mid-western part of the Grand-Duchy of Luxembourg and has a total area of 288 km². Mean monthly temperatures range from 18°C in July to a minimum of 0°C in January while mean annual precipitation in the catchment varies around 850 mm (1971–2000) (Pfister et al., 2000). The catchment covers three geological formations, the Devonian schists of the Ardennes massif in the northwest, Triassic sandy marls in the center and a small area of Luxemburg sandstone (Jurassic) in the southern part of the catchment (Martínez-Carreras et al., 2012). Our study areas are headwaters named Colpach in the Schist area (19.4 km²) and Wollefsbach in the Marl area (4.5 km²). As both catchments are located in distinctly different geologies (Figure 1) and land use settings, they differ considerably with respect to runoff generation and the dominant controls (e.g. Bos et al. 1996, Martínez-Carreras et al. 2012, Fenicia et al. 2014, Wrede et al. 2015 and Jackisch 2015).

**2.1.1 Colpach catchment: perceptual model of structure and functioning**

The Colpach catchment has a total area of 19.4 km² and an elevation range between 265 to 512 m a.s.l. It is situated in the northern part of the Attert basin on the Devonian schists of the Ardennes massif. Around 65 % of the catchment, mainly the steep hillslopes, are forested (Figure 2). In contrast, the plateaus at the hill tops are predominantly used for agriculture and pasture. Several geophysical experiments and drillings showed that bedrock and surface topography are distinctly different. The bedrock is undulating and rough with ridges, depressions and cracks (compare perceptional model Figure 3 A and also ERT image in Figure 6 B). Depressions in the bedrock interface are filled with weathered, silty materials which may form local pools and reservoirs with a high water holding capacity. These features are particularly important since lateral water flow along the bedrock interface is the dominant runoff process (Wrede et al., 2015). Specifically the subsurface structure is deemed to cause typical threshold-like runoff behavior similar to the fill-and-spill mechanism proposed by Tromp-Van Meerveld & McDonnell (2006). Further indication that fill-and-spill is a dominant process is given by the fact that the parent rock is reported as impermeable, which makes deep percolation through shallower



un-weathered schist layers into a large groundwater body unlikely (Juilleret et al., 2011).
The lack of significant observations of base flow underpins this notion. Furthermore,
surface runoff has been rarely observed in the catchment, except along forest roads,
which suggests a high infiltrability of the prevailing soils (Bos et al., 1996). This is in line
with distributed permeameter measurements and soil sampling performed by Jackisch
(2015). Moreover, numerous irrigation and dye staining experiments highlight the
important role of vertical structures for rapid infiltration and subsequent subsurface runoff
formation (Jackisch (2015), Figure 2 B).

### 2.1.2   Wollefsbach catchment: perceptual model of structure and functioning

The Wollefsbach catchment is located in the Triassic sandy marls formation of the Attert
basin. It has a size of 4.5 km² and low topographic gradients, with 61 m of maximum
elevation difference resulting in gentle slopes covering altitudes between 245 to 306 m
a.s.l. The catchment is intensively used for agriculture and pasture and only around 7 %
are forested (Figure 2 C) and hillslope are often tile drained (compare perceptional model
sketch in Figure 3 B). The marly soils are highly heterogeneous, from sandy loams to
thick clay lenses. Generally they are very silty with high water holding capacities and
mostly low saturated hydraulic conductivity of the soil matrix. Similar to the Colpach
catchment, vertical preferential flow paths play a major role for the runoff generation;
their origin, however, is distinctly different between the seasons. Biopores are dominant
in spring and autumn due to the high abundance of earthworms. Because earthworms are
dormant during midsummer and winter, their burrows are partly disconnected by
ploughing, shrinking and swelling of the soils (Figure 2 D, see also Figure 4). At the same
time, soil cracks emerge during long dry spells in midsummer due to the considerable
amount of smectite clay minerals in these soils, which drastically increase soil
infiltrability in summer (Zehe et al. 2007, Figure 4). The seasonally varying interaction of
both types of preferential flow paths with a dense man-made subsurface drainage network
is probably the reason for the flashy runoff regime of this catchment, where discharge
drops rapidly to baseflow when precipitation events end. However, as the exact position
of the subsurface drainage network and the worm burrows as well as the threshold for soil
crack emergence are unknown, the specific influence of each structure on runoff
generation is difficult to estimate.





### 2.1.3 Water balance and seasonality


The water balance of the Colpach and Wollefsbach catchments for several hydrological
years, presented as normalized double mass curves, are shown in Figure 5. Normalized
double mass curves are well suited for describing the annual water balance of a catchment
because they relate the cumulated runoff to the cumulated precipitation both divided by
the sum of the annual precipitation. Annual runoff coefficients in the Colpach catchment
vary around $0.51 \pm 0.06$ within the four hydrological years (Figure 5 A). In the
Wollefsbach catchment the annual runoff coefficients are smaller than in the Colpach
catchment and vary across a wider range, from 0.26 to 0.46 (Figure 5 B). In both
catchments the winter period is characterized by step-like changes which reflect fast
water mobilization during rainfall events partly due to rapid subsurface flow. In contrast,
the summer regime in the vegetation period is characterized by a smooth and almost flat
line. Accumulated rainfall input is not transformed into additional runoff but
predominantly into evapotranspiration (Jackisch 2015). Additionally we used a
temperature index model from Menzel et al., (2003) to detect the onset of the vegetation
period and to separate the vegetation-controlled summer regime from the winter period
which was already successfully used by Seibert et al. (2016) to mark onset and duration
on the summer regime in 22 catchments of the Bavarian Danube basin.

### 2.2 Data basis


### 2.2.1 Surface topography and land use


Topographic analyses are based on a 5 m LIDAR digital elevation model which was
aggregated and smoothed to 10 m resolution. Land use data from the "Occupation
Biophysic du Sol" is based on CORINE land use classes analyzed by colour infrared areal
images which were generated in 1999 from the "Administration du cadaster et de la
Topographie" at a scale of 1:15000.

### 2.2.2 Subsurface structure and bedrock topography


We used hillslope scale 2D electrical resistivity tomography (ERT) in combination with
augers and soil pits to estimate bedrock topography in the Schist area. Our auger profiles
revealed, in line with Juilleret et al., (2011) and Wrede et al., (2015), that the vertical soil
setup comprises a weathered silty soil layer with a downwards increasing fraction of rock
fragments, which is underlain by a transition zone of weathered bedrock fragments





followed by non-weathered and impermeable bedrock. Spatial subsurface information of
representative hillslopes were obtained from 2D ERT sections collected using a GeoTom
(GeoLog) device at four profiles on two hillslopes in the Colpach catchment. We used a
Wenner configuration with electrode spacing of 0.5 m and 25 depth levels: electrode
positions were recorded at a sub-centimeter accuracy using a total station providing 3D
position information. Application of a robust inversion scheme as implemented in
Res2Dinv (Loke, 2003) resulted in two-layered subsurface resistivity model Figure 6 B.
The upper 1 - 3 m are characterized by high resistivity values larger than 1500 $\Omega$*m. This
is underlain by a layer of generally lower resistivity values smaller than 1500 $\Omega$*m. In
line with the study of (Wrede et al., 2015) and in correspondence with the maximum
depth of the local auger profiles, we interpreted the transition from high to low resistivity
values to reflect the transition zone between bedrock and unconsolidated soil. In
consequence, we regard the 1500 $\Omega$m isoline as being representative for the soil-bedrock
interface.

### 2.2.3 Soil hydraulic properties and infiltrability

We determined soil texture, saturated hydraulic conductivity and the soil water retention
curve for 51 soil samples in the Schist area and 28 in the Marl area. Saturated hydraulic
conductivity was measured with undisturbed 250 ml ring samples with the KSAT
apparatus (UMS GmbH). The apparatus records the falling head of the water supply
though a highly sensitive pressure transducer which is used to calculate the flux. The soil
water retention curve of the drying branch was measured with the same samples in the
HYPROP apparatus (UMS GmbH) and subsequently in the WP4C dew point hygrometer
(Decagon Devices Inc.). The HYPROP records total mass and matric head in two depths
in the sample over some days when it was exposed to free evaporation (Peters and
Durner, 2008, Jackisch 2015 for further details). For both geological settings we
estimated a mean soil retention curve by grouping the observation points of all soil
samples (51 and 28, respectively), and averaging them in steps of 0.05 pF. We then fitted
a van Genuchten-Mualem model using a maximum likelihood method to these averaged
values (Table 1 and Figure 7).

### 2.2.4 Meteorological forcing and discharge

Meteorological data are based on observations from two official meteorological stations
(Useldange and Roodt) from the Administration des services techniques de l'agriculture
Luxembourg (ASTA). Air temperature, relative humidity, wind speed and global
radiation are provided on a temporal resolution of 1 h while precipitation data have a
temporal interval of 5 min. Precipitation was intensively quality checked by six
distrometers which are stationed within the Attert basin and by several randomly selected
rainfall events against rain radar observations, both by visual inspection. Discharge
observations are provided from the Luxembourg Institute of Science and Technology
(LIST).
**2.2.5 Sap flow, soil moisture data and meteorological forcing**
The Attert basin is instrumented with 45 automated sensor clusters. A single sensor
cluster measures rainfall, and soil moisture in three profiles with sensors in various
depths. In this study we use 38 soil moisture sensors located in the Schist area and 28
sensors located in the Marl area in 10 and 50 cm depth. The measurements (sensors
Decagon 5TE) have a temporal resolution of 5 min and an accuracy of ±3 % volumetric
water content (VWC) with a resolution of 0.08 % VWC. Furthermore we use sap flow
measurements based on the heat ratio method (Burgess et al., 2001) at 61 trees at a
temporal resolution of 12 h- means. Selected trees are either European Beech or Oak
trees.
**2.3 The physically based model CATFLOW**
Model simulations were performed using the physically-based hydrological model
CATFLOW (Maurer, 1997; Zehe et al., 2001). The model has been successfully used and
specified in numerous studies (e.g. Zehe et al., 2005; Zehe et al. 2010; Wienhöfer and
Zehe, 2014; Zehe et al., 2014). The basic modelling unit is a two-dimensional hillslope.
The hillslope profile is discretized by curvilinear orthogonal coordinates in vertical and
downslope directions; the third dimension is represented via a variable width of the slope
perpendicular to the slope line at each node. Soil water dynamics are simulated based on
the Richards equation in the pressure based form and numerically solved using an implicit
mass conservative "Picard iteration" (Celia et al., 1990). The model can simulate
unsaturated and saturated subsurface flow and has hence no separate groundwater routine.
Soil hydraulic functions after van Genuchten-Mualem are mostly used, though several
other parameterizations are possible. Overland flow is simulated using the diffusion wave
approximation of the Saint-Venant equation and explicit upstreaming. The hillslope
module can simulate infiltration excess runoff, saturation excess runoff, re-infiltration of



surface runoff, lateral water flow in the subsurface as well as return flow. For catchment
modelling several hillslopes can be interconnected by a river network for collecting and
routing their runoff contributions, i.e. surface runoff or subsurface flow leaving the
hillslope, to the catchment outlet.

**2.3.1 Evaporation controls, root water uptake and vegetation phenology**

Soil evaporation, plant transpiration and evaporation from the interception store is
simulated based on the Penman–Monteith equation. Soil moisture dependence of the soil
albedo is also accounted for as specified in Zehe et al., (2001). Annual cycles of plant
phenological parameters, plant albedo and plant roughness are accounted for in the form
of tabulated data derived within the Weiherbach project (Zehe et al., 2001). Optionally,
the impact of local topography on wind speed and on radiation may be considered, if
respective data are available. The atmospheric resistance is equal to wind speed in the
boundary layer over the squared friction velocity u* [L T$^{-1}$]. The former depends on
observed wind speed, plant roughness and thus plant height. The friction velocity depends
on observed wind speed as well as atmospheric stability, which is represented through six
stability classes depending on prevailing global radiation, air temperature and humidity.
The canopy resistance is the product of leaf area index and leaf resistance, which in turn
depends on stomata and cuticular resistance. The stomata resistance varies around a
minimum value, which depends on the Julian day as well as on a function of air
temperature, water availability in the root zone, the water vapour saturation deficit and
photosynthetic active radiation (Jarvis, (1976)). The soil conductance depends mainly on
the thickness of a topsoil layer controlling water vapour transport which is divided by the
water vapour diffusivity in soil. Root extraction is accounted for as a sink term that
operates as a flux per volume during potential evapotranspiration; the transpired water
is extracted uniformly along the entire root depth. When soil water content in the root
zone drops below a certain threshold root water uptake is controlled by the difference in
matric and root water potentials.

**2.3.2 Generation of rapid vertical and lateral flow paths**

Vertical and lateral preferential flow paths are represented as connected flow paths
containing an artificial porous medium with high hydraulic conductivity and very low
retention properties. This approach has also been followed by others (Nieber and Warner
1991; Castiglione et al. 2003; Lamy et al. 2009; Nieber and Sidle 2010), and was proven



to be suitable to predict hillslope scale preferential flow and tracer transport in the
Weiherbach catchment, a drained and agriculture dominated site in Germany (Klaus and
Zehe, 2011) and at the Heumöser hillslope, a forested site with fine textured marly soils
in Austria (Wienhöfer and Zehe, 2014). A Poisson process allocates the starting points of
the vertical structures sequentially along the soil surface and stepwise extends the vertical
preferential pathways downwards to their depth, while allowing for a lateral step with a
probability of typically 0.05 to 0.1 to establish tortuosity. Lateral preferential flow paths
to represent either pipes at the bedrock interface or the tile drains are generated in the
same manner: starting at the interface to the stream and stepwise extending them upslope,
again with a small probability for a vertical upward or downwards step to allow for a
tortuosity.
**3. Parametrization of the representative hillslope models**
**3.1 Colpach catchment**
*Surface topography, spatial discretization and land use*
We selected a real hillslope profile with a length of 350 m, a maximal elevation above the
stream of 54 m and a total area of 42600 m², which represents the average elevation
difference and lengths of all delineated 241 hillslopes in the Colpach catchment (Figure 6
A). The hillslope was discretized with a 1 m grid size in the downslope and 0.1 m grid
size in the vertical direction down to the depth of 2 m (Figure 3C). Land use was set to
mixed forest for the entire hillslope since forest covers 65 % of the catchment area,
especially the steeper hillslopes (Figure 3). Due to the absence of local data on the plant
phenological cycle we used fixed tabulated data characterizing the annual cycle of leaf
area index, ground cover, root depth, plant height, minimum stomata resistance and plant
roughness (Manning's n) derived within the study of Zehe et al., (2001) for the
Weiherbach catchment. We are aware that particularly the onset of the vegetation phase
depends strongly on the local climate setting. As such, this assumption may be reflected
in model bias during this period of the year; we explore the sensitivity of model results
for an improved estimate of the start of the vegetation period within the virtual
experiment 3. Boundary conditions were set to atmospheric boundary at the top, no flow
boundary at the right margin, free outflow boundary condition at the left boundary and a
gravitational flow boundary condition at the lower boundary. We used spin-up runs with



initial states of 70 % of saturation for the entire hydrological year of interest and used the
final soil moisture pattern for model initialization.

*Bedrock-topography and permeability, rapid subsurface flow paths and soil hydraulic*
*functions*
The shape of the bedrock interface was extracted from the ERT image based on the 1500
$\Omega$/m contour line (Figure 6 B and Figure 3c). As a result, the soil depths in the hillslope
varied between 1 m to 1.8 m with local depressions that form water holding pools. Since
no quantitative data on bedrock permeability are available we use the relative
impermeable bedrock parametrization of Wienhöfer and Zehe (2014) but increased the
bedrock porosity from 0.35 to 0.4 to account for storage in fractures and cracks (Table 1).
The silty soil above the bedrock was characterized by the representative hydraulic
parameters obtained from field samples listed in Table 1. Macropore depths were drawn
from a normal distribution with mean 1 m and standard deviation 0.3 m in agreement with
the reported mean soil depth and dye staining experiments. Additionally, macropores
were slightly tortuous with a probability for a lateral step of 5 %. Conductivity of vertical
macropores was set to $5 \times 10^{-2}$ m s$^{-1}$, which corresponds well with the observation for
earthworm burrows (Shipitalo and Butt, 1999). Weathered Schist at the soil–bedrock
interface was represented by a 0.2 m thick layer using the parameter of the macropore
medium from Wienhöfer and Zehe, (2014; Table 1). Stochastic heterogeneity of soil
hydraulic properties was accounted for by perturbing the saturated hydraulic conductivity
with multiples generated with a two dimensional turning band generator (Zehe et al.
2010a). As proposed in (Zehe et al., 2010b) we used a rather short range of 5 m and a
nugget to sill ratio of 0.75.
**3.2 Wollefsbach catchment**
*Surface topography, spatial discretization and land use*
Topography and land use is much more uniform in the Wollefsbach compared to the
Colpach. We again selected a hillslope with an area of 373600 m², which is 853 m long
and has a maximal elevation of 53 m above the river. The hillslope was discretized into
550 horizontal and 20 vertical elements with an overall hillslope thickness of 2 m (Figure
3 D). The vertical grid size was in general set to 0.1 m while grid size in downslope
direction varied between 0.1 m within and close to the rapid flow path and 2 m within
reaches without macropores (Figure 3). Land use was divided into grassland within the




steeper and lower part of the hillslope and, after 425 m from the creek, to corn cultivation.
This configuration reflects well the typical land use pattern in the catchment. Due to the
absence of local data we used fixed tabulated data characterizing grassland and corn
derived within the Weiherbach catchment (Zehe et al., 2001).

*Bedrock-topography and permeability, rapid subsurface flow paths and soil hydraulic*
*functions*
Contrary to the Colpach, geophysical measurements and augers revealed bedrock and
surface as being more or less parallel. Soil depth was set to constant 1 m and the soil was
parameterized using the representative soil retention curves shown in Figure 7. Marly
bedrock was again parameterized according to values Wienhöfer & Zehe (2014) proposed
for the marly bedrock at the Heumöser hillslope (Table 1). Because of the vertical and
lateral drainage structures in the catchment we generated a rather dense network of
vertical fast flow path with the structure generator, which were partly connected to a 10
cm thick tile drain. The latter was generated in the standard depth of 80 cm assuming that
it extended 400 m upslope originating from the hillslope creek interface. The position of
macropores was selected with a Poisson process along the soil surface with a minimum
distance of 3 m. Vertical flow path and the pipe were parametrized using an artificial
porous medium (Table 1) proposed by Wienhöfer and Zehe, (2014). Boundary conditions
were set to atmospheric boundary at the top, no flow boundary at the right margin and a
free outflow boundary condition and a gravitational flow boundary condition were
prescribed at the left boundary and the lower boundary. The model was initialized in the
same manner as described for the Colpach.
**3.3    Model benchmarking**
Both reference hillslopes (A1 reference model Colpach; A2 reference model
Wollefsbach) were set up to reproduce the normalized double mass curves in both
catchments of the hydrological year 2014 within which a few test simulations were
judged using the Kling-Gupta efficiency (KGE) (Gupta et al. 2009). In the next step we
compared the sum of simulated overland flow, subsurface storm flow across the right
hillslope boundary and deep percolation across the bottom boundary to observed stream
flow hydrograph, both visually and also based on the KGE, the Nash-Sutcliffe efficiency
(NSE) and the logarithmic NSE (log NSE). Furthermore, we validated the model setups



without any tuning against available soil moisture observations as well as against sap
flow velocities observed in the in the Colpach.

### 3.3.1    Virtual experiments

We performed three virtual experiments in the Colpach by changing surface and bedrock
topography as well as plant physiological cycles to explore first-order controls on the
water balance and the relative importance of the underlying observations. Additionally,
we performed a virtual experiment in the Wollefsbach by increasing the hydraulic
conductivity of the soils in the summer months to account for the effect of emerging soil
cracks on runoff generation.

*VE1 Colpach Experiment 1, double gradient:* The reference representative hillslope (A1)
was selected to match the average hillslope length and elevation range of 241 delineated
hillslope profiles in the Colpach catchment (Figure 6 A). To check how strongly the
simulated stream flow depends on the gradient in relief energy we increased the elevation
range by a factor of two, yielding a maximum elevation of 108 m above the creek by a
preserved hillslope length.

*VE2 Colpach Experiment 2, bedrock topography:* To shed light on the role of bedrock
topography we compared four additional simulations with the reference model setup
(A1). In the second hillslope the 0.2 m thick bedrock interface was removed (VE2.1). In
the third hillslope we removed the bedrock-interface and assumed parallel bedrock
topography with the surface at 1 m depth (VE2.2). The fourth hillslope likewise has
surface bedrock topography at 1 m depth, but we added a no-flow boundary condition for
the lower 70 % of the right boundary to create a depression which has approximately the
same volume as all depressions in the reference slope (VE2.3). This approach is expected
to create a small storage volume that might create subsurface runoff by filling and
spilling, as shown by Graeff et al., (2009). Last not least, we used the reference hillslope
but removed the vertical macropores (VE2.4).

*VE3 Colpach Experiment 3, changed onset of phenological cycle:* The reference hillslope
employed a fixed phenological cycle for mixed forest, which was derived in the
Weiherbach catchment (Zehe et al., 2001). However, the start and end of the phenological
cycle is subject to inter-annual variations which can be described well by a temperature





index model proposed by Menzel et al., (2003). Hence, we changed the onset and end of
the vegetation period in the parameter table in accordance with the temperature index
model, while keeping unchanged the other temporal patterns in-between, like the leaf area
index or root depth.

*VE4 Wollefsbach Experiment 4, challenge of dealing with emergent soil structures:* The
fact that either worm burrows in spring/ fall or shrinkage cracks during dry spells in
midsummer control the soil hydraulic behavior and runoff formation in the Wollefsbach,
implies that the vertical flow paths in the model structure should be time-, or more
precisely,   state-dependent.   As   CATFLOW   does   not   treat   soil   hydraulic
parameters/macropores as state dependent but as constant, one cannot expect a rigid
parameter setup to reproduce the hydrographs in winter and summer in an acceptable
manner. This expectation was corroborated by our first simulation carried out with the
reference model for the Wollefsbach (A2). We hence tested the additional value of a state
dependent soil structure in a first tentative approach by subdividing the simulation period
into a winter and summer period, based on the temperature index model of Menzel et al.,
(2003). The winter period was simulated with the setup specified in the reference model
(A2) until the onset of the summer period. We then started a second simulation with an
elevated infiltrability using the final state of the first simulation as initial state.
Infiltrability was increased by an enlarged hydraulic conductivity of the upper 100 cm by
a factor of 25, 50 and 75 within three different model runs. Since the model run with the
factor 75 yielded the best results with respect to the discharge (estimated on KGE), we
used this model setup for the vegetation period in virtual experiment 4.

## 4. Results

### 4.1 Double mass curves and stream flow time series

As depicted in Figure 8 A and C the reference hillslope models A1 and A2 reproduced the
typical shape of the normalized double mass curves – the steep, almost linear increase in
the winter period and the transition to the much flatter summer regime – in both
catchments very well. This is further illustrated by the comparison of two typical
saturation patterns simulated for winter and summer in the Colpach (Figure 10 B). Local
bedrock depressions are close to saturation during the winter season, and hence ready to
create runoff by filling and spilling in response to rapid infiltration through vertical



macropores. These pools are, however, fairly dry during summer after a period of
transpiration and root water uptake. The absence of free water-sustaining fill and spill
runoff generation explains the flat summer regime in the double mass curves and thus the
dominant role of transpiration which may be sustained even by strongly bound soil water.
The KGEs of 0.87 and 0.94 in the Colpach and Wollefsbach corroborate that within the
error ranges both double mass curves are almost perfectly explained by the models. We
may hence state that the hillslope models closely portray the seasonal pattern of the
catchment's water balance, also because simulated and observed annual runoff
coefficients match very well (Table 2). A closer look at the simulated and observed runoff
time series (Figure 8 B and D) indicates, however, that the models performed differently
in both catchments in the winter and the summer season. Generally we observed in both
catchments a better matching of observed stream flow during the wet winter period and
clear deficiencies during dry summer conditions. For instance, the reference Colpach
model (A1) misses the steep and flashy runoff events in June and July (Figure 8 B). Note
that the representative hillslope model (A1) of the Colpach also performed well with
respect in matching the hydrograph of the previous year, with an NSE of 0.72.
Furthermore, the parameter setup was tested within uncalibrated simulations for the
Weierbach, a headwater in the same geological setting, again with acceptable results of a
KGE of 0.81 and a NSE of 0.68.
In the Wollefsbach we found visually even stronger seasonal differences in model
performance with respect to matching the hydrograph. This is further shown by the low
overall model performance of 0.26 for the NSE and 0.64 for the KGE in contrast to a
relative high logNSE of 0.82 (Table 2). The differences in the logNSE and NSE further
explain the strong influence of the overestimated runoff events in August and September
on the NSE that have almost no volume because of their short duration as a result of the
flashy runoff regime of the catchment. The reference model (A2) led to a strong
overestimation of runoff behavior in summer due to strong overland flow production
during convective events in August and September (Figure 8 D). No significant overland
flow was simulated during the rest of the year. This suggests that without the
representation of cracks by using separate summer macroporostiy (VE4), soil infiltrability
is obviously too small compared to the real system as soon as cracks have emerged.





**4.2 Simulated and observed soil moisture dynamics**

It is of great interest as to what extent the model performance in reproducing soil moisture or sap flow observations is consistent with the strengths and deficiencies of the simulated hydrographs and double mass curves. Clearly we cannot expect that our simulations are capable of reproducing the observed spatial variability of both distributed data sources, as this needs a fully distributed model setup, which may have to account for the variability of the terrestrial filter properties and of the rainfall and radiation forcing within the catchment. A representative hillslope should, however, be ergodic and hence match the averaged temporal dynamics and plot within the envelope of available soil moisture observations, which are displayed in Figure 9 for both catchments and both observation depths. Note that the spread in the observations in 10 cm is much larger in the Colpach catchment, particularly in the dry months of June and July, while the opposite holds true for soil moisture observations at 50 cm depth. This suggests that the differences in runoff generation in the two catchments might also arise from differences in storage and storage-related controls.

To estimate the representative soil moisture dynamics from the distributed observations we employed a twelve-hour rolling median, which we compared to the spatially averaged soil moisture simulated at the respective depths. Model goodness was determined visually and by means of the KGE and the Spearman Rank correlation. Simulated soil moisture at 10 cm depth for the Colpach was systematically too high by around a volume of 25 %. Yet the simulation is within the envelope that is spanned by the observations and the KGE of 0.69 suggests some predictive power for top soil moisture. Also soil moisture dynamics are matched well with a spearman rank correlation of 0.8. This holds true especially if compared to the median of spearman rank correlations of all sensor pairs, which is 0.66. In line with the findings presented in the previous section, the model showed deficiencies in the summer period with respect to capturing the strong declines in soil moisture in June and July. Simulated soil moisture at 50 cm exhibited a strong positive bias as it is above the envelope spanned by the observations and has no predictive power (KGE -0.89). Yet the changes in deeper soil moisture storage are in good accordance, as reflected by the spearman rank correlation of 0.91.





Contrary to what we found in the Colpach, the simulation in 10 cm underestimated the
rolling median of soil moisture time series observed in the Wollefsbach (Figure 9 C); yet
it fell into the state space spanned by the observations. The predictive power is with a
KGE of 0.45 worse than in the Colpach, while the match of the temporal dynamics is,
with a rank correlation coefficient of 0.68, also slightly poorer. Again the model failed in
reproducing the strong decline in soil moisture between May and July. It is, however,
interesting to note that the model is nearly unbiased during August and September. This is
especially interesting since the reference model does not perform well with respect to the
discharge simulations in this time period. Simulated soil moisture at 50 cm depth showed
similar model deficiencies as found for the Colpach, while the bias was slightly smaller.

**4.3 Normalized simulated transpiration versus normalized sap flow velocities**

As sap flow provides a proxy for transpiration, we compared normalized, averaged sap
flow velocities of all beech and oak trees to the normalized simulated transpiration of the
reference hillslope model of the Colpach. Mean sap flow stayed close to zero until the end
of April and started to rise after the predicted onset of the vegetation period. In contrast,
simulated transpiration is already slightly above zero in January and exhibits a first peak
at the beginning of March. This indicates that the predefined phenological cycle is, as
expected, inappropriate for matching with the first sprout in this area. Simulations and
observations are in good accordance during midsummer. In the period between August
and October the simulations underestimate the observations, because the pre-defined
phenology declines too early. These deficiencies of the model to properly match onset
and end of the vegetation period are reflected in the rank correlation coefficient of 0.62.
Due to the fact that periods of high transpiration and sap flow activity have, while being
out of phase, a similar length, the model still matches the accumulated annual
evapotranspiration well and the annual water balance is reproduced properly.

**4.4 Virtual experiments to search for first-order controls**

*VE1 Colpach Experiment 1, double gradient:* Opposite to our expectation we found that
the doubling of the topographic gradient had only a minor effect on the simulated double
mass curves and hydrographs, respectively (Figure 11 F). The KGE decreased from 0.85
to 0.8 while logNSE increased from 0.75 to 0.85. In fact runoff peaks were slightly
smaller in the winter period compared to the reference hillslope model. While this appears



counterintuitive on first sight, it may be attributed to the decrease in subsurface storage
volumes in the bedrock depressions in the steeper hillslope.

*VE2 Colpach Experiment 2, bedrock topography:* Contrary to the VE1 this finding and in
line with our expectation, virtual experiment 2 revealed the key importance of bedrock
topography. The removal of the soil bedrock interface, which allowed for rapid flow in
the weathered schist, reduced the NSE to 0.8 and the KGE even more to 0.7 (Table 2),
mainly due to a reduction in simulated flood peaks (Figure 11 B). The key role of rapid
flow paths is further corroborated by the model run (VE 2.4) which left out the vertical
macropores from the reference setup (Figure 11 E); again simulated peaks are too small
and the KGE is reduced to 0.7. The removal of the bedrock depression by assuming
surface-parallel bedrock topography did completely change the simulated rainfall runoff
behavior (Figure 11 C). The corresponding hydrograph appears very much like a
separated base flow component due to the absence of peaks and high frequency
components; this is in line with the clear reduction of the NSE to 0.56 and the KGE to
0.59 (Table 2).

Most interestingly we found that the hillslope with the conceptualized depression in the
hill foot/riparian zone performed even marginally better than the reference setup,
particularly with the better match of the largest summer flood in August (Figure 11 D).
The corresponding KGE of 0.91 and NSE of 0.88 (Table 2) suggest a nearly perfect
performance within the error margins. We hence state that the results of virtual
experiment 2 are fully in line with our expectations for a system which works according
to the fill and spill concept. Yet, they provide a surprise, which is not so much the finding
that several bedrock architectures perform similarly well, but that the location and
connectedness of the bedrock seems to be of minor importance at this scale, as further
explained in the discussion.

*VE3 Colpach Experiment 3, changed onset of phenological cycle:* Our third virtual
experiment revealed that the improved representation of the onset and end of the
phenological cycle based on the temperature index model of Menzel et al., (2003) clearly
improved the model performance with respect to matching the observed onset of sap flow
activity (Figure 10 A). This went along with an improved simulation of the double mass
curves (Figure 12 A). Given a KGE of 0.95 and the fact that simulated and observed



annual runoff coefficients are with 0.54 and 0.56 in almost perfect accordance, this setup
is the most behavioral one of all tested setups (Table 2). A closer look at the simulated
hydrograph reveals also a clear improvement, particularly with respect to the logNSE
(Table 2). The more realistic runoff reaction in early summer can be explained by the fact
that bedrock depressions are still wet enough to create runoff due to the later onset of
transpiration. Yet the model still showed deficiencies, for instance, the overestimation of
the first runoff events in May and the underestimation of subsequent events in June and
July. The clear improvements which were already caused by this minor and
straightforward implementation of the dynamic onset of plant phenology suggests that
much more substantial improvements of transpiration controls might be a key for
substantially improving hydrological models.

*VE4 Wollefsbach Experiment 4, challenge of dealing with emergent soil structures:* The
virtual experiment in the Wollefsbach which uses a separated summer soil hydraulic
conductivity revealed and improved strongly matching runoff (Figure 8 D); this is
reflected in the increase in KGE from 0.64 to 0.76 and NSE from 0.26 to 0.74 compared
to the reference model (A2s). However the simple quick fix of multiplying the hydraulic
conductivity by 75 has also clear drawbacks, as the value was selected by manual
calibration on discharge. The model performance with respect to soil moisture became
clearly worse as shown in Figure 9 C and D. The overall spearman rank correlation
decrease to 0.51 at 10 cm depth, which can be explained by the fact that water flushes
through the soil at 10 and 50 cm to the bedrock interface due to the increased hydraulic
conductivity of the entire soil profile. We may hence state that such a quick fix of
increasing the hydraulic conductivity of the top soil to improve simulated stream flow
production is inappropriate for capturing the effect of emergent soil cracks for the right
reasons.
**5 Discussion**
This study provides strong evidence that physically based simulations with representative
hillslope models may closely portray water storage and release behavior of two distinctly
different lower mesoscale catchments. In line with the blueprint of Freeze and Harlan,
(1969) we disclaimed any form of calibration of the underlying model. Instead the model
structures were set up as one-to-one images of our perceptual models of the catchment





structure and the dominant processes, using a variety of different data and process
insights. Hillslope models were then parameterized according to the available very
diverse data as well as from the literature, within a handful of test simulations and
benchmarked against different response and storage measures without any additional
parameter tuning. In the following we discuss what can be learned from the successful
part of this approach with respect to a) the widely discussed role of soil heterogeneity and
preferential flow paths, b) the potential of physically based models to accommodate and
thus integrate multi-dimensional data, and c) the validity of the assumptions underlying
the representative hillslope concept. We then discuss the feasibility of the concept of
functional units and related to this, the partly astonishing findings we obtained within
three virtual experiments. Finally, we change perspective on the various deficiencies and
discuss the related inherent limitations of the representative hillslope concept itself and,
more importantly, the limitations of the theories underpinning physically based models,
with emphasis on bio-physical processes and emergent behavior. We conclude the
discussion with a final reflection on our notion of complexity and simplicity.
**5.1 Can a representative hillslope model mimic the functioning of a catchment?**
The attempt to model catchment behavior using a two-dimensional representative
hillslope implies a symmetry assumption in the sense that the water balance is dominated
by the interplay of hillslope parallel and vertical fluxes and the related driving gradients
(Zehe et al., 2014). This assumption is corroborated by the good but yet seasonally
dependent performance of both hillslope models with respect to matching the water
balance and the hydrographs. We particularly learn that the timing of runoff events in
these two catchments is controlled by the structural properties of the hillslopes,
particularly bedrock topography in concert with soil permeability. This is remarkable for
the Colpach catchment which has a size of 19.4 km$^2$, but in line with Robinson et al.,
(1995) who showed that catchments smaller than 20 km$^2$ are hillslope dominated. It is
obvious that a representative hillslope model cannot perform well as soon as the time
scales of flood routing and rainfall variability start dominating the response behavior and
timing of stream flow.

With respect to matching observed storage behavior both hillslope models had clear
deficiencies in simulating the observed average soil moisture data. While simulated and
observed average soil moisture dynamics were satisfyingly matched within the topsoil,





both models were biased at 50 cm depth. This might be explained by the fact that we used
a single soil type for the entire soil profile while porosity in the deeper ranges is most
likely lower (e.g. higher skeleton fraction). Particularly, the models failed in reproducing
the strong decline in soil moisture between May and July 2014. A likely reason for this is
that plant roots in the model extract water uniformly within the root zone, while plants
possibly optimize the amount of energy they have to invest to access soil water sustaining
transpiration (Hildebrandt et al., 2015). We also found that benchmarking of the model
against sap flow data provided additional information about the representation of
vegetation controls, which cannot be extracted from the double mass curve or discharge
data. We hence conclude that the proposed concept offers various opportunities for
integrating diverse field observations and testing their hydrological consistencies,
including soil water retention data and images from geophysics and dye staining
experiments. We further conclude that the idea of hillslope-scale functional units, which
act similarly with respect to runoff generation and might hence serve as building blocks
for catchment models, has been corroborated. This is particularly underpinned by the fact
that the parameterization of the reference model was – without tuning – transferable to a
headwater in the same geological setting. In this respect we were astonished by the fact
that the huge observed variability of soil water retention properties could be represented
in a rather straight forward manner.

### 809  5.2 Is subsurface heterogeneity a dead end for a representative hillslope model?

Both hillslopes were parametrized using a representative soil water retention curve using
experimental data from either 53 or 28 undisturbed soil cores within a maximum
likelihood approach. Since the two hillslope models are able to successfully simulate the
water balance, we may in line with Ebel and Loague, (2006) conclude that heterogeneity
of retention properties is not too important for reproducing runoff of lower mesoscale
catchments. This finding is also in line with the common knowledge that stream flow is a
low dimensional variable, which may be reproduced by a three to four parameter model.
We are fully aware that spatial variability of retention properties might be of importance
when dealing with solute transport or infiltration patterns in catchments. We also like to
stress that our approach may support the optimization of soil sampling with respect to the
minimum sample size that is needed to robustly determine a representative retention
curve for water balance modelling. We leave this here for further virtual experiments.





The fact that the overwhelming heterogeneity of subsurface must not be a dead end for
the idea of a representative hillslope model can also be concluded from our representation
of preferential flow. The straightforward implementation as connected flow paths
containing an artificial porous medium was sufficient to reproduce runoff generation and
the water balance for the whole time period in the Colpach and for the winter period in
the Wollefsbach catchment. This becomes more evident when looking at Figure 11 B and
F where the removal of either the bedrock interface or the vertical macropores reduced
the KGE from 0.84 to 0.72 and 0.71, respectively. Despite the known limitations of this
approach, or in fact of all approaches to simulate marcopore flow (Beven and Germann,
2013), it seems that capturing the topology and connectedness of rapid flow paths is
crucial for setting up representative hillslopes at this scale. We also may conclude that
particularly the hydraulic properties of the macropores observed at other places
(Wienhöfer and Zehe, 2014; Klaus and Zehe, 2010) can be transferred across system
borders. But this fact should not be misinterpreted; it is not the exact position or the exact
extent of the macropore setup which is important but rather the hydraulic behavior of the
macropore medium and the idea of a connected flow path topology. We are aware that
such flow path topology is not uniquely defined, as shown by Wienhöfer and Zehe
(2014). It seems that equifinality and the concept of a representative hillslope is rather
more a blessing than a curse since there is an infinite number of possible macropore
setups which yield the same runoff characteristics. If this were not the case, we could not
transfer macropore setups from the literature across system borders and successfully
simulate two distinct runoff regimes which are strongly influenced by preferential flow.
**5.3 Hillslope scale structures as first order control on water storage and release**
**5.3.1 Is surface topography a first order control in the Colpach?**
The potential energy difference is a major control on runoff as runoff generation is mostly
driven by gravity. Nevertheless, surface topography can only be a poor means to
characterize relevant potential energy differences for runoff generation. For instance
Jackisch (2015) showed that variables extracted from topography could not explain
differences in runoff generation between the three geological settings of the Attert
catchment. Extracted topographic variables such as the topographic wetness index
suggested that two catchments, which produce runoff in a distinctly different fashion,
should operate similarly. Furthermore, Fenicia et al., (2016) report in the same research





area that a conceptual model built upon geological hydrological response units (HRU)
performed in a superior fashion with respect to explaining spatial variability of the
streamflow than a model built upon topographic HRUs. In line with this we found that a
doubled gradient in relief energy had almost no influence on the simulated water balance
in the Colpach, despite a marginal reduction of simulated flood peaks. The
counterintuitive peak reduction reflects the decrease in subsurface storage volume in the
bedrock depressions in the steeper hillslope. From this finding we may conclude that
runoff production in this highly permeable and highly porous soil setting is not limited by
the topographic gradient, as its doubling did not substantially increase subsurface flow
velocities. Nevertheless further virtual experiments with reduction of the topographic
gradient are needed to estimate the lower threshold above which simulations become
insensitive to gradient changes.

### 5.3.2 Is bedrock topography a first order control in the Colpach?

Contrary to the findings regarding surface topography our simulations were highly
sensitive to changes in bedrock topography and the presence/absence of a bedrock
interface and macropores. Particularly, the assumption of surface-parallel bedrock
changed the response behavior of the hillslope completely, by produced a hydrograph
which looked like a separated base flow component. Our findings fully corroborate the
argumentation of Wrede et al. (2015) who stated that the dominant runoff generation in
the Colpach works according to the fill and spill mechanism (Tromp-Van Meerveld and
McDonnell, 2006). In line with Hopp and McDonnell, (2009) we also found that local
depressions in bedrock topography and a varying soil depth jointly control filling and
spilling during subsurface runoff production. The real surprise of this virtual experiment
was, however, that the catchment scale fingerprint of a fill and spill system can be
simulated in several ways. One alternative was a spatially explicit representation of
bedrock topography as observed with electrical resistivity tomography at two typical
hillslopes, which implies that connectivity between these depressions depends on
subsurface wetness (Lehmann et al., 2006). The other feasible alternative was a hillslope
that combined surface-parallel bedrock topography with a large depression at the hill foot
sector, which had the same total storage volume as the explicit bedrock topography. This
second alternative produced a small riparian zone and performed even slightly better with
respect to matching the hydrograph. Hence we may conclude that it is not the detailed
representation of the subsurface architecture, including location of the depressions along





the slope line, but the correct representation of the total volume of all depressions which
makes up the behavioral catchment model. The last finding has important implications for
geophysical explorations of the shallow subsurface, as it is likely possible to extract a
representative catchment average storage volume from geophysical data at a few
hillslopes and translate this into a conceptualized storage volume in the model. This
procedure is most likely sufficient, which is promising as geophysical exploration of the
subsurface of each individual hillslope in a catchment is unfeasible.

### 895 5.3.3 Is vegetation a first order control in the Colpach?

The model performance with respect to matching the average sap flow dynamics and
particularly the results of the related virtual experiment 3 corroborates that the
phenological cycle of beech trees is a first order control on the water balance in this area.
As the onset of the vegetation phase depends strongly on the local climate setting, our
first simulation based on predefined phenological cycles was biased when the system
switched from dominance of abiotic controls to dominance of bio-physical controls. In
line with Seibert et al. (2016) we found that a simple temperature index proposed by
Menzel et al., (2003) is feasible for both detecting the tipping point between summer and
the winter regime in the double mass curve and to determine start and endpoint of the
vegetation phase in the model. This is underpinned by the model results with the
improved onset of the vegetation which showed indeed only small improvement in the
simulation of the discharge but an almost perfect fit of the normalized double mass curves
with a KGE of 0.95. As the SVAT modules need air temperature anyway, a dynamic
phenological cycle is straightforward to implement in CATFLOW.

### 910 5.4 Limitations and challenges

### 911 5.4.1 The need to go distributed

An example of the limitations of our single hillslope approach is the deficiencies that both
models encountered to capture flashy rainfall runoff events in the vegetation period.
Besides the existence of emergent structures, these events might likely be caused by
localized convective storms, probably with a strong contribution of the riparian zones
(Martínez-Carreras et al., 2015) and forest roads in the Colpach catchment and by
localized overland flow in the Wollefsbach catchment (Martínez-Carreras et al., 2012).
Such fingerprints of a non-uniform rainfall forcing cannot be captured by a simulation
with a single hillslope; it requires a distributed model setup and a river network.



Similarly, one cannot expect a single hillslope to reproduce the observed variability of
soil moisture and sap flow, which reflects the interplay of the spatial pattern in rainfall
and radiative forcing in combination with the heterogeneity of local soil and land use
properties. Yet we suggest that a representative hillslope model structure provides the
right start-up for a subsurface parameterization, i.e. a functional unit when setting up a
fully distributed catchment model consisting of several hillslopes and an interconnecting
river network. Simulations with distributed rainfall and using the same functional unit
parameterization for all hillslopes would tell how the variability in response and storage
behavior can be explained compared to the single hillslope. In the case where different
functional units are necessary to reproduce the variability of distributed fluxes and storage
dynamics, these can either be generated by further perturbation or one might choose to
measure the key variable in accordance with the virtual experiments. We also suggest that
our statistical approach to compare simulated and observed soil moisture and sap flow is
more appropriate than a point/grid cell-based evaluation. The latter technique would only
be appropriate in the case of perfect observations of rainfall and terrestrial properties with
perfect spatio-temporal resolution.
**5.4.2 The inherent limitations of the Richards equation**
The presented simulations are affected by inherent limitations of the Richards equation
and the approach we used to represent rapid flow. The major deficiency of the Richards
equation is the local equilibrium assumption and related to this, the assumption that
gravity driven flow and capillarity controlled flow are always controlled by the same
Darcian frictional loss term $k\,(\theta)$. This assumption can and should be relaxed particularly
during strong rainfall forcing, for instance by treating gravity flow as a separate advection
term. The related key challenges are not only the numerical solution of such an advection
diffusion equation but the proper representation of the energy loss for the advection and
the fact that kinetic energy of soil water flow may no longer be neglected. The limitations
of the Richards equation should not be mistaken for non-importance of capillary forces in
soils: there would be no soil water storage and probably no terrestrial vegetation without
capillary forces. Capillarity is a first-order control of soil water dynamics during radiation
driven conditions and the Darcy-Richards concept is sufficient for simulating soil water
dynamics in this context (Zehe et al., 2010, 2014).
Any physical theory or model, while being based on inference, is yet an empirical fit to
observables with its purpose to explain a class of phenomena as broadly as possible





(Popper, 1935). All physically-based models we use today are incomplete in the sense
that there are phenomena which fall outside their range of validity. A representative
hillslope model remains hence "a lousy model" as it needs to rely on an incomplete
description of the water fluxes in the catchment. Nevertheless, there is much to be learned
from imperfect solutions, as long as we are aware of their advantages and drawbacks and
as long as we stick to the blueprint of Freeze and Harlan (1969). Our study shows that
beside the usually discussed limitations, the real challenge is to deal with emergent
behavior. This is in fact a straightforward implication of the fact that both soil structure
and vegetation structure are rather "slowly" varying morphological state variables of the
hydrological system than constant/ stationary parameters. Soil structures in the clay-rich
soils of the Wollefsbach are non-stationary due to the active/ dormant phase of earth
worm burrows and due to crack formation during dry spells. Our use of summer and
winter macroporosity was a tentative first step to show the need and the potential of
treating hydraulic properties not as static but dynamic. A proper representation of such
non-linear feedbacks in physically based models is a two-fold challenge with respect to
gaining the necessary understanding of the underlying thresholds and with respect to
stability of the numerical solution when dealing with such a non-linear feedback.
**5.4.3 Bio-physical controls on transpiration –the weakest link in the chain?**
Numerical experiment 3 (VE3) corroborated that the improved representation of the onset
of the vegetation phase as a temperature dependent variable yielded superior simulations
of all forms of water release. While this was straightforward to implement, it remains a
tentative first step, because the evolution of plant morphological parameters remains the
same within every year. We argue that much more substantial improvements of bio-
physical controls might be the key for further improving the fundamental basis of our
models. The ideal solution is of course coupling with a dynamic vegetation model, as
proposed by Schymanski et al., (2008) or Tietjen et al., (2010) for pristine vegetation in
semi-arid areas. This implies that phenology evolves in response to climate and
hydrological controls, thereby creating feedbacks. It furthermore offers more realistic
ways for implementing root water uptake, as a distributed process along the root zone for
instance by optimizing the amount of energy plants have to invest to access soil water
(Hildebrandt et al., 2015). The challenge is, however, that ecological limitations in humid
regions are less obvious, which implies that these models cannot simply be transferred to
our regions. The coupling of hydrological models and crop models which have been



developed in an independent fashion is furthermore not a straightforward exercise
(Jackisch et al., 2014). Despite these challenges, we need a paradigm shift in accepting
that transpiration is part of the plant's metabolism, gas exchange and photosynthesis and
thus reflects (optimal) behavior of plants (Schymanski et al., 2009) rather than plants
acting as a water pump. The literature is full of different and much more realistic models
for parameterizing stomata conductance (Damour et al., 2010) that step beyond the heavy
parameterization of Jarvis (1972), which reflects the outworn paradigm of the "plant as
water pump".
**6. Conclusions**
We overall conclude that perceptual models of catchments can be translated into
representative hillslope models which successfully portray the spatial aggregated
functioning. This was demonstrated for two distinctly different catchments and implies
that hillslope-scale functional units might indeed be identified as building blocks for
catchments and that their behavioral parameter sets are transferable within the same
landscape setting. The general idea to translate a perceptual model into a model structure
is not new and has already been applied with a conceptual rainfall-runoff model
framework even within the same area (Wrede et al., 2015). Here the scientific asset is that
we use a physically-based model which can be parameterized directly on field
measurements since the parameters are directly related to the physical processes and their
controlling structures. The drawback is that physically-based models are parameter
intensive, limited due to the incomplete representation of physics and are data greedy
which means that they need to be based on extensive field data which is only available in
a few research catchments around the world. But what might seem like a drawback is also
to their benefit since their simulations reflect naturally both the strength and limitations of
the underlying representations of bio-physical processes. This offers the option to learn
not only from the successful part of the study but particularly also from model
deficiencies as they unmask limitations in the theoretical underpinnings and of the
representative hillslope concept itself. We conclude that using physically-based models,
even in such an aggregated and abstract manner provides insights into cause and effect
relationships explaining what happens due to which reasons. This cannot be obtained to
such an extent by conceptual models simply because they have been designed to
reproduce rainfall-runoff relations as parsimoniously as possible. Their advantage is the



simpler mathematical formulation and thereby a strong reduction in the number of
parameters. This fact makes them easy to calibrate and hence the first choice for studies
in data sparse catchments as well as for comparative hydrology. Nonetheless, a simple
mathematical representation does not automatically mean that the model structures and
the parameters are straightforward to interpret. This is evident not only in the fact that
internal states and parameters of conceptual models are difficult to compare and derive
from field observations, but also that their model structure is neither simple nor intuitive
to interpret. As a consequence, model improvements and model comparisons often need
to be benchmarked purely on a statistical foundation. Physically-based models are not
based on simple mathematical equations, but their setup reflects much more intuitively
the perception of the system, particularly in the subsurface.
Therefore, we close with the note that simplicity in hydrology should not be mistaken for
a mathematically simplified process description, parameter parsimony or minimum of
model elements. To us it rather means achieving the perfect level of abstraction of the
system we want to analyze – perfect in the sense of Antoine de Saint-Exupéry[1] –
balancing necessary complexity with the greatest possible simplicity. A behavioral
representative hillslope is hence not a simple average of all the existing hillslopes in a
catchment but a behavioral, maybe not yet perfect, abstraction of a catchment – a picture
in the sense of Heidegger.


---

[1] *"A perfect picture is not achieved when there is nothing left to be added, but when there is nothing left to be taken away."* Antoine de Saint-Exupéry



*Acknowledgements*
This research contributes to the "Catchments As Organized Systems (CAOS)" research group (FOR 1598)
funded by the German Science Foundation (DFG). Laurent Pfister and Jean-Francois Iffly from the
Luxembourg Institute of Science and Technology (LIST) are acknowledged for organizing the permissions
for the experiments and providing discharge data for Wollefsbach and Colpach. We furthermore thank the
CAOS team of phase I, in particular subproject G, for providing soil moisture data as well as Malte Neuper
for support, validation and discussions on the quality of the precipitation data.





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





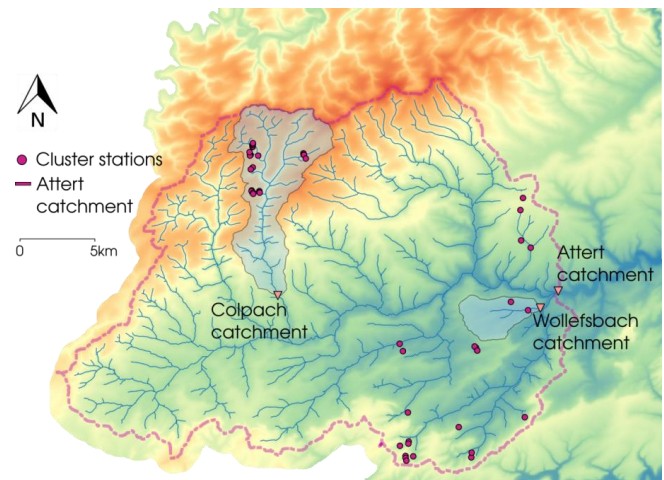


**Figure 1 Map of the Attert basin with the two selected headwater catchments of this study (Colpach and**
**Wollefsbach). In addition, the cluster sites of the CAOS research unit are displayed.**




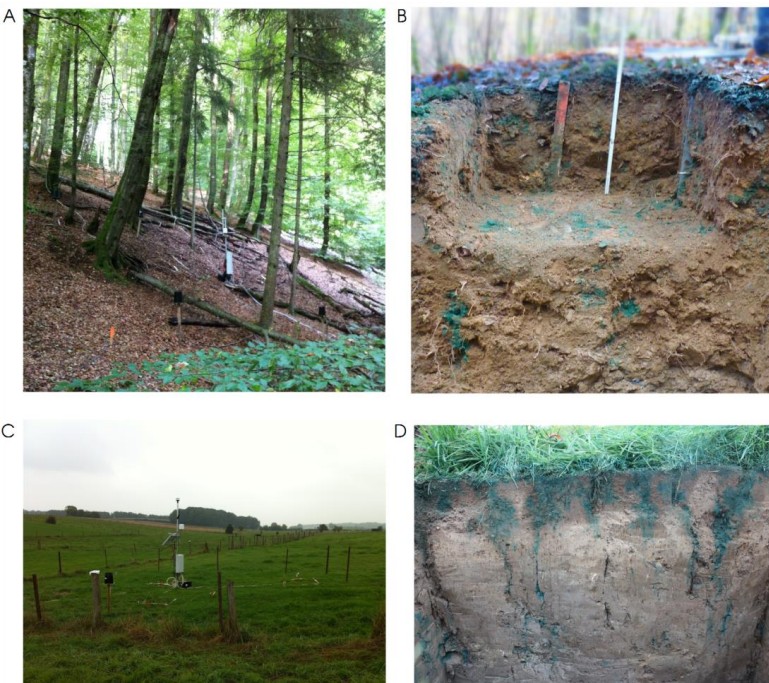


**Figure 2 (A) Typical steep forested hillslope in the Colpach catchment; (B) Soil profile in the Colpach catchment after a brilliant blue sprinkling experiment was conducted. The punctual appearance of blue color illustrates the influence of vertical structures on soil water movement in this Schist area. (C) Plain pasture site of the Wollefsbach catchment; (D) Soil profile in the Wollefsbach catchment after a brilliant blue experiment showing the influence of soil cracks and vertical structures on the soil water movement.**




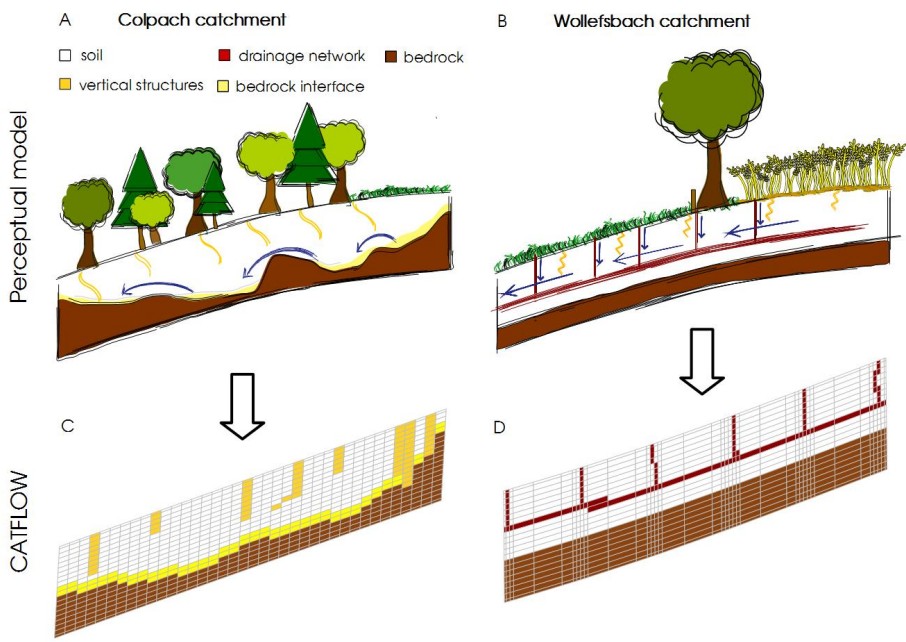


**Figure 3 Perceptual models of the (A) Colpach and (B) Wollefsbach and their translation into a representative**
**hillslope model for CATFLOW. Small sections of the CATFLOW hillslope are displayed for the (C) Colpach and**
**for the (D) Wollefsbach.**





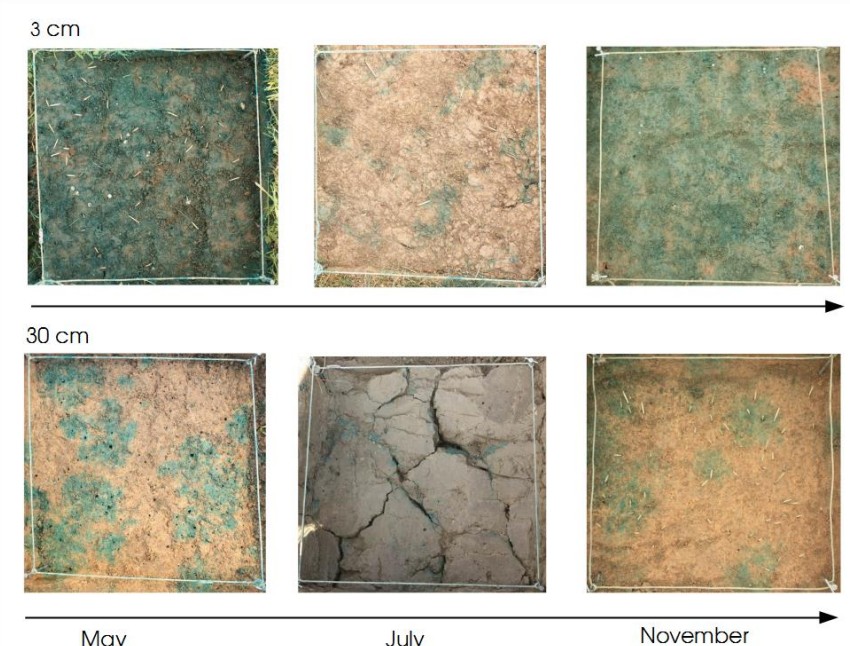


**Figure 4 Emergent structures in the Wollefsbach catchment for the sampling dates. In May macropore flow**
**through earth worm burrows dominates infiltration, while in July clearly visible soil cracks occur. In contrast, a**
**more homogenous infiltration pattern is visible in November, especially at 3 cm depth.**






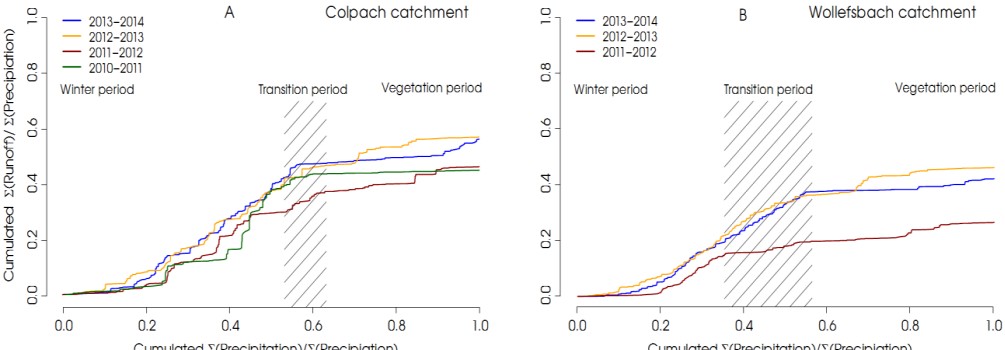


**Figure 5 Normalized double mass curves for each hydrological year from 2010 to 2014 in the Colpach catchment**
**(A) and from 2011 to 2014 in the Wollefsbach catchment (B). The transition period marks the time of the years**
**when the catchment shifts from the winter period to the vegetation period. The separation of the seasons is based**
**on a temperature index model from Menzel et al., (2003). Since the season shift varies between the hydrological**
**years the transition period is displayed as an area.**




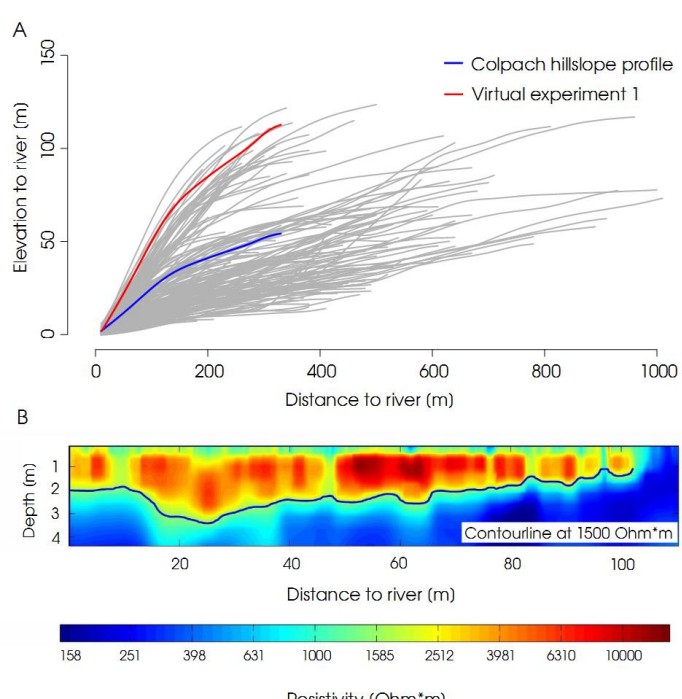


**Figure 6 (A) Profile of all hillslope extracted from a DEM in the Colpach catchment. The two hillslope profiles
we used in this study are highlighted in blue (reference model setup) and red (virtual experiment 1 (VE1) steeper
hillslope). (B) Bedrock topography of a hillslope in the Schist area measured using ERT. The contour line
displays the 1500 Ωm isoline which is interpreted as soil bedrock interface.**











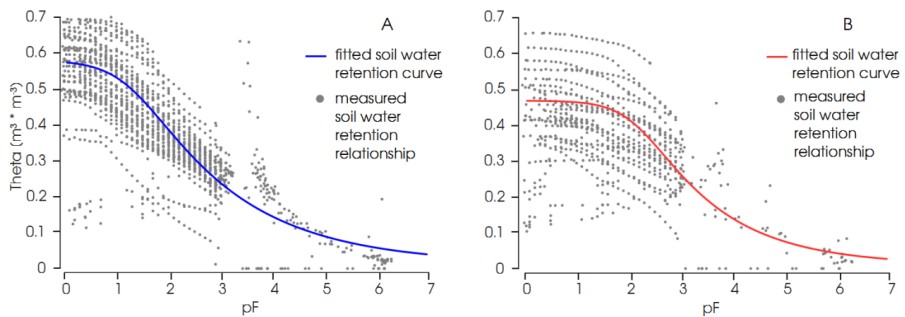


**Figure 7 Fitted soil water retention curves and measured soil water retention relationships for the Colpach (A)**
**and Wollefsbach (B) catchment.**





**Table 1 Hydraulic and transport parameter values used for different materials in the basic model setups.**

| Type of structure | Saturated hydraulic conductivity $K_s$ (m s$^{-1}$) | Total porosity $\Theta_s$ (–) | Residual water content $\Theta_r$ (–) | Reciprocal air entry value $\alpha$ (m$^{-1}$) | Shape parameter $n$ (–) |
|---|---|---|---|---|---|
| *Colpach* | | | | | |
| Soil layer | 5×10$^{-4}$ | 0.57 | 0.05 | 6.45 | 1.5 |
| Soil bedrock interface | 2×10$^{-3}$ | 0.35 | 0.05 | 7.5 | 1.5 |
| Bedrock | 1×10$^{-9}$ | 0.4 | 0.05 | 0.5 | 2 |
| | | | | | |
| *Wollefsbach* | | | | | |
| Soil layer | 2.92×10$^{-4}$ | 0.46 | 0.05 | 0.66 | 1.1 |
| Drainage system | 2×10$^{-3}$ | 0.25 | 0.1 | 7.5 | 1.5 |
| Bedrock | 5×10$^{-9}$ | 0.45 | 0.11 | 0.5 | 2 |






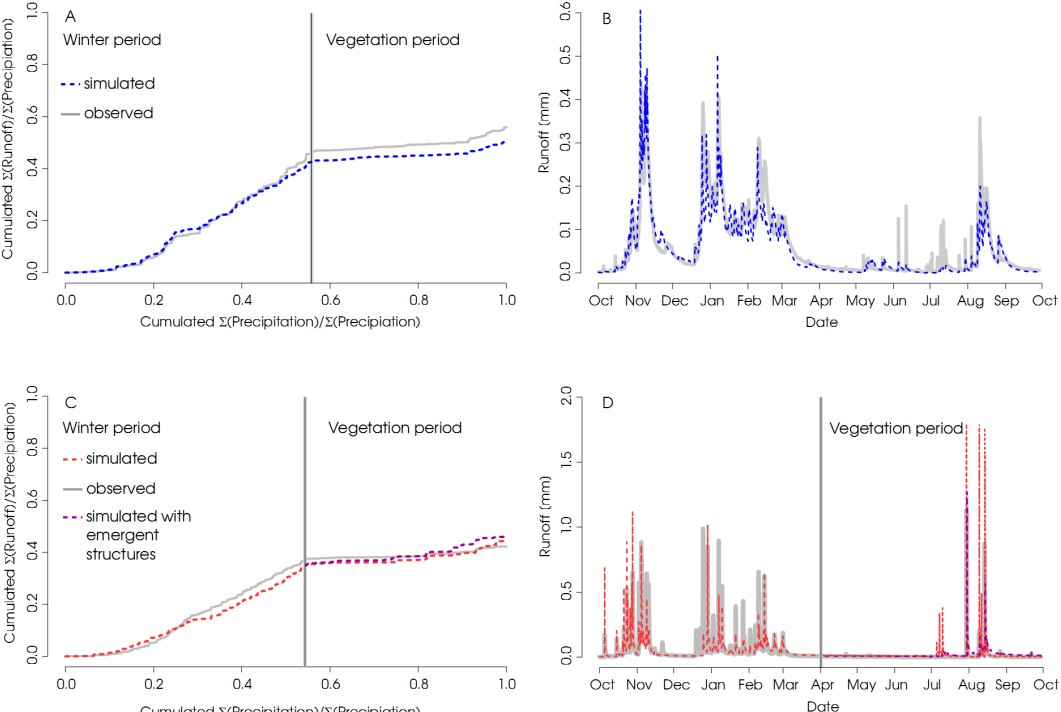


**Figure 8 Simulated and observed normalized double mass curves of the Colpach (A) and the Wollefsbach (C) catchment. The double mass curves are separated into a winter and a vegetation period after Menzel et al., (2003). The simulated runoff against the observed runoff are displayed in (B) for the Colpach and in (D) for the Wollefsbach. Moreover, in (D) the simulation period is separated into a winter and vegetation period and the model result with the emergent structures through the increased hydraulic conductivity (VE4) is displayed for the vegetation period.**






**Table 2 Benchmarks for simulated double mass curves and simulated discharge for all model setups used in this study.**

| Model setup | Double mass curve: | Discharge: | | |
|---|---|---|---|---|
| | KGE | NSE | LogNSE | KGE |
| *Colpach models* | | | | |
| **A1:** Reference Colpach model | 0.87 | 0.84 | 0.75 | 0.85 |
| *Virtual Experiments 1* | | | | |
| **VE1:** Steeper topography | 0.86 | 0.81 | 0.85 | 0.8 |
| *Virtual Experiments 2* | | | | |
| **VE2.1:** No bedrock interface | 0.82 | 0.8 | 0.72 | 0.72 |
| **VE2.2:** Parallel bedrock | 0.78 | 0.56 | 0.69 | 0.59 |
| **VE2.3:** Conceptualized storage | 0.95 | 0.88 | 0.8 | 0.91 |
| **VE2.4:** No vertical structures | | | | |
| | 0.71 | | | |
| *Virtual Experiments 3* | | | | |
| **VE3:** Improved starting point for the phenological cycle | | | | |
| | 0.95 | 0.84 | 0.8 | 0.86 |
| *Wollefsbach models* | | | | |
| **A2:** Reference Wollefsbach model | | | | |
| | 0.94 | 0.26 | 0.82 | 0.64 |
| *Virtual Experiment 4* | | | | |
| **VE4:** Emergent structures | | | | |
| | 0.95 | 0.74 | 0.61 | 0.76 |







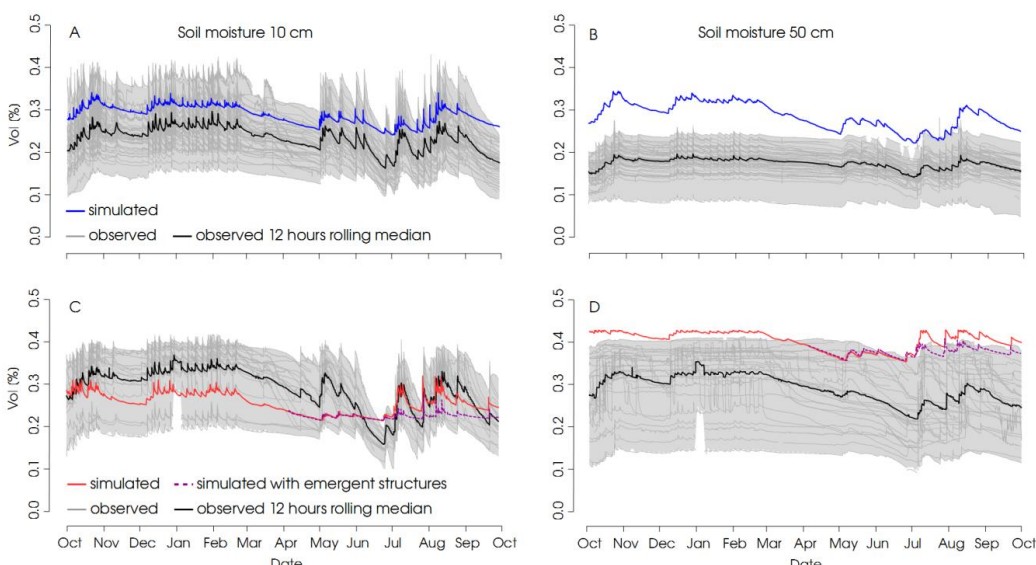


**Figure 9 Observed soil moisture at 10 and 50 cm depths in the schist (A and B) and marl (C and D) area of the**
**Attert catchment. Additionally the 12 hours rolling median (black) derived from the soil moisture observations**
**and the simulated soil moisture dynamics at the respective depths (blue Colpach; red Wollefsbach; green**
**Wollefsbach with emergent structures) are displayed.**




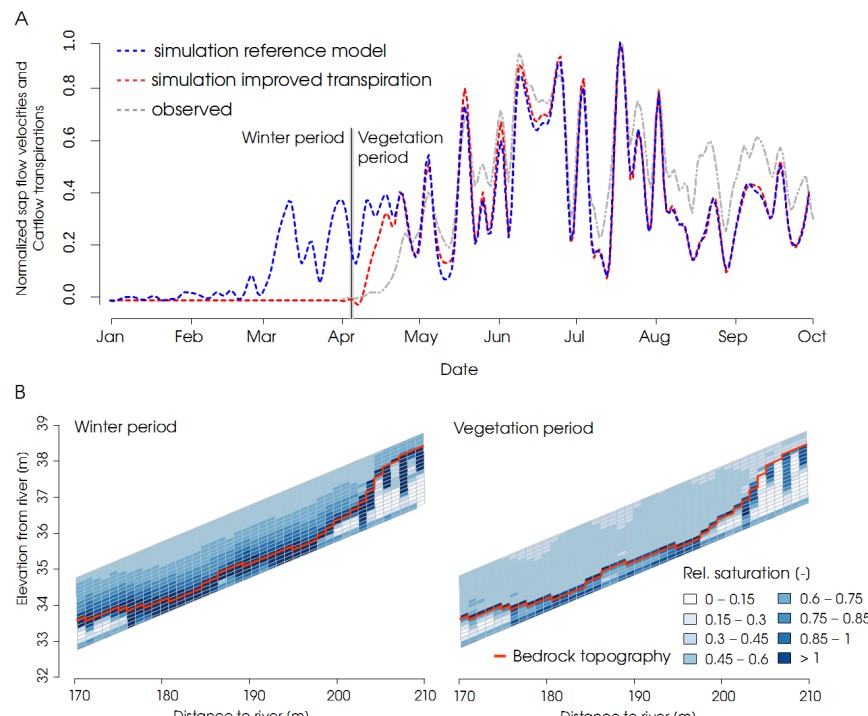

**Figure 10 (A) Normalized observed average sap velocities from 28 trees in the Colpach catchment against normalized simulated transpiration from the reference model as well as from the model with the improved transpiration routine; both were smoothed with a three day rolling mean. (B) Section of the Colpach reference model in the winter period (January) and in the vegetation period (June) with the relative saturation for every cell.**






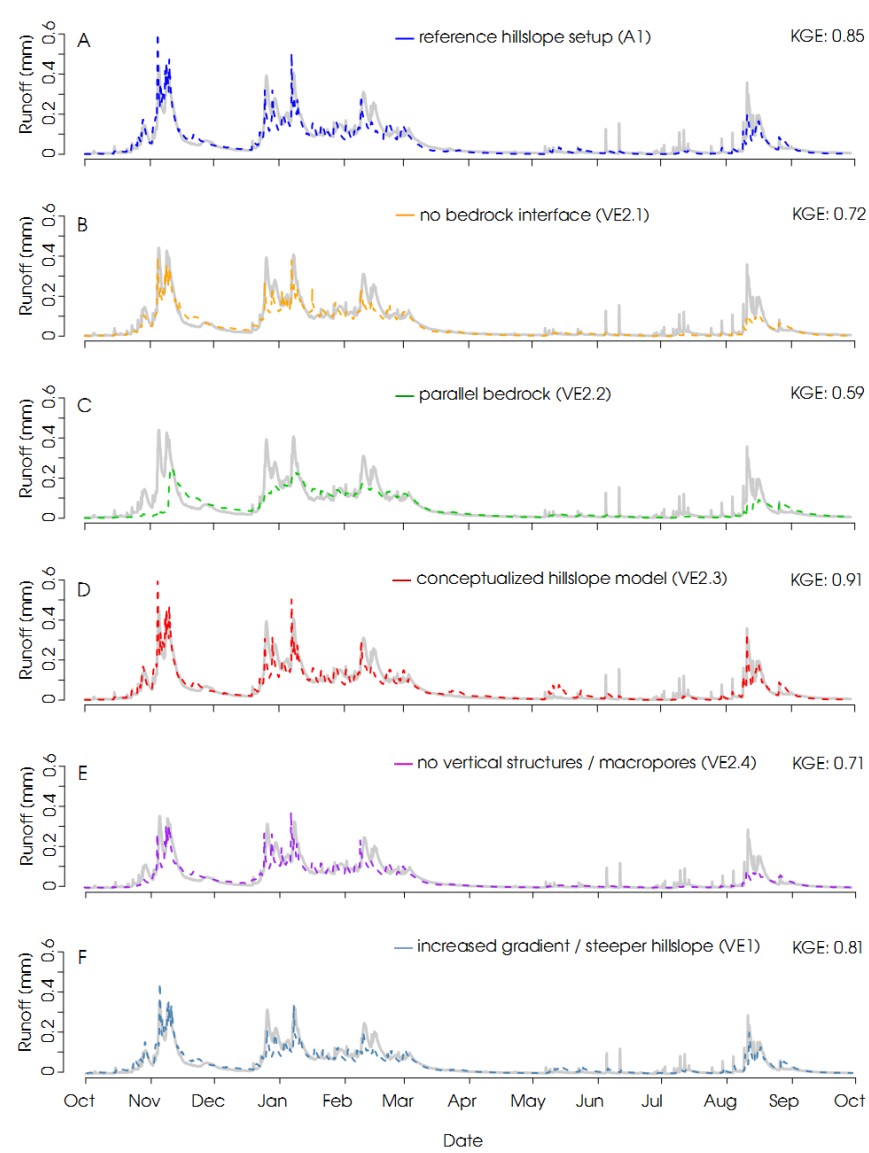


**Figure 11 Virtual experiment 2: (A) Simulated runoff from the reference model setup (A1), (B) model setup with**
**no bedrock interface (VE2.1), (C) model setup with a parallel bedrock to the surface topography (VE2.2), (D)**
**model setup with a conceptualized storage at the hill foot (VE2.3), (E) hillslope without vertical structures**
**(VE2.4), (F) increased topographic gradient through a steeper hillslope topography (VE1)**







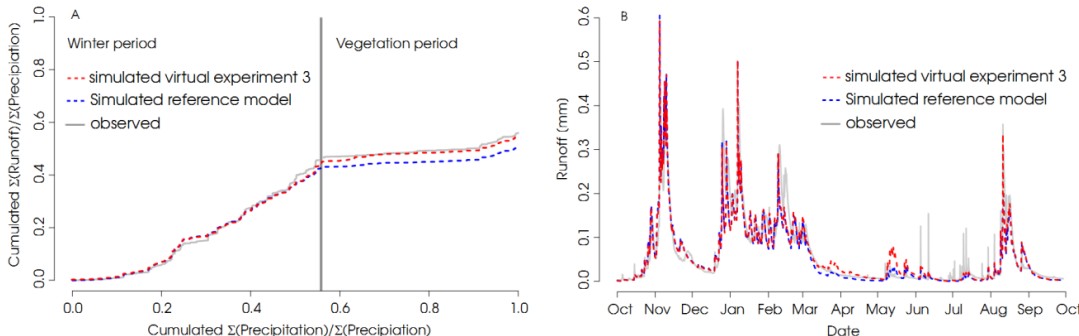


**Figure 12 Virtual experiment 3: Normalized double mass curves (A) and discharge (B) from the model with the**
**improved starting point of vegetation period (red), from the original model setup (blue) as well as the observed**
**normalized double mass curve and runoff (grey).**