# Peer review of "Picturing and modeling catchments by representative"

_Hydrology and Earth System Sciences, 2016_

## Short Comment (SC1) · 29 Jun 2016

**J Ding**

johnding\_toronto@yahoo.com

Received and published: 29 June 2016

Subject: A picture-perfect hillslope model?

**The Discussion Paper: Highlight**

I've enjoyed reading the authors' metaphysical discourse on, and in-depth analysis of, a high-resolution physics-based simulation of the headwater Colpach catchment in Luxembourg. What piqued my curiosity is their interesting but, to them, surprising finding on the utility of a conceptualized depression or storage in a hillslope.
The virtual storage is located on the foot and riparian zone of a representative hillslope, measuring 350 m in length, 54 m in height, and 42,600 m2 in area. Equivalent in volume to all the depression storages in their reference model, it is created by an artificial *vertical*-barrier on the "left" boundary (not the "right" as in Line 555). The barrier is located in the lower 70% of a 1 m-deep soil profile above the weathered Schist bedrock. This is in addition to the no-flow boundary condition on the right-side margin.

The real hillslope is one of the 241 hillslopes comprising the 19.4–km2 catchment, and representative in terms of their average length and height. The catchment, in terms of the area, almost doubles the sum of its representative hillslopes, i.e., 241 times 4.26 ha.

(See, among others, Subsection 4.4, Virtual experiments to search for first–order controls, Lines 708–717; Subsection 5.3.2, Is bedrock topography a first–order control at the Colpach? Lines 884–889; Table 2, VE2.3; and Figure 11(D).)

**Balancing Nature's energy: Picturing a Muskingum River reach**

From a parametric hydrology perspective on flood simulation, I imagine a catchment's storage *an invisible hand* in its response to rainfall–excess forcing (Ding, 2011, Section 7.5).

To keep track of filling and spilling of the lumped storage space, the authors may wish to consider an additional energy–balance equation. This is expressed in the form of a nonlinear Muskingum routing model (Ding, 2011, Equations 4 and 5):
**Discussion** paper

HESSD
$$Q = c^N S^N - c_1 \frac{dS}{dt}, \quad N > 0, \ c > 0, \ c_1 \ge 0,$$
(1)

$$S = (1/c)[c_1I + (1 - c_1)Q]^{1/N}.$$
(2)

Note in Equation (1), the discharge Q is a function of the storage S, and, more importantly, its time-derivative dS/dt. The latter term simulates the often observed S - Q loop, a manifestation of the system hysteresis.

For a hillslope *recession* hydrograph, the degree of nonlinearity N is 2, meaning a quadratic storage (Ding, 2013, Equation 4; 2015). What remains are a scale parameter c and a weight factor  $c_1$ . (The weight  $c_1$  varies between zero and one, likely having a mean value of 1/2.) Having thus a maximum of two parameters, and a minimum of one, this storage (component) model is as concise as a kernel of truths should be.

**A newly emerging picture of the catchment**

The authors' finding on the utility of a conceptualized storage in a representative Colpach hillslope opens up a new vista of the headwater landscape. The expansive painting will be dotted by Muskingum–type objects in the river network itself as well as the hillslopes upstream.

An enhanceable lumped–storage element modelled after the Muskingum could be added, in parallel to or in series of, a CATFLOW–type distributed hillslope model. This will expand a lone hillslope model to truly a catchment one.

The authors, in their concluding paragraph, quote approvingly a maxim, attributed to Antoine de Saint–Exupéry, of drawing *a perfect picture* of the world. I characterize as *hysteresistical* his drawing manner of adding and subtracting things in a proverbial endless circle.

In his eye and his only, will *adding* a number of *lumped-storage* objects to a *distributed* hillslope model make it, the hillslope and catchment model, not picture-perfect? I hope not.

**Afterword**

This Short Comment is a latest of many sketches of many a catchment landscape.

**References**

Ding, J. Y.: A measure of watershed nonlinearity: interpreting a variable instantaneous unit hydrograph model on two vastly different sized watersheds, Hydrol. Earth Syst. Sci., 15, 405-423, doi:10.5194/hess-15-405-2011, 2011.

Ding, J. Y.: Technical Note: A measure of watershed nonlinearity II: re-introducing an IFP inverse fractional power transform for streamflow recession analysis, Hydrol. Earth Syst. Sci. Discuss., 10, 15659-15680, doi:10.5194/hessd-10-15659-2013, 2013.

Ding, J. Y.: Applying an exact solution of the Brutsaert and Nieber baseflow model for watershed yield prediction: A case study for the Spoon River at Seville, Illinois, USA, Journal of Water Resource and Hydraulic Engineering, 4, 160–161, 2015.
**DOI:10.5963/JWRHE0402005**

(This is an extended abstract of a poster (paper no. WRE1305) presented at 2015 International Conference on Water Resource and Environment (WRE2015), Beijing, China, July 25–28.)

---

## Author Comment (AC1) · 4 Jul 2016

First of all, thank you, Dr. Ding, for contributing your thoughts and comments to the discussion of our manuscript. In this reply we take the chance to highlight the main avenues of our manuscript once more:

In order to obtain a catchment scale model, you propose to add a Muskingum type lumped storage element to our hillslope scale distributed "CATFLOW" model (Zehe et al. 2001). Though we acknowledge that this might be an interesting approach for some studies /areas, in this case we explicitly chose to focus on the representation of a headwater catchment by a single hillslope for various reasons. As we point out in the manuscript our research catchment is most likely hillslope-dominated and hence not dominated by the river network (Kirkby, 1976; Robinson et al., 1995). Furthermore,

topography is not a first order control in the area as shown by Fenicia et al. (2016) and Jackisch (2015). With this in mind and combined with the fact that the soil and geological maps which are available for this lower-mesoscale catchment are almost uniform, we feel it is difficult to justify a catchment scale distributed model.

We believe, it does not make much sense to calculate the same hillslope 241 times to then connect the results with a flow routing model, if river flow routing is not a dominant process at the scale of interest. In our opinion, a representative hillslope model is a better starting point for the headwater catchment we investigated, and makes the identification of dominant processes and structures more straightforward. Additionally, as we state in our discussion paper, we are not looking from a flood simulation perspective (as you state in your comment) on hydrological modeling. We think that physically-based models as diverse and parameter-rich as "CATFLOW" are not the right tool if the goal is the most accurate rainfall-runoff simulation.

We understand the wish for using a hydrological model in a distributed form where every model element is connected to a real-world object. But in some cases this might be a rather difficult task in the context of the availability and the information content of spatially distributed data. We agree with you that a possible next step could be a distributed modelling approach; however, we would follow this path not because we are convinced to achieve a perfect rainfall-runoff simulation, but to learn more about, for example, how distributed precipitation fields trigger different (or similar) responses in landscape elements, temporally and spatially.

You are right; Line 555 should be the left and not the right boundary.

"The catchment, in terms of the area, almost doubles the sum of its representative hillslopes, i.e., 241 times 4.26 ha" - The hillslopes have different sizes and are not all 4.26 ha. Our calculations are based on a single hillslope which was picked from the distribution of all hillslopes, according to its average distance and elevation to the creek.
**Discussion** paper

cussion paper.

Ding again for his comments and recommendations on our dis-

References:

We thank Dr.

Fenicia, F., Kavetski, D., Savenije, H.H.G., Pfister, L., 2016. From spatially variable streamflow to distributed hydrological models: Analysis of key modeling decisions. Water Resour. Res. 1–36. doi:10.1002/2015WR017398
Jackisch, C., 2015. Linking structure and functioning of hydrological systems. KIT - Karlsruher Institut of Technology. doi:10.5445/IR/1000051494
Kirkby, M., 1976. Tests of the random network model, and its application to basin hydrology. Earth Surf. Process. 1, 197–212. doi:10.1002/esp.3290010302
Robinson, J.S., Sivapalan, M., Snell, J.D., 1995. On the relative roles of hillslope processes, channel routing, and network geomorphology in the hydrologic response of natural catchments. Water Resour. Res. 31, 3089–3101. doi:10.1029/95WR01948
Zehe, E., Maurer, T., Ihringer, J., Plate, E., 2001. Modeling water flow and mass transport in a loess catchment. Phys. Chem. Earth, Part B Hydrol. Ocean. Atmos. 26, 487–507. doi:10.1016/S1464-1909(01)00041-7

---

## Referee Comment (RC1) · Anonymous Referee #1 · 19 Jul 2016

Summary and Recommendation:

In this paper the authors address two basic questions:

1) If you have a lot of spatially-distributed information about the geology and soilhydraulic properties in a catchment, can you parameterize a high-dimensional, spatially-distributed model (without any calibration or inverse optimization) to accurately represent water flow within a single 2-d hillslope, based on that existing knowledge?

2) If your knowledge-based (not optimized) model domain and parameterization prove reasonably representative, can you then extrapolate this representative 2-d hillslope across the 3-d volume of the entire catchment, to simulate hydrograph dynamics and the annual water balance for the entire catchment?

To address these questions the authors employ a Richards-equation-based model with evapotranspiration module and overland flow routing modules. They apply the model to simulate hillslope-scale soil moisture dynamics and water-balance partitioning—and after extrapolation, whole-catchment streamflow dynamics—from two catchments in Luxembourg with varying geology, topography, soil, and vegetation. In addition to the modeling, their analysis includes extensive, and impressive data sets representing spatially distributed soil-hydraulic properties, geologic features, plant transpiration, and topography. These questions, the observations, and the methodological approach adopted here are of interest in scientific hydrology and would be received with interest by readers of HESS. The writing is mostly clear in a grammatical sense, though somewhat desultory and technically ambiguous in many areas. The model setup is technically sophisticated in many ways, though not so in others. The graphics are of very high quality. The organization of the paper is logical, though as I suggest below, there are significant portions that might be omitted to maintain a consistent focus of the paper throughout. Overall there were too many competing objectives in the paper. As such, any salient result or conclusion is hard to discern. The results and discussion could be greatly improved.

I recommend that this paper could be accepted for publication in HESS, but I believe some major revisions are needed beforehand, possibly including revised simulations and results. Those suggested revisions are outlined in the following General Comments section, with references to more specific Technical Comments.

General Comments:

The manuscript can be improved upon significantly by reducing total length, omitting or clarifying the use of excessive jargon, and possibly removing sections of the paper to minimize superfluous and unfocused commentary. Some specific instances are noted in Technical Comments 5-10, 24, 27, and others.

The manuscript may be much improved by omitting much of the commentary about

**HESSD**
modeling evapotranspiration, along with the virtual experiment 3 (VE3) and associated results, and rather focusing on the importance of explicitly representing (or not) landscape heterogeneity for the purpose of simulating hydrographs. The authors talk quite a lot about all the uncertainties associated with ET modeling in hillslope/catchment hydrology, but their modeling approach does not reflect the state of the science (e.g. as presented in disciplines such as hydrometeorology and plant biophysics), so this discussion does not seem warranted. See Technical Comments 12, 29, and 56.

The methodological approach is inadequately described in many instances, with some revision being needed. See Technical Comments 16-26, 36, 39, and others.

There are some aspects of the model domain and parameters that seem very unrealistic. For example, the bedrock for both modeled catchments is parameterized to have porosity of 40-45% (Table 1). That's comparable to, or greater than, the porosity of many soils. I can't imagine how Schist—a metamorphic crystalline rock—can be composed of 40% air space. This is certainly not consistent with most reported porosities for Schist, which are typically 10% or less. This will have a large impact on the flow simulations, since about half of the hillslope domain is bedrock. If there is some justification for this, and some other aspects of the model, then perhaps the only revision that is needed is to provide that justification. Otherwise, many of the simulations may need to be repeated with more appropriate parameterization. See Technical Comments 19-26, and others.

I strongly suggest that the authors consider refocusing this paper on two subject areas. First, concentrate on the modeling of spatially-distributed soil moisture dynamics and the temporal dynamics in the hydrograph. You use a spatially distributed model but nowhere do you assess the models ability to accurately represent any spatial pattern. You have an enormous amount of interesting spatially distributed data, so this could become one of the most rigorous tests of the Richards equation model at the hillslope scale ever published. See Technical Comments 33-34, 47-48, and 54. I recommend the second focal point to be the argument that extrapolating the parameterization of
a single 2-d hillslope to an entire 3-d catchment may, or may not, be defensible. At present this is stated as an objective, but the results and discussion are ambiguous, or possibly in disagreement, about this point. Showing these outcomes would be challenging enough, and a good contribution to contemporary scientific hydrology. Toward that aim, I suggest omitting the virtual experiments and associated discussion. I found the virtual experiments and the associated discussion around them to be desultory and vastly oversimplified. It was not clear to me how they relate to any previously stated objective. Those virtual experiments mainly consisted of changing a single variable (e.g. total relief, timing of bud break for plants, or hydraulic conductivity), and speculating broadly about the implications of the resulting simulations. See Technical Comments 50, 52-53, and 55-56.

Last, the authors seem to be advocating that high-dimensional, spatially-distributed models will always be wrong for a variety of reasons, but that they're still important for helping us learn about which state-variables and flow processes dominantly control the emergent streamflow dynamics at the catchment outlet. I think this is a relevant and worthwhile argument, but I encourage the authors to better focus their writing on this topic. This effort may be aided by omitting some other sections of the paper, as noted above. At present the authors present some seemingly conflicting (at least to me) statements about what exactly is the merit of taking this approach, and just how well it did or did not work out for them. See Technical Comments 34, 46, 47-48, and 54.

Technical Comments/Corrections:

1) Line 54: Change "reflect" to "reflects".

2) Line 57: I suggest using "e.g." throughout the manuscript when the cited work is only one example of all the works that could be cited to support a particular statement, as you did here for the Brontstert and Plate reference, but not the others. Certainly there must be innumerable works on hillslope energy and sediment fluxes.
3) Line 58: I would suggest using either "perceptual" or "perceptional" as an adjective to describe "model", but not both.

4) Lines 58-61: Citations not needed for this subjective statement.

5) Line 67: Not immediately clear to the reader what distinguishes a "conceptual model" from a "perceptual model".

6) Lines 85-88: Consider rephrasing or deleting. Not clear what is the message of the sentence. The result of any mathematical model must be compared to observations in the modeled system. This goes without saying, and certainly doesn't require a supporting citation. Perhaps I am just not clear what you mean by "benchmarking".

7) Lines 49-97: This is very clearly written, but for sake of making your paper as concise as possible—and therefore more likely to be read in full—you might consider abbreviating the section, or deleting. It doesn't assert much, or highlight some problem with the status quo in catchment modeling. It mainly states that model-based analysis are useful for learning about hydrological systems, which I think is already acknowledged ubiquitously in the community of hydrological scientists. It's your call, but there is no shortage of papers on hillslope modeling, and I always prefer to read a more concise one than a longer one.

8) Line 101: Use of "behavioral" is ambiguous, maybe omit or clarify.

9) Line 113: "functional behavior of catchments of organized complexity" is hard to interpret. Caution in using too much ambiguous jargon.

10) Line 128-130: Consider rephrasing using the plainest language possible, since this is seemingly an important part of your rationale statement for the study.

11) Line 137-140: Rephrase "behavioral physically model structures". Also, comparing model outputs to observed data sets (like tracer time series) doesn't inherently reduce the number of degrees of freedom in the modeling procedure. It might help constrain parameter values. If that observed data set somehow informs the modeler that a par-
ticular parameter is unnecessary, or that a spatially-distributed domain can be adequately represented in a lumped way, then the degrees of freedom might be reduced. Are the works you cite here examples of the latter? For example, in the application of Richards equation with the van Genuchten-Mualem soil-hydraulic model, you can't just decide based on some observation that you no longer need the shape parameter in the hydraulic model – it still has to be there. If the observational data set leads you to a coarser grid resolution for the domain, then that would be a reduction in degrees of freedom, since the number of spatially distributed elements where the equation is solved/averaged is reduced. But really, for multi-parameter models that are spatially distributed, the degrees of freedom are always grossly high, not even considering the fact that we most often ignore anisotropy and hysteresis in soil hydraulics in hillslopeto catchment-scale applications. That's the whole motivation for lumped models, right?

12) Lines 166-184: Evapotranspiration is represented in a rudimentary way in many hydrological models precisely because those models have the primary aim of predicting hydrographs. For modelers with this primary interest, there will inevitably be a greater effort spent on representing such processes as non-equilibrium flow than on ET. because the former is of more interest. Models aimed at predicting streamflow use long-standing, and possibly antiquated ET models, because they're convenient, not because enhanced knowledge of stomatal dynamics and plant phenology is absent. Tree physiologists and hydrometeorologists have highly advanced understanding of these phenomena, and their discipline-specific models reflect that. These arguments are worth keeping in mind when you're noting the "uncertainty in the community on how to represent plant physiological controls on transpiration in hydrological and land surface models." Is it uncertainty, or just lack of interest/effort to study and implement models that reflect contemporary knowledge in plant physiology and boundary-layer biophysics? As an example, you go to great lengths here to incorporate small-scale non-uniformities into the subsurface flow domain, but you use a pretty standard version of the PM approach for ET that's been around for over 30 years now. You assume homogenous land cover in Colpach catchment, and you use vegetation parameters
from a non-local catchment in Germany, with phenology assumed invariant from year to year (I assume that's what you mean by "fixed"). With that model setup, it's really not appropriate for you to be talking about how uncertain the community is in how to represent the complexity of these processes in models. It wouldn't be very complex for you to considerably improve this standard version of the PM model just by using site-specific data, a dynamic representation of phenology, and to accurately reflect the relative abundance of forest versus other vegetation cover in the catchment. I don't want to be overly critical, because some assumptions and simplifications are always made in modelling. My main point is that you don't want to go on philosophizing about how we learn from still-uncertain models, if the model you're employing is not nearly state-of-the-art, or not parameterized nearly as carefully as it could be.

13) Line 198-201: Or stated more directly, "To assess the ability of a spatiallydistributed, physically-based model to accurately simulate multiple state and flux variables (not just streamflow), when the parameterization of the model is based on observed catchment properties, not on an optimization algorithm."

14) Line 240-242: This concept certainly precedes the work of Zehe 2014. For example, consider these important papers, which are notably absent from works cited in your introduction:

[Berne et al., 2005; Harman and Sivapalan, 2009]

Berne, A., R. Uijlenhoet, and P. A. Troch (2005), Similarity analysis of subsurface flow response of hillslopes with complex geometry, Water Resources Research, 41(9) Harman, C., and M. Sivapalan (2009), A similarity framework to assess controls on shallow subsurface flow dynamics in hillslopes, Water Resources Research, 45 15) Lines 323-324: Please provide some explanation of what you mean by the onset of vegetation period? Presumably that would be timing of leaf development for crops and deciduous plants, but evergreen plants will be physiologically active even through winter and spring, albeit at lower rates than in summer. Then there is of course an extended pe-
riod when crops and deciduous plants transition from no foliage to the maximum leaf area they will obtain that year. Than transition can span weeks to more than a month. 16) Lines 355-368, and Figure 7: The variability among measured moisture retention curves for your soil samples is remarkable. Can they be logically grouped in any way, for example, by landscape position or soil depth? If so, a color scheme to illustrate that would be very interesting. Some additional detail about where, and at what depths, the soil samples were taken is needed.

17) Lines 385-388: Are the sapflow sensors collocated with rainfall and soil moisture measurements? Please elaborate on the exact type of sensors, depth of installation into trees, and other important details about their operation.

18) Lines 424-426: Any consideration of the soil-moisture dependence of vapor diffusivity in soil? It varies as a power-law function of air-filled porosity, over about 4 orders of magnitude.

19) Lines 432-446: How are the probabilities of the Poisson process determined? Is this based on some knowledge that informs your perceptual model, or are though chosen arbitrarily? Or, do you determine some best-performing parameters based on a sensitivity/optimization process? Please describe in a little more detail, and consider providing an image of what these structures look like in your final model domain. Is this what we're seeing in Figure 3C,D? If so, please just allude to that figure here in the text.

20) Lines 453-454: Considering the horizontal resolution of model elements is 1-m, I'm wondering how realistically the vertically-oriented, preferential-flow zones can be represented? Certainly the macropores in your photographs are not 1-m wide. Representing their tortuosity by vertically-offset grid cells of 1-m width seems like a gross distortion as well. Can you discuss how you rationalize this model domain and horizontal grid resolution, especially with regard to those grid cells that are imposed to represent preferential flow structures?

HESSD
Also, can you provide detail about the dye-irrigation studies from which the photograph were derived? The pictures are very insightful. However, it is well known that irrigation studies often impose exceptionally high input fluxes, and with sprinkler systems that exhibit enormous spatial variability. Can you comment specifically on the irrigation rates and the spatial uniformity of the irrigation system, and how those irrigation rates compare to the frequency distribution of rainfall intensities that occur at these field sites?

21) Lines 465-466: I'm personally not familiar with the phrasing "free outflow boundary" and "gravitational flow boundary". Please clarify exactly what these mean, and maybe represent in an equation. Does "free outflow boundary" imply a seepage-face, where there is no flow until water-pressure head exceeds atmospheric pressure? And does gravitational flow boundary imply a zero gradient in soil-water pressure head (i.e. flow is governed by the elevation gradient and saturated hydraulic conductivity)?

22) Line 477: 40 - 45% porosity seems exceptionally high for a fractured bedrock. Any evidence to support that number? That's equally porous as most soils. It's hard for me to visualize how a cubic meter of schist could be 40% air space. Porosity for metamorphic crystalline rocks is typically reported to be less than 10%. I think this parameterization is patently wrong, and will have a significant effect on your simulation results since a majority of the domain is defined as bedrock.

23) Line 479: Here again, it would be good to know about the depth distribution of the soil samples collected for hydraulic characterization. Did any actually come from that depth?

24) Lines 486-490: This sounds wonderfully sophisticated. I have no idea what it means. I'm probably not alone in that regard. It seems like important information about how spatial heterogeneity of hydraulic properties are generated in the model domain, so maybe a sentence or two—in plain language—to build the intuition of the reader/s that are not intimately familiar with this jargon.
25) Line 528: Do you mean left boundary? The right boundary is no flow.

26) Line 527-530: You're simulating flow through a single 2-dimensional hillslope profile, so how can you compare the composite discharge (overland flow, subsurface flow, and deep drainage) to the measured streamflow for the whole 3-d catchment? Are you integrating the hillslope response over the entire 3rd dimension of the catchment? Please explain this. Also, do you think it is appropriate to include the deep drainage flux in this composite outflow when comparing to the stream hydrograph? It could conceivably travel through an aquifer system that discharges outside the boundaries of your catchment, no?

27) Lines 531-533: What exactly do you mean by "validated"? What's the difference between validating and benchmarking? The phrase "tuning against" is not readily interpretable.

28) Line 555: Left boundary?

29) Line 566-568: The important trend for the catchment water balance is the timing of the leaf area expansion, maximum, and decline (in fall). The leaf area is the dominant control on transpiration and net radiation. So, I don't understand how or why you change the timing of phenology without changing the temporal dynamics of leaf area. Please explain the rationale for this.

30) Line 594: Figure 9B, rather than 10B?

31) Lines 614-615: Are you talking about Weierbach catchment in Germany? It's irrelevant. I suggest you delete that and stay focused on your catchments.

32) Line 619-622: Run-on sentence that is very hard to interpret. Please rephrase. Also, please explain what inference you think is made possible by comparing NSE with log(NSE).

33) Lines 622-627: These statements are questionable. Are you claiming that infiltration-excess overland flow is occurring, or overland flow due to saturation ex-
cess? You say that the model erroneously generates overland flow in the summer in Wollefsbach due to convective storms. The saturated-hydraulic-conductivity parameter you report in Table 1 is 2.9 x 10-4 m/s, or about 1.8 cm/minute. This is quite a high hydraulic conductivityTwhat one might expect for coarse sand. Are your surficial soils sandy? Are those convective storms producing rainfall flux greater than 1.8 cm/minute? Seems doubtful storms like that would occur frequently. How is the model generating so much overland flow if the rainfall rates are (I assume) always considerably lower than the saturated conductivity? In Figure 9D it looks like your simulated-average-soil moisture is in excess of almost all the measured time series. Maybe you're way overestimating soil-water storage and generating saturation-excess overland flow, rather than infiltration excess overland flow. If that's the case, and if it's saturation-excess overland flow, then you can't immediately assume that enhancing Ksat to represent soil cracks is the next, necessary step to improving the model. You need to get the soil moisture dynamics correct before you can go off exploring that speculation. You're Ksat value is already pretty high, is it based on measurements?

Also, it's somewhat a shame that you have all those soil moisture observations, and a spatially-distributed model, but you only compare the mean-simulated soil moisture (I assume it's the mean) to the observations. The spatially-distributed model gives you lots of spatially-distributed results to compare to spatially-distributed observations. If you're just going to look at the average-simulated soil moisture, you're giving up all that detail that is provided by the model, and one has to ask why not just use a lumped model? You should compare simulated soil moisture at specific points in the landscape to the observed soil moisture at those same points.

34) Lines 631-635: I am quite confused by this statement. You are using a spatiallydistributed model, so why are you claiming that it's unrealistic to expect the model to accurately represent the spatially-distributed nature of soil-moisture dynamics? It should be able to represent at least coarsely the spatial distribution of soil moisture, for example, differences in upslope versus downslope positions, or differences in ar-
eas overlying saturated bedrock depressions versus those areas where the bedrock roughly parallels the land surface. Again, if you don't expect your spatially-distributed model to accurately represent any of these spatial patterns (which are important for runoff), then why are you using a spatially-distributed model to begin with?

35) Lines 641-642: This statement is inevitably true for every catchment in the world, and hence does not rely on any measured or simulated soil moisture dynamics. Maybe just delete.

36) Lines 671-684, and Figure 10A: This material needs much improvement. First, measuring sapflow in trees is a delicate business, with major discrepancies existing between methodologies, and significant errors arising from inexact application of methods (e.g. due to radially-varying flux rates within the sapwood, a well-documented phenomenon in the tree physiology literature). There are several reviews of this topic in the plant physiology literature (e.g. Steppe et al. 2010, Agricultural and Forest Meteorology, 150). You have provided essentially no detail about the nature of your field-based sapflow measurements. Also, how are you normalizing the measurements?

Second, you use a version of the PM model to simulate water vapor flux from the plant canopy, not sapflow (L3 T-1). The two cannot be assumed to be equal. If you consider the tree as a system spanning the point of your sapflux measurement (breast height on the stem) to the canopy, then the sapflow (input to the system) only equals the volumetric flow out of the leaves (outflow from the system) if the system is in steady state (i.e. inflow = outflow and storage is constant). Water storage in the tree stems and canopy foliage is dynamic. I am not sure how good or bad is the assumption of steady state in your system, but it is certainly an assumption you should carefully consider and provide some justification for why this comparison (between measured sapflow and modeled canopy vapor flux) is valid. Without that, you should probably omit this text and figure 10A from the manuscript.

37) Lines 691-692: You might be careful in projecting your own expectations, surprises,

**HESSD**
and uncertainties onto your readers. The result you describe here is not counterintuitive to me; it's exactly what I would expect, for the exactly the reason you state. Higher gradient = more rapid drainage = less persistent storage (assuming all else is equal, which is what you've assumed for this virtual experiment).

38) Line 694: "is" rather than "and"

39) Table 2: Please provide some rationale for why you use multiple error metrics (e.g. NSE, KGE, logNSE) instead of just one. It's just confusing to the reader when you talk about quality of results in one case using KGE, and in another case using NSE. Also, the different metrics show different sensitivities to the domain-changes utilized in the virtual experiments. Why? Which one is most appropriate in light of those differences? You're using these various metrics to make inference about the relative importance of different model features, so you need to argue why one or the other metric is better. Or just use one metric for clarity.

40) Line 739: Sentence is unclear, please rephrase.

41) Figure 8D: Here, and in some other cases, the reader cannot see much of the observed and simulated dynamics because of the relatively long time scale of the x-axis, and the flash nature of the catchment. Consider using an axis break on the x-axis, or some other mechanism to expand the time series, so the reader can clearly see the dynamics. In Figure 8D, it's impossible for me to see what differences might exist between observations, and the standard and emergent-structure scenarios.

42) Line 756: Rephrase and omit use of "one to one". It implies exactness in the representation of scale, which is not the case. For example, your model grid cells are much larger than the preferential flow structures they are modified to represent.

43) Line 766: Again, careful with projecting your own reactions and perceptions onto your reader. The use of the word "astonishing" here seems hyperbolic to me.

44) Lines 780-781: Why is that remarkable? It's a predominantly upland catchment
with forested hillslopes. Was it your initial expectation (null hypothesis) that the model would be incapable of simulating streamflow?

45) Figure 1: Please add a color scale so we can see what are the associated elevations.

46) Lines 796-801: These statements are not very clear. You state, "We also found that benchmarking of the model against sapflow data provided additional information about the representation of vegetation controls, which cannot be extracted from the double mass curve or discharge data." Exactly what information are you talking about? Your Figure 10A basically just shows that your modeled water vapor flux from the canopy (based on PM model) trends in a similar way as field-based measurements of sapflow within tree stems. That's certainly to be expected, since both are driven by net radiation, but as noted in comment 36 above, it is not necessarily meaningful to compare magnitudes of those two fluxes, because they're not the same thing. So, the sapflow data don't tell you definitively that your simulations are right or wrong. Also, you say that this additional information could not be extracted from the double mass curves. Again, exactly what information are you talking about? Certainly the double mass curves in Figure 12A show a clear effect of changing the timing of bud break. The statement in lines 799-801 is presented as a conclusion, but it's not really a conclusion is it? You knew from the beginning that using a spatially-distributed, highly parameterized model would offer the opportunity to incorporate knowledge from soil-hydraulic measurements and geotechnical surveysTknowledge that would not necessarily be incorporated explicitly in a lumped model?

47) Lines 802-804: Are you so sure this can be concluded? It seems to me that if you want to advocate the use of highly parameterized, spatially-distributed models for the sake of learning about catchments, you need to illustrate that the model is accurately representing some of the spatial dynamics in the hillslope (or catchment that is a composite of your hillslopes). In those cases the matching between simulated and observed averages is not that great for soil moisture—there are systematic errors
in all cases (Figure 9A-D). By comparing average-simulated soil moisture for the whole hillslope to the average-observed soil moisture for the whole catchment, you're failing to rigorously test the spatially-explicit predictions made possible by the model. You should try to show that the model actually properly represents spatial variability in soil moisture, saturated-zone expansion, hydraulic gradients, etc. If not, then it's hard to argue that the distributed model teaches us anything more than we would learn from a lumped model.

48) Lines 812-816: I fully agree with this statement. You don't need a high-dimensional, spatially-distributed model if all you want to do is predict runoff at an annual timescale, or even at shorter time scales. Use a transfer function, maybe even a time variable transfer function—you will still have vastly fewer degrees of freedom than in the spatially-distributed Richards equation model. But doesn't this statement contradict the overall message of your paper, that those more complex models are needed for learning about catchment functioning? The spatial variability of soil-hydraulic properties may be quite important, in fact, for properly simulating all those runoff peaks in the summer, where your model does quite poorly (Figure 8B,D, and Figure 12B).

49) Lines 823-825: Unclear, please rephrase.

50) Line 823-844: I would suggest you delete all of this. It seems wildly speculative and I have no idea how, based on the analyses performed in this paper, you conclude that, "equifinality and the concept of a representative hillslope is rather more a blessing than a curse since there is an infinite number of possible macropore setups which yield the same runoff characteristics. If this were not the case, we could not transfer macropore setups from the literature across system borders and successfully simulate two distinct runoff regimes which are strongly influenced by preferential flow." An infinite number of macropore setups that yield the same runoff characteristics AĂŤwhat are you talking about here? You tested 2 such scenarios (Figure 3C,D). Your model does a fairly poor job at matching runoff peaks at many times of year. Those peaks are the hydrological attribute most likely to be influenced by the activation or latency of preferential flow.

**HESSD**
paths. When you say you that you "successfully simulate two runoff regimes" I presume you are talking about the double-mass curves, because your simulated hydrographs show significant errors at many times of the year.

51) Line 864-866: Do you mean, "below which", instead of "above which"? I wouldn't spend much time on that. By changing slope and nothing else, you're vastly oversimplifying how soils, geology, and geomorphology affect streamflow, and how all those variables are related to topography in naturally evolving landscapes.

52) Lines 888-889: How do you justify this statement? Did all of your virtual experiments where you manipulate the bedrock topography have an equal volume of depressions? And if so, how do you go about quantifying the volume of depressions in an undulating rock surface? What constitutes a depression or a high point, versus a portion of the rock surface that is part of a datum plane?

53) Lines 867 and 895: You're using the questions as section headers, but the subsequent content does not answer the questions. First of all, define what you mean by "first order control". Do you just mean that the response variable (annual runoff ratio, or other?) is a linear function of the independent variable (bedrock topography, or vegetation)? If so, do your data corroborate such a linear relationship? I'm not sure you can say, based on the limited scenarios of bedrock topography you tried. You would have to come up with some quantitative metric distinguishing one bedrock scenario from another. In terms of vegetation, all you did was try 2 different times for bud break. Can you discern a linear relationship between "vegetation" and some response variable based on these tests?

54) Lines 912-923: Of course you can! You can setup heterogeneous rainfall inputs at the soil surface in your model domain. You can setup different scenarios of incoming solar radiation along the hillslope domain to emulate aspect-related differences in the radiation balance, water budget, and possibly soil hydraulic/or geological characteristics. I'm really struggling to understand how you continually advocate for spatially
distributed models, but continually state that one can't expect them to accurately represent spatially-explicit hydrological processes. If you don't expect a spatially distributed model to accurately represent spatially-explicit hydrological processes, then why use it, instead of a lumped model?

55) Lines 937-969: I would suggest deleting this to shorten and focus your discussion. There is not much reference to your analysis in this section, it's a little bit of a ramble, and you don't say anything about the Richards equations that hasn't already been said many times before over the last 70-80 years.

56) Lines 971-993: I recommend you remove this from the manuscript. You're pontificating about all the ways the land surface models must be fundamentally improved for hydrological modeling, and in doing so you're demonstrating that you have no awareness of the related disciplines of hydrometeorology, plant physiology, and biophysics. All the phenomena that you imply are important, for example, "implies that phenology evolves in response to climate and hydrological controls, thereby creating feedbacks" are in fact known to be important by people in those fields, and others. The upgrades to our model representations that you suggest should happen, have in fact happened, and continue to be upgraded, for example, models that link plant metabolism and water use, or that utilize spatially- and temporally-dynamic root uptake schemes. You conclude by saying that the literature is full of more realistic models for parameterizing stomatal conductance, but you still use a fairly standard version of the PM model with non-local parameters. So I don't think your analyses are very relevant to the state of practice in evapotranspiration modeling. I think you should stay focused on the representation of runoff processes. HESSD

---

## Referee Comment (RC2) · Anonymous Referee #2 · 17 Aug 2016

The manuscript presents 2D distributed physically-based modeling of water flow dynamics at two representative hillslopes differing in geology, soil, and vegetation characteristics. Authors proposed several scenarios with different conceptualizations of hillslope to analyze first-order controls on soil water dynamics and limitations in current modeling approaches. The manuscript conveys an interesting topic, potentially attracting readers of wide hydrological community. However, several points deserve further attention and need to be clarified/improved.

General comments

1. Rather than a rigorous study on hillslope modeling with detailed data-model comparison, authors set up representative hillslopes built on two perceptual models. As a result, no comparison of spatially dependent variables (although measured) was presented. Authors avoided intentionally the first step in modeling. The idea of using the state-of-the-art distributed model for analyzing soil water dynamics in a simplistic representative hillslope segment seems awkward as full potential of the model is not exploited.

2. The hydraulic functioning of the simulated hillslope segment is questioned. Given the vertical height of the 2D flow domain (2 m), maximum soil depth of 1.8 m, and thickness of the soil-bedrock interface (0.2 m), bedrock was excluded at some locations. This hydraulic setting affects deep percolation fluxes across the interface. Furthermore, saturated hydraulic conductivities are too high for soil layer, soil-bedrock interface, and drainage system and too low for bedrock. Can still be laminar flow assumed for near saturated conditions (Richards eq.)? Is there any measurement indicating such high Ks value of the soil-bedrock interface? Can Ks value of the soil-bedrock interface be higher than upper soil? Similarly, bedrock porosity values seem too high, is there any (experimental) justification? Grid size of 1 m used for macropores seems unrealistic.

3. Runoff processes (subsurface lateral flow, overland flow, and deep percolation) and their partitioning mostly depend on hydraulic setup of the hillslope (see point 2). Observed catchment streamflow was compared with the sum of the three runoff component. It would be interesting to see individual contribution of each process to streamflow.

4. Some details of the model were not unveiled in the manuscript. For instance, algorithm of the root water uptake module remains unclear as well as the parameters (not sufficient to refer to previous study). As one hydrological year was considered in the simulations and most of runoff occurred during the winter season (Figure 8), no information was given on snowmelt runoff. More comments are appended below.

5. The manuscript is too long. Some parts can be shorten and condensed (see Detailed comments). Please focus more on hydrology and less on philosophy. Conclusions part resembles Discussion.

6. Discussion should be condensed, there are too many points which are discussed. The results obtained in this study are not related to previous literature, i.e. no deep discussion is provided. Focus on the main aspects of the results. Some discussion parts are too vague and superficial, adding limited value to objectives ("to identify limits in our theories and related physically-based models") and overall knowledge.

Detailed comments

Lines 14-27: This is too long introduction in the abstract. Please shorten.

L16-20: Conceptual models can also be physically-based. The use of conceptual models is incorrect.

L34-6: Not true, internal water storage was not simulated well (see Figure 9).

L79-85: The literature body of recent hillslope 1D&2D modeling applications is somehow limited to a narrow window (mostly of the author's group). For instance, predictions of 2D Richards-based hillslope model with a provision for preferential flow were compared with field data (hillslope discharge and spatially distributed pressure heads) in recent studies.

L110-3: Yes, hillslopes are indeed important in some headwater catchments. However, wetlands/riparian zones may control the runoff generation in other headwater catchments.

L166-8: Is there any source to justify this statement?

L172-4: Add a few references here.

L181-4: A reference is needed here to support the statement on water balance.

Introduction section seems to be quite long, it can be effectively shorten without losing the central messages.

L223-5: Delete "concentration".

L268-70: "...perceptual...".

L311-4: Double mass curve is not well suited for studying annual water balance since it relates runoff to precipitation. There is no provision for e.g. storage changes.

L421-4: Delete "a function of".

L424-6: This is not clear, please explain.

L428-30: The algorithm of RWU is not clear.

L439-42: Any experimental justification for arrangement of the structures? Taken from previous studies on different hillslopes?

L450-3: Define slope angle. Was variable hillslope width considered (total area remains unclear)? The same pertains to Wollefsbach hillslope.

L453-4: Grid size of 1 m seems as a crude approximation for macropores.

L464-6: Boundary conditions are not clearly explained. Does free outflow refer to free drainage BC (with unit hydraulic gradient condition)? Is gravitational flow boundary condition seepage face BC?

L472-3: Ohm*m. In Figure 6B, contour line of 1500 ohm*m is situated in depths ranging from 1.0 m to 3.2 m for 100 m hillslope length. How such spatial configuration could be simulated in 2 m high hillslope segment? Please use the same hillslope segment in Figure 3 as ERT cross-section shown in Figure 6B.

L474-7: Model of van Genuchten assumes zero air-entry value. Thus, alpha parameter is not reciprocal to air entry value. Not clear explanation of the soil hydraulic parameters - list also the parameters of macropores in Table 1. Furthermore, show the (variable) depths of the soil structures in Table 1.

L486-9: I did not find any band generator in Zehe et al. (2010a).

L501-3: Be more specific, what kind of data?

L508-9: Pressure-water content relationship was measured in 0-3 pF range in detail, resulting in large spread of data points. This is not true for smaller pressure head values. These aspects may invalidate the fitted representative curves used in modeling.

L518-22: Boundary and initial conditions are identical for the two hillslopes. Do not repeat the information.

L523: "Model scenarios"

L527-31: Not left boundary (Figure 3CD)? What is the added value of log NSE compared to NSE criterion?

L551: "In VE2.1 scenario, . . ."

L558-9: Instead of "Last not least" use "Finally".

L565-8: Need to say what parameters were changed and what were kept unchanged. Otherwise it is a black-box.

L581-2: Delete "until the onset of the summer period".

L582-3: "Summer period was started with ...".

L585-7: Does this mean that value of the saturated hydraulic conductivity was increased 75 times compared to reference scenario? Reference scenario used already high Ks value.

L593-4: This a harsh break from runoff to 2D saturation distribution, I suggest discussing runoff first and then move to saturation.

L602-5: This can't be concluded, so far only runoff component of hillslope balance was shown. No comparison on hillslope storage was made.

L607-9: I would not say that in case of Wollefsbach, please see the winter period in Figure 8D. To support this statement, provide efficiency coefficients for both winter and summer period.

L610-2: Is hydrological year 2013 meant here? I miss the point. A1 scenario for Colpach has NSE = 0.84, so why discuss smaller NSE? Instead, scenario A2 for Wollefsbach requires discussion (NSE = 0.26).

L613-5: Not clear what is discussed, Weirbach simulation?

L622-4: Please provide the volume proportions (overland and subsurface flow, deep percolation) of runoff for both hillslopes.

L624-5: Infiltration-excess overland flow was most likely not simulated due to extremely high Ks value of (top)soil. Was the extent of overland flow decreased by further increasing Ks value of bulk soil (emergent structures)? Such short high flow events also could not produce saturation-excess overland flow. Please check.

L631-5: Not clear ". . . terrestrial filter properties . . .".

L666-8: This suggests misrepresentation of the soil profile (e.g., soil layering) as well as misparametrization of the soil hydraulic properties.

L670: This paragraph needs a reference to Figure 10A. Evapotranspiration module is treated as a black-box for readers. We are left unaware what parameters were changed. Was there a difference between potential and actual ET fluxes? Did any water stress occur? Add information on Wollefsbach hillslope.

L690-2: Do not write "it may be", this must be exhaustively explained by the model. Less storage above the interface for steeper hillslope setting would lead to increased storage in bedrock. Note that deep percolation was considered when comparison with observed streamflow was performed. This would also cause a delay in simulated runoff compared to reference scenario.

L708-10: Please make a reference to VE2.3.

L732-5: Please reword.

L737-41: Delete "revealed and" and "matching".

L773-6: Instead of "parallel" use "lateral".

L778-80: Instead of "in concert with" use "and".

L788-90: Satisfying match was obtained due to large measurement variability. Many scenarios with different parameter sets would fall within measured soil water content range (even different modeling approaches can provide similar match).

L796-9: Instead of "benchmarking" use "comparison".

L802-4: Given the comments above, I would hesitate to make such statements.

L810-2: The values contradict the previous statement.

L812-5: I do not agree with this statement, see Figure 9.

L828-30: Check KGE values (see Table 2 and Figure 11).

L884-6: Is VE2.3 scenario discussed here?

L886-9: It may be also due to location of a large depression (considered at the hill-foot region in this scenario). Therefore, these statements are not fully justified by the results.

L908-9: Not clear what are SVAT modules.

L911: The message of this section remains unclear. What is suggested here? The need to use distributed model of representative hillslope with spatially uniform rainfall and pET fluxes?

L912-3: Not clear ". . . models encountered to capture flashy . . .".

L936: This section is too vague.

L995-7: This can't be concluded in such a general way. Beside hillslopes, riparian zone may play an important role in runoff generation in some catchments.

L1171-3: Cite this study in HESS.

L1272-5: Improve the reference.

Figure 3: Add dimensions instead of "small section".

Figure 4: Dimensions are necessary.

Figure 9: No green color found. Stick to Colpach and Wollefsbach.

Figure 10: Relative saturation > 1?

Figure 12: Add hillslope location.

Table 2: Check KGE values shown in Table 2 and Figure 11.

---

## Author Comment (AC2) · 17 Aug 2016

Reply to Anonymous Referee #1:

**Ralf Loritz (RL):** We would like to thank the Anonymous Referee #1 for her/his insight and thoughtful comments. We are thankful for the effort the referee put into this review in the form of the high quality of his comments. In the revised manuscript we will follow many of the reviewer's recommendations, because this will definitely improve our study. Furthermore we see from the comments that some parts of our study need better or more detailed explanations which we aim to provide in a revised version of our manuscript.

*Reviewer: Summary and Recommendation:*

*In this paper the authors address two basic questions:*

*1) If you have a lot of spatially-distributed information about the geology and soil-hydraulic properties in a catchment, can you parameterize a high-dimensional, spatially-distributed model (without any calibration or inverse optimization) to accurately represent water flow within a single 2-d hillslope, based on that existing knowledge?*

*2) If your knowledge-based (not optimized) model domain and parameterization prove reasonably representative, can you then extrapolate this representative 2-d hillslope across the 3-d volume of the entire catchment, to simulate hydrograph dynamics and the annual water balance for the entire catchment?*

*To address these questions the authors employ a Richards-equation-based model with evapotranspiration module and overland flow routing modules. They apply the model to simulate hillslope-scale soil moisture dynamics and water-balance partitioning and after extrapolation, whole-catchment streamflow dynamics from two catchments in Luxembourg with varying geology, topography, soil, and vegetation. In addition to the modeling, their analysis includes extensive and impressive data sets representing spatially distributed soil-hydraulic properties, geologic features, plant transpiration, and topography. These questions, the observations, and the methodological approach adopted here are of interest in scientific hydrology and would be received with interest by readers of HESS. The writing is mostly clear in a grammatical sense, though somewhat desultory and technically ambiguous in many areas. The model setup is technically sophisticated in many ways, though not so in others. The graphics are of very high quality. The organization of the paper is logical, though as I suggest below, there are significant portions that might be omitted to maintain a consistent focus of the paper throughout. Overall there were too many competing objectives in the paper. As such, any salient result or conclusion is hard to discern. The results and discussion could be greatly improved.*

*I recommend that this paper could be accepted for publication in HESS, but I believe some major revisions are needed beforehand, possibly including revised simulations and results. Those suggested revisions are outlined in the following g General Comment's section, with references to more specific Technical Comments.*

**RL:** We thank the reviewer for this comment. Additionally to the two questions, the paper addresses a third question. It deals with the problem of how to find the most important information or data source

that is needed for setting up hillslope models. This is the basic idea of the virtual experiments. We agree with the reviewer that this part of the study can be much improved, but we do not agree that the reported results are not of interest.

*Reviewers: General Comments:*

*Reviewer: The manuscript can be improved upon significantly by reducing total length, omitting or clarifying the use of excessive jargon, and possibly removing sections of the paper to minimize superfluous and unfocused commentary. Some specific instances are noted in Technical Comments 5-10, 24, 27, and others. The manuscript may be much improved by omitting much of the commentary about modeling evapotranspiration, along with the virtual experiment 3 (VE3) and associated results, and rather focusing on the importance of explicitly representing (or not) landscape heterogeneity for the purpose of simulating hydrographs. The authors talk quite a lot about all the uncertainties associated with ET modeling in hillslope/catchment hydrology, but their modeling approach does not reflect the state of the science (e.g. as presented in disciplines such as hydrometeorology and plant biophysics), so this discussion does not seem warranted. See Technical Comments 12, 29, and 56. The methodological approach is inadequately described in many instances, with some revision being needed. See Technical Comments 16-26, 36, 39, and others. There are some aspects of the model domain and parameters that seem very unrealistic. For example, the bedrock for both modeled catchments is parameterized to have porosity of 40-45% (Table 1). That's comparable to, or greater than, the porosity of many soils. I can't imagine how Schist a metamorphic crystalline rock can be composed of 40% air space. This is certainly not consistent with most reported porosities for Schist, which are typically 10% or less. This will have a large impact on the flow simulations, since about half of the hillslope domain is bedrock. If there is some justification for this, and some other aspects of the model, then perhaps the only revision that is needed is to provide that justification. Otherwise, many of the simulations may need to be repeated with more appropriate parameterization. See Technical Comments 19-26, and others.*

*I strongly suggest that the authors consider refocusing this paper on two subject areas. First, concentrate on the modeling of spatially-distributed soil moisture dynamics and the temporal dynamics in the hydrograph. You use a spatially distributed model but nowhere do you assess the models ability to accurately represent any spatial pattern. You have an enormous amount of interesting spatially distributed data, so this could become one of the most rigorous tests of the Richards equation model at the hillslope scale ever published. See Technical Comments 33-34, 47-48, and 54. I recommend the second focal point to be the argument that extrapolating the parameterization of a single 2-d hillslope to an entire 3-d catchment may, or may not, be defensible. At present this is stated as an objective, but the results and discussion are ambiguous, or possibly in disagreement, about this point. Showing these outcomes would be challenging enough, and a good contribution to contemporary scientific hydrology. Toward that aim, I suggest omitting the virtual experiments and associated discussion. I found the virtual experiments and the associated discussion around them to be desultory and vastly oversimplified. It was not clear to me how they relate to any previously stated objective. Those virtual experiments mainly consisted of changing a single variable (e.g. total relief, timing of bud break for plants, or hydraulic*

*conductivity), and speculating broadly about the implications of the resulting simulations. See Technical Comments 50, 52-53, and 55-56.*

**RL:** We agree with the reviewer that the manuscript can be streamlined, particularly with respect to the discussion of the ET approach. We will further elaborate this in the technical comments.

The reviewer is also right that there are too many related objectives in the paper. We thought and still think that an exhaustive hillslope model study should address the two questions summarized by the reviewer and our proposed third question about the importance of different data sources. We agree that an exhaustive treatment of all these three questions is difficult to be digested within a single paper. A natural point to streamline the study is thus to remove most of the virtual experiments and treat them in a more exhaustive manner in a separate study - in this respect we look forward to the editors' advice.

We will focus our revised manuscript with respect to the two below summarized question of the reviewer and explain in more detail how we parameterized our two models. Our goal is to make our model structure and the choice of parameters as transparent as possible. Hence we will set up our vegetation parameters as far as this is possible with observed values (*technical comment 12*) and will rework and better explain the way how we chose our macropore parametrization (*technical comment 19 & 20*). Furthermore we will show that the large heterogeneity in the soil samples cannot be grouped by landscape characteristics in a simple manner (*technical comment 16*). Finally, we will try to better account for the soil moisture variability along the hillslope and be more careful with the term "distributed" (*technical comment 34*).

*""*

*1) If you have a lot of spatially-distributed information about the geology and soil-hydraulic properties in a catchment, can you parameterize a high-dimensional, spatially-distributed model (without any calibration or inverse optimization) to accurately represent water flow within a single 2-d hillslope, based on that existing knowledge?*

*2) If your knowledge-based (not optimized) model domain and parameterization prove reasonably representative, can you then extrapolate this representative 2-d hillslope across the 3-d volume of the entire catchment, to simulate hydrograph dynamics and the annual water balance for the entire catchment?*

*""*

**Reviewer:** *Last, the authors seem to be advocating that high-dimensional, spatially-distributed models will always be wrong for a variety of reasons, but that they're still important for helping us learn about which state-variables and flow processes dominantly control the emergent streamflow dynamics at the catchment outlet. I think this is a relevant and worthwhile argument, but I encourage the authors to better focus their writing on this topic. This effort may be aided by omitting some other sections of the paper, as noted above. At present the authors present some seemingly conflicting (at least to me) statements about what exactly is the merit of taking this approach, and just how well it did or did not work out for them. See Technical Comments 34, 46, 47-48, and 54.*

**RL**: We will restructure our discussion in the revised manuscript. Although we would rather use the term "approximation" than "wrong" – the clue is to get the "best" approximation which is as simple as possible but not oversimplified. We come back to this point when dealing with the related technical comments.

**General comment: Bedrock parametrization**

*Reviewer: There are some aspects of the model domain and parameters that seem very unrealistic. For example, the bedrock for both modeled catchments is parameterized to have porosity of 40-45% (Table 1). That's comparable to, or greater than, the porosity of many soils. I can't imagine how Schist a metamorphic crystalline rock can be composed of 40% air space. This is certainly not consistent with most reported porosities for Schist, which are typically 10% or less. This will have a large impact on the flow simulations, since about half of the hillslope domain is bedrock. If there is some justification for this, and some other aspects of the model, then perhaps the only revision that is needed is to provide that justification. Otherwise, many of the simulations may need to be repeated with more appropriate parameterization.*

**RL**: You are right that Schist does not have a porosity of 40%. But this Schist formation is highly fractured and additionally covered by periglacial deposits. We are only modelling the upper 2 m of the subsurface, and hence only the upper meter of the weathered bedrock. Citing Wrede et al. (2015), the periglacial deposits "*may store significant amounts of water, more than expected from an analysis of soil mapping information alone*". Furthermore we assume a saturated hydraulic conductivity $K_s$ (m s$^{-1}$) of 5x10$^{-9}$. We chose such a low value because we expect no major groundwater body beneath the hillslope (Bos et al., 1996). Hence there is almost no water flow through the bedrock. Deep percolation (water leaving the hillslope through the lower boundary) in our reference model makes up around 0.001%. To follow up on the reviewer's comment we repeated the simulation for the Colpach with a reduced bedrock porosity of 0.1 (Figure 1 & 2). As you can see there bedrock porosity is not a sensitive parameter. We will stress this in the revised manuscript and explain the low sensitivity of this parameter because of the low permeability to avoid further confusion about this point.

[Figure]

**Figure 1 Observed (grey) and simulated discharge of the reference model (blue) and the reference model with the reduced bedrock porosity of 0.1 (red).**

[Figure]

**Figure 2 Double mass curves of the observed (grey) and the simulated discharge of the reference model (blue) and of the reference model with a reduced bedrock porosity of 0.1 (red).**

Technical comments:

*Reviewer: Comments 1, 2, 3, 4, 8, 13, 21, 25, 27, 28, 38, 40, 42, 43, 45, 49.*

*RL:* We agree with the reviewer in these points. These mistakes/statements will be corrected and/or rephrased.

*Reviewer: 5) Line 67:* *Not immediately clear to the reader what distinguishes a "conceptual model" from a "perceptual model".*

RL: We apologize we thought these terms need no further specification. We will give a short definition to improve the clarity of the presentation. A conceptual model is a bucket-style model, for instance the HBV beta store for soil moisture accounting or a linear reservoir. A perceptional model reflects our imagination on the dominant processes and structures that govern for instance runoff formation in a catchment. Personally, we consider the classification of hydrological models into conceptual and physically-based models inappropriate; since it implies that conceptual models are not based on physics and that physically based models do not have an underlying concept. We actually prefer the definitions of Gupta et al. (2012) but we used the terms nevertheless, because we thought it is well-established hydrological jargon and to avoid confusion.

**Reviewer 6) Lines 85-88**: *Consider rephrasing or deleting. Not clear what is the message of the sentence. The result of any mathematical model must be compared to observations in the modeled system. This goes without saying, and certainly doesn't require a supporting citation. Perhaps I am just not clear what you mean by "benchmarking".*

**RL:** With this we mean that physically-based models can be evaluated against observations of stream flow, soil moisture states and even tracer data in a straightforward manner. This is not the case for conceptual models and thus an advantage. We will replace the term "benchmarking".

**Reviewer 7) Lines 49-97:** *This is very clearly written, but for sake of making your paper as concise as possible and therefore more likely to be read in full. You might consider abbreviating the section, or deleting. It doesn't assert much, or highlight some problem with the status quo in catchment modeling. It mainly states that model-based analysis are useful for learning about hydrological systems, which I think is already acknowledged ubiquitously in the community of hydrological scientists. It's your call, but there is no shortage of papers on hillslope modeling, and I always prefer to read a more concise one than a longer one.*

**RL:** You are right. We will definitely streamline this paragraph, though it contains a key point – the picture/image idea. By using a suitable color code for plotting for instance soil parameter of a two-dimensional hillslope, this plot resembles a perceptional sketch of a hillslope or catchment. In fact this "structural setup" can be tested against augers or ERT images. *Is the bedrock in our model in the same depth as observed or consistent with available ERT images?* Moreover, such information could be used for setting up the hillslope model. This is not possible with "conceptual models" because they are not spatially explicit and not thermodynamically consistent (fluxes are not driven by gradients). Hence, we can use much more information which is independent from our target data for an a-priori setup of the hillslope model.

**Reviewer 9) Lines 113:** *"functional behavior of catchments of organized complexity" is hard to interpret. Caution in using too much ambiguous jargon.*

**RL:** This formulation will be removed from a revised manuscript. However the term "organized complexity" was coined by Jim Dooge (Dooge, 1986) to characterize catchments that already exhibit too much heterogeneity to be treated in a deterministic, physically based way but a are yet too small for a conceptual treatment.

**Reviewer 10) Lines 128-130**: *Consider rephrasing using the plainest language possible, since this is seemingly an important part of your rationale statement for the study*

**RL:** This is indeed an important part of our rationale, thank you for pointing out that our text is not easy to read and interpret. We agree that this should be improved and we have tried to rewrite these lines to convey the message better. We have changed the sentence to: "We propose to start with the perceptual models (Top panel of Figure 3), which provide qualitative information (such as impermeable bedrock with shallow periglacial, highly porous soils on top, with a network of vertical and lateral flow paths) and to transfer this into a first-guess parametrization of the hillslope using the available data or literature values."

*Reviewer 11) Lines 137-140: Rephrase "behavioral physically model structures". Also, comparing model outputs to observed data sets (like tracer time series) doesn't inherently reduce the number of degrees of freedom in the modeling procedure. It might help constrain parameter values. If that observed data set somehow informs the modeler that a particular parameter is unnecessary, or that a spatially-distributed domain can be adequately represented in a lumped way, then the degrees of freedom might be reduced. Are the works you cite here examples of the latter? For example, in the application of Richards equation with the van Genuchten-Mualem soil-hydraulic model, you can't just decide based on some observation that you no longer need the shape parameter in the hydraulic model – it still has to be there. If the observational data set leads you to a coarser grid resolution for the domain, then that would be a reduction in degrees of freedom, since the number of spatially distributed elements where the equation is solved/averaged is reduced. But really, for multi-parameter models that are spatially distributed, the degrees of freedom are always grossly high, not even considering the fact that we most often ignore anisotropy and hysteresis in soil hydraulics in hillslope to catchment-scale applications. That's the whole motivation for lumped models, right?*

**RL:** Here we refer to studies of Klaus and Zehe (2011) and Wienhöfer and Zehe (2014) and how they simulated flow and tracer transport at a tile drained field as well as at a forested hillslope in a two-step procedure. In a first set they used a basic two 2d hillslope model and represented vertical and lateral preferential flow paths by different spatial densities and hydraulic conductivities. From the 400/120 trials several networks reproduced the observed till drain outflow/hillslope runoff in acceptable manner, though the networks were quite different. These hillslope structure were hence equally likely since they all produced the same amount of fast subsurface flow, either through a higher number of less conductive macropores or though "fewer more conductive macropores". In second step they simulated tracer transport through these behavioral hillslope structures (or architectures if you wish) and which reduced the number of behavioral ones to 4 or even to zero. Tracer data impose an additional constrained as not only the flow (the filter velocity) must be reproduced but also the transport velocity in the porous medium. This is what we mean with reducing the degrees of freedom in the "model space".

**The sentence will be rephrased to:** But the number of model structures that are physically meaningful can be reduced by using complementary observations such as tracers.

*Reviewer 12) Lines 166-184: Evapotranspiration is represented in a rudimentary way in many hydrological models precisely because those models have the primary aim of predicting hydrographs. For*

*modelers with this primary interest, there will inevitably be a greater effort spent on representing such processes as non-equilibrium flow than on ET, because the former is of more interest. Models aimed at predicting streamflow use long-standing, and possibly antiquated ET models, because they're convenient, not because enhanced knowledge of stomatal dynamics and plant phenology is absent. Tree physiologists and hydrometeorologists have highly advanced understanding of these phenomena, and their discipline-specific models reflect that. These arguments are worth keeping in mind when you're noting the "uncertainty in the community on how to represent plant physiological controls on transpiration in hydrological and land surface models." Is it uncertainty, or just lack of interest/effort to study and implement models that reflect contemporary knowledge in plant physiology and boundary-layer biophysics? As an example, you go to great lengths here to incorporate small-scale non-uniformities into the subsurface flow domain, but you use a pretty standard version of the PM approach for ET that's been around for over 30 years now. You assume homogenous land cover in Colpach catchment, and you use vegetation parameters from a non-local catchment in Germany, with phenology assumed invariant from year to year (I assume that's what you mean by "fixed"). With that model setup, it's really not appropriate for you to be talking about how uncertain the community is in how to represent the complexity of these processes in models. It wouldn't be very complex for you to considerably improve this standard version of the PM model just by using site-specific data, a dynamic representation of phenology, and to accurately reflect the relative abundance of forest versus other vegetation cover in the catchment. I don't want to be overly critical, because some assumptions and simplifications are always made in modelling. My main point is that you don't want to go on philosophizing about how we learn from still-uncertain models, if the model you're employing is not nearly state-of-the-art, or not parameterized nearly as carefully as it could be.*

**RL:** The reviewer is more than right with this point, the land surface model community uses far more sophisticated approaches for ET catchment modelers usually do. In fact it was not our goal to highlight our evapotranspiration routine as sophisticated, but to stress the deficiencies of the approach. Our opinion is that hydrological modeling does not end with a successful simulation of runoff especially if ET is the major outward flux of the catchment half of the year. Furthermore, ET is one of the main controls of the fill level of the storages in hydrological model. Since in almost all hydrological models the bulk of runoff is produced as a function of the model state, ET becomes increasingly important in long-term simulations.

The reason why more sophisticated approaches are ignored in the catchment modeler community might be, as the reviewer states, laziness or disinterest because simple models "seem to do the job" when the focus is on streamflow. We hence agree that the transfer of an annual phenological cycle from one catchment to another and the assumption of temporal invariance is crude. However, it is not at all unusual practice and often not mentioned in model studies at all. But the reviewer is right we did not give our ET parametrization the same attention as we have done it with other parts of our model. We will follow the reviewer's advice and set up our ET model in a revised manuscript as far as this is possible with observed values. In both catchments we will use the temperature index model from Menzel et al. (2003) to define the start and end of the vegetation period. We use this model because it could successfully identify the tipping points between the summer and winter season in both double mass curves. In the Colpach catchment we have access to observed LAI values for August and September and

we will use them within future simulations. Unfortunately we have no LAI values in the Wollefsbach and hence need to take values from the literature. We will use the reported values for corn by Breuer et al. (2003), which are in fact close to the values we already used. Additionally we will add a table to a revised manuscript were we will list the vegetation parameters from our ET routine.

*Reviewer 14) Lines 240-242*: *This concept certainly precedes the work of Zehe 2014. For example, consider these important papers, which are notably absent from works cited in your introduction:*

**RL:** We are sorry that we missed the above mentioned two studies (probably we missed even more) and will consider adding them to our revised introduction. Thank you for these references.

The term "functional units" was first proposed by Zehe et al. (2014) as an as advancement of the HRU concept. The latter is of course much older. The core idea of the functional unit concept is that similarity with respect to the energy balance and runoff formation emerges at different scales (the small field and the hillslope scale), because the land surface and subsurface characteristics controlling the related gradients and "resistance terms" have different characteristic length. The concept that hillslopes are key elements controlling runoff generation in the landscape is of course much older, and goes back to Troch's Hillslope Boussinesq model (the work of Berne deals with this). However, the term "functional unit" states clearly that one can learn in a representative fashion about runoff formation by targeted clustering of multiple observations at the hillslope scale and that the hillslopes are key building blocks for setting up hydrological models. In fact this is shown within our study. We revise this passage along these lines.

*Reviewer 15) Lines 323-324*: *Please provide some explanation of what you mean by the onset of vegetation period? Presumably that would be timing of leaf development for crops and deciduous plants, but evergreen plants will be physiologically active even through winter and spring, albeit at lower rates than in summer. Then there is of course an extended period when crops and deciduous plants transition from no foliage to the maximum leaf area they will obtain that year. Than transition can span weeks to more than a month.*

**RL:** With the term "onset of the vegetation period" we mean the bud break of the deciduous trees as a main seasonal influence on evapotranspiration. We refer here to Beech trees as these are dominant in the Colpach. In the "VE3" we mainly adjusted the day of the first flushing, which of course implies that time to full coverage is the same as in the German Weiherbach.

*Reviewer 16) Lines 355-368, and Figure 7:* *The variability among measured moisture retention curves for your soil samples is remarkable. Can they be logically grouped in any way, for example, by landscape position or soil depth? If so, a color scheme to illustrate that would be very interesting. Some additional detail about where, and at what depths, the soil samples were taken is needed.*

**RL**: Yes, the variability is remarkable and this is exactly what we intended to show. Young soils on periglacial slope deposits prevail in the headwater. They exhibit large heterogeneity which cannot be grouped in a simple manner as detailed in Jackisch (2015) and Jackisch et al. (2016). This is due to a) the general mismatch of the scale of 250 mL undisturbed core samples with the relevant flow paths and b) the high content of gravel and voids, which affect the retention curve especially above field capacity and concerning its scaling with available pore space. For the study at hand the focus does not lie on this analysis but on the implications coming from it: A representative retention curve from measurements can be directly and successfully employed in the physically-based model Catflow, even for a heterogeneous headwater catchment like ours. We think that the latter is a noteworthy and non-intuitive finding. To clarify this, we will update the graph adding information about sampling depth and relative distance to the stream network. Moreover we will revise the description accordingly.

**Reviewer 17)** *Lines 385-388: Are the sapflow sensors collocated with rainfall and soil moisture measurements? Please elaborate on the exact type of sensors, depth of installation into trees, and other important details about their operation.*

**RL:** We will add further information about the sap flow measurements in our manuscript: "Furthermore we use sap flow measurements from 61 trees at 24 of the sensor cluster sites. The measurement technique is based on the heat ratio method (Burgess et al., 2001), sensors are East30Sensors 3-needle sap flow sensors. As a proxy for (the volume of) sap flow we use the maximum sap velocity of the measurements from three xylem depths 5, 18 and 30 mm as recorded by each sensor. To represent the daytime flux, we use 12-h daily means between 8am and 8pm."

Additionally we would like to highlight that the detailed spatial and temporal analyses of this sap flow dataset are currently being revised in a manuscript by Sibylle Hassler. Here we want to use the sap flow data mainly as an additional possibility to test whether our model represents a major hydrological flux which is clearly difficult to measure realistically.

**Reviewer 18) Lines 424-426:** *Any consideration of the soil-moisture dependence of vapor diffusivity in soil? It varies as a power-law function of air-filled porosity, over about 4 orders of magnitude.*

**RL:** Catflow doesn't account for movement of water vapor in the pore space. This might be a shortcoming particular in arid areas. We are aware of the work of Chris Milly on this issue.

**Reviewer 19) Lines 432-446:** *How are the probabilities of the Poisson process determined? Is this based on some knowledge that informs your perceptual model, or are though chosen arbitrarily? Or, do you determine some best-performing parameters based on a sensitivity/optimization process? Please describe in a little more detail, and consider providing an image of what these structures look like in your final model domain. Is this what we're seeing in Figure 3C, D? If so, please just allude to that figure here in the text.*

**RL:** We again apologize this was not written clearly enough. Normally the values for the probability of the Poison process can be estimated for instance based on spatial mapping of worm burrows (Zehe and Blöschl, 2004). However the parameterization in this study was chosen rather arbitrary since only qualitative information's were available for this parameter. In our discussion paper we followed the studies of Klaus and Zehe (2011) for the Colpach and the parametrization of Wienhöfer and Zehe (2014) for the Wollefsbach. The depth of the vertical macropores of around 1m with a standard deviation of 0.3 m is based on the results of Jackisch et al. (2016). By incidence (or not) this worked well that there was no further sensitivity or optimization process necessary. We will add a reference to Figure 3 C, D in the revised manuscript.

*Reviewer 20) Lines 453-454: Considering the horizontal resolution of model elements is 1-m, I'm wondering how realistically the vertically-oriented, preferential-flow zones can be represented? Certainly the macropores in your photographs are not 1-m wide. Representing their tortuosity by vertically-offset grid cells of 1-m width seems like a gross distortion as well. Can you discuss how you rationalize this model domain and horizontal grid resolution, especially with regard to those grid cells that are imposed to represent preferential flow structures? Also, can you provide detail about the dye-irrigation studies from which the photograph was derived? The pictures are very insightful. However, it is well known that irrigation studies often impose exceptionally high input fluxes, and with sprinkler systems that exhibit enormous spatial variability. Can you comment specifically on the irrigation rates and the spatial uniformity of the irrigation system, and how those irrigation rates compare to the frequency distribution of rainfall intensities that occur at these field sites?*

**RL:** This is a good point that needs to be explained better. At a grid size of 1 m the Poisson process does often generate several macropores in the same grid cell as the generation process is carried out at a much smaller grid (1-2cm). All macropores contribute to the enhanced conductance of the model element; the enlarged conductance is hence an effective representation of the subscale macropores. This infiltrability is sufficient to allow a fast flow to the bedrock in the Colpach and drainage in vertical and lateral direction in the Wollefsbach. To improve this we usually work with an adaptive grid size of a few centimeters, generate the macropores, and reduce the grid resolution in areas without macropores. This was done in the Wollefsbach and will be done in a revised manuscript in the Colpach.

We employed the same approach in the Colpach in Figure 3, and simulated the runoff of the hydrological year 2014. The result is basically the same as with the coarser grid (Figure 4). This corroborates that the spatial extent of the macropores is not too important as long as their combination represents the total amount of fast flow. In a revised manuscript we will reduce the size of macroporous grid cells in the Colpach from 1 m to 10 cm (Figure 3) similar to the hillslope model used in the Wollefsbach catchment to avoid confusion. We again use the macroporous medium proposed by Wienhöfer and Zehe (2014) which corresponds well with reported maximum velocities from Angermann et al. (2016) in the Colpach catchment. We will further use a fixed distances of 2 m for the lateral distance of the vertical macropores in the Colpach and of 3 m in the Wollefsbach instead of the Poison process to make our study more transparent. We chose this value again rather arbitrary with respect to create an image of the perceptual model and on qualitative information on macropore flow from field experiments (Jackisch et al., 2016).

We can add various model runs where we change the lateral distance of the vertical macropores to our study. But we would like to highlight that such macropore sensitivity studies with Catflow are already published (Klaus and Zehe, 2011; Wienhöfer and Zehe, 2014).

[Figure]

**Figure 3 Section of the Colpach hillslope with a reduced grid size for macropores.**

[Figure]

**Figure 4 Observed (grey) and simulated discharge (blue) of the reference model with the reduced grid size for macropores.**

We will add the following information to the article. The dye tracer images, in Figure 3B and D, were obtained with high rainfall intensities (Jackisch et al., 2016). The aim of these rainfall simulations was to visualize the macropore networks in the topsoil. For actual rainfall events, both in reality as well as in the model the degree of preferential infiltration depend not only on the structure of the macropores but also on the rainfall intensity and the antecedent moisture content.

**Reviewer 23) Line 479:** *Here again, it would be good to know about the depth distribution of the soil samples collected for hydraulic characterization. Did any actually come from that depth?*

**RL:** We will provide information about their location and the depth distribution (See technical comment 16).

**Reviewer 24) Lines 486-490:** *This sounds wonderfully sophisticated. I have no idea what it means. I'm probably not alone in that regard. It seems like important information about how spatial heterogeneity of hydraulic properties are generated in the model domain, so maybe a sentence or two in plain language to build the intuition of the reader/s that are not intimately familiar with this jargon.*

**RL:** We have adapted the article as follows: **"**We added correlated noise to the hydraulic conductivity**.** To this end we generated a random field of ln(ks) with the observed mean and a variance of 2 using a turning band generator. As we had no local information on the correlation length, we used a range of 5 m which corresponds to the range of soil moisture observations (Zehe et al., 2010) found for a distributed soil moisture network in a forested site in the Ore Mountains. We used a spherical variogram function with a nugget of 0.5 and a sill of 1.5.**"**

Since the variability in the soil retention properties cannot be easily explained by their position and depth it is not straight forward to implement different soil layers in the hillslope model (See technical comment 16). But we do know from numerous experiments that the skeleton fraction in the Colpach is rather high with values above 50% in deeper soil layers. In the revision process we will try a model run with reduced porosity in the deeper soil layers.

**Reviewer 26) Line 527-530:** *You're simulating flow through a single 2-dimensional hillslope profile, so how can you compare the composite discharge (overland flow, subsurface flow, and deep drainage) to the measured streamflow for the whole 3-d catchment? Are you integrating the hillslope response over the entire 3rd dimension of the catchment? Please explain this. Also, do you think it is appropriate to include the deep drainage flux in this composite outflow when comparing to the stream hydrograph? It could conceivably travel through an aquifer system that discharges outside the boundaries of your catchment, no?*

**RL:** We compare specific discharge observations and specific flow simulated with Catflow, by normalizing the former with the catchment area and the latter with the hillslope area mm ((l*h)/m^2). We will add a short explanation. Deep percolation is included as stated above but does not change the result significantly since it is quite low (0.001% of the overall discharge). We agree with the second point.

*Reviewer 29) Line 566-568:  The important trend for the catchment water balance is the timing of the leaf area expansion, maximum, and decline (in fall).  The leaf area is the dominant control on transpiration and net radiation. So, I don't understand how or why you change the timing of phenology without changing the temporal dynamics of leaf area. Please explain the rationale for this*

**RL:** Sorry for being unprecise here. We shifted the start and the end of the LAI cycle from the original values to the values predicted by the temperature index, while not changing the LAI values. If both the start and endpoint are shifted, the cycle within remains the same. Since this is a virtual experiment, this part will be removed in a revised manuscript and the vegetation will be parameterized differently in our model (see technical comment 12).

*Reviewer 30) Line 594: Figure 9B, rather than 10B?*

**RL:** We refer to the simulated saturation patterns in 10 b showing the 2 d pattern in summer and winter. We will stress this.

*Reviewer 31) Lines 614-615: Are you talking about Weierbach catchment in Germany? It's irrelevant. I suggest you delete that and stay focused on your catchment*

**RL:** We are sorry. It is a little confusing. We talk about the Weierbach catchment which is a headwater of the Colpach in Luxembourg and thereby in the same hydrological landscape. This transferability is hence relevant and corroborates the hypothesis on functional units. The Weiherbach in Germany is mostly written with an "h". We will clarify this in our manuscript and apologize again.

*Reviewer 32) Line 619-622:  Run-on sentence that is very hard to interpret.  Please rephrase. Also, please explain what inference you think is made possible by comparing NSE with log(NSE).*

**RL:** We will rephrase this passage- The NSE was chosen because it is a common measure for the quality of model results in hydrology with regard to high flows; logNSE a better quality measure for low flows.

*Reviewer 33) Lines  622-627: These  statements  are  questionable. Are  you  claiming  that infiltration-excess  overland  flow  is  occurring,  or  overland  flow  due  to  saturation  excess?  You say that the model erroneously generates overland flow in the summer in Wollefsbach due to convective storms. The*

*saturated-hydraulic-conductivity parameter you report in Table 1 is 2.9 x 10-4 m/s, or about 1.8 cm/minute. This is quite a high hydraulic conductivity what one might expect for coarse sand. Are your surficial soils sandy? Are those convective storms producing rainfall flux greater than 1.8 cm/minute? Seems doubtful storms like that would occur frequently. How is the model generating so much overland flow if the rainfall rates are (I assume) always considerably lower than the saturated conductivity? In Figure 9D it looks like your simulated-average-soil moisture is in excess of almost all the measured time series. Maybe you're way overestimating soil-water storage and generating saturation-excess overland flow, rather than infiltration excess overland flow. If that's the case, and if it's saturation-excess overland flow, then you can't immediately assume that enhancing Ksat to represent soil cracks is the next, necessary step to improving the model. You need to get the soil moisture dynamics correct before you can go off exploring that speculation. You're Ksat value is already pretty high, is it based on measurements? Also, it's somewhat a shame that you have all those soil moisture observations, and a spatially-distributed model, but you only compare the mean-simulated soil moisture (assume it's the mean) to the observations. The spatially-distributed model gives you lots of spatially-distributed results to compare to spatially-distributed observations. If you're just going to look at the average-simulated soil moisture, you're giving up all that detail that is provided by the model, and one has to ask why not just use a lumped model? You should compare simulated soil moisture at specific points in the landscape to the observed soil moisture at those same points*

**RL:** Good point, we agree that ksat is large, though it has been measured like that. Hortonian overland flow occurs in desert catchments, because of wetting problems due to the extreme dryness, although ksat is even larger there. So speaking about the real system, Hortonian overland flow definitely occurs in the Wollefsbach catchment, which is also visible in frequent erosion events (Martínez-Carreras et al., 2012). With respect to the generation of Hortonian overland flow within the model is not ksat that counts but ku(theta). Note that the observed rolling mean of the topsoil moisture and the average of the simulation in summer are between 0.3 and 0.2 (Figure 9 c in the Discussion paper). This implies an unsaturated hydraulic conductivity of $3.5 \ 10^{-11}$ m/s (compare Figure 5). With such a value the system definitely develops Hortonian overland flow (both the model and the real system). The key question (in real systems) is whether locally generated infiltration excess reaches the stream or not. The latter depends on the question whether a connected flow path of low infiltrability exists, or not. We admit that a proper investigation of overland flow path connectivity is not within the scope of a 2 d hillslope model.

[Figure]

Figure 5 : Unsaturated hydraulic conductivity of the Marl soil at a value of 0.3 volumetric water content.

*Reviewer 34) Lines 631-635:* *I am quite confused by this statement.  You are using a spatially-distributed model, so why are you claiming that it's unrealistic to expect the model to accurately represent the spatially-distributed nature of soil-moisture dynamics?  It should be able to represent at least coarsely the spatial distribution of soil moisture, for example, differences in upslope versus downslope positions, or differences in areas overlying saturated bedrock depressions versus those areas where the bedrock roughly parallels the land surface.  Again, if you don't expect your spatially-distributed model to accurately represent any of these spatial patterns (which are important for runoff), then why are you using a spatially-distributed model to begin with?*

**RL**: Thanks for this important point and sorry for being unprecise. We have of course a spatially distributed model along the hillslope. We believe that the full spatial variability of the soil moisture data is caused by 1.) spatial heterogeneity of rainfall, 2.) heterogeneity of soil properties and 3.) variability along the hillslope. Although our model cannot account yet for 1.) and 2.), we already account for the variability along the hillslope. In the revised manuscript we will use virtual observations along the hillslope in 10 and 50 cm depth (Figure 6). By doing so we can better account for the variability of the soil moisture observations in our 2 d profile (Figure 7). As we are not modelling one particular hillslope, we think a site-specific comparison is out of scope. Nevertheless we will try some color coding for the position of the soil moisture observation as well as for the virtual observation. But we would like to stress that the soil moisture observation similar to the soil water retention properties are not easy to classify by different landscape positions (For example by: up- or downslope).

[Figure]

**Figure 6 Sketch of our 2d representative hillslope with 20 virtual observation points. The position in lateral direction was chosen randomly the position in vertical direction is 10 cm.**

[Figure]

**Figure 7 Top soil moisture observation in 10 cm depth against an ensemble of 20 virtual observations in the respective depth in vertical direction and randomly chosen position in lateral direction.**

In principle it is of course possible to account for the spatial heterogeneity of soil properties by means of random fields, as proposed by the reviewer. To illustrate this we generated a random field of porosities with an unconditional, sequential gaussian simulation using the R package "gstat". One may also optionally account for a reduced porosity of deeper soils layers by reducing the mean porosity with depth. Figure 8 corroborates that a simulation within such a heterogeneous domain could resemble most of the variance of the soil moisture observations.

[Figure]

**Figure 8 Observed top soil moisture observation in 10 cm depth as well as an ensemble of 20 virtual observations in the respectively depth in vertical direction and randomly chosen position in lateral direction. The soil parameter were varied in the observed range using random fields with a linear trend.**

***Reviewer 35) Lines 641-642:*** *This statement is inevitably true for every catchment in the world, and hence does not rely on any measured or simulated soil moisture dynamics. Maybe just delete.*

**RL:** Sorry for being unprecise. We meant that runoff generation in the Marls (Wollefsbach) is partly intensity-controlled due to the occurrence of Hortonian overland flow (as outlined above). We remove this statement to stay brief.

***Reviewer 36) Lines 671-684, and Figure 10A:*** *This material needs much improvement. First, measuring sapflow in trees is a delicate business, with major discrepancies existing between methodologies, and significant errors arising from inexact application of methods (e.g. due to radially-varying flux rates within the sapwood, a well-documented phenomenon in the tree physiology literature). There are several reviews of this topic in the plant physiology literature (e.g. Steppe et al. 2010, Agricultural and Forest Meteorology, 150). You have provided essentially no detail about the nature of your field-based sapflow measurements. Also, how are you normalizing the measurements? Second, you use a version of the PM model to simulate water vapor flux from the plant canopy, not sapflow (L3 T-1). The two cannot be assumed to be equal. If you consider the tree as a system spanning the point of your sapflux measurement (breast height on the stem) to the canopy, then the sapflow (input to the system) only equals the volumetric flow out of the leaves (outflow from the system) if the system is in steady state (i.e. inflow = outflow and storage is constant). Water storage in the tree stems and canopy foliage is dynamic. I am not sure how good or bad is the assumption of steady state in your system, but it is*

*certainly an assumption you should carefully consider and provide some justification for why this comparison (between measured sapflow and modeled canopy vapor flux) is valid. Without that, you should probably omit this text and figure 10A from the manuscript.*

**RL:** We agree with the reviewer that sap flow measurements are delicate and not directly comparable with simulated ET from PM models. We will stress this in the revised manuscript. But we want to highlight that an exhaustive discussion of all related uncertainties that go along with all the observations (soil moisture, discharge, rainfall, soil water retention functions, sap flow, ERT measurements, etc.) we use might be out of scope.

If our sap flow observations were trustworthy, they cannot be directly compared to PM simulations results, as the former is a velocity and the latter is a normalized flow. This is in fact why we a) normalized both observed sap flow and ET by dividing their values by their range and do in fact only discuss the correlation among the normalized values. We still think that this comparison provides added value because it yields the models deficiencies and capabilities to match sap flow dynamics, and whether the maximum and minimum values coincide. We will better explain this in the revised manuscript. Furthermore we will update Figure 10 in the manuscript and show the ensemble of all 61 sap flow observation similar to our soil moisture plots.

**Reviewer 37) Lines 691-692:** *You might be careful in projecting your own expectations, surprises, and uncertainties onto your readers. The result you describe here is not counterintuitive to me; it's exactly what I would expect, for the exactly the reason you state. Higher gradient = more rapid drainage = less persistent storage (assuming all else is equal, which is what you've assumed for this virtual experiment)*

**RL**: We partly disagree with the reviewer. When presenting our results at the latest EGU conference, people were wondering if our hillslope model would also work acceptable if we change the hillslope topography. In fact this is not the case, as we show in the virtual experiment number 1. We will avoid the term astonishing and remove this part from our revised manuscript since it is a virtual experiment.

**Reviewer 39) Table 2**: *Please provide some rationale for why you use multiple error metrics (e.g. NSE, KGE, logNSE) instead of just one. It's just confusing to the reader when you talk about quality of results in one case using KGE, and in another case using NSE. Also, the different metrics show different sensitivities to the domain-changes utilized in the virtual experiments. Why? Which one is most appropriate in light of those differences? You're using these various metrics to make inference about the relative importance of different model features, so you need to argue why one or the other metric is better. Or just use one metric for clarity.*

**RL:** We believe that it is good practice to calculate multiple error metrics when showing model results. Every error metric or objective function has its advantages and disadvantages. There is a long discussion about this in hydrology (e.g. Kling and Gupta, 2009; Schaefli and Gupta, 2007). In the revised manuscript

we will shortly explain the advantages and different sensitivities of the different metrics and stress why we use different ones (see also Comment 32).

*Reviewer 44) Lines 780-781: Why is that remarkable? It's a predominantly upland catchment with forested hillslopes. Was it your initial expectation (null hypothesis) that the model would be incapable of simulating streamflow?*

**RL:** At least we found it remarkable that a catchment of 20 km$^2$ can be represented using a single hillslope (at least to some degree).

Generally, it is an interesting point why we were surprised. We used a physically-based hillslope model and parameterize the model with measured data when possible and take values from other studies and regions when we had no measurements. By doing so we are able to simulate the water balance and the streamflow of two lower mesoscale catchments to a certain extent. It is always subjective if and when a model simulation performs well and when it doesn't, especially considering that hydrological modelling studies are often rather data mining approaches that show only tables of objective functions to demonstrate that their model is working well. But keeping the numerous studies in hydrology in mind which state that a hydrological model cannot be parameterized by measurements, our results were at least for us surprising. We were also surprised that our model can mimic the dynamic of the sap flow observations even if they are not too accurate and our evapotranspiration routine is not state of the art. And yes, we were also surprised that our soil moisture simulations are within the margin of observation and not too far off the 12-hour rolling median of the soil moisture observations.

We are sorry for using the word 'surprised' but we had issues to find papers where the authors tested their models in such an extensive data driven way? We had the feeling that the catchment hydrology community agreed that we cannot use measurements to set up physically-based models and actually use them because of the well documented limitations.

*Reviewer 47) Lines 802-804: Are you so sure this can be concluded? It seems to me that if you want to advocate the use of highly parameterized, spatially-distributed models for the sake of learning about catchments, you need to illustrate that the model is accurately representing some of the spatial dynamics in the hillslope (or catchment that is a composite of your hillslopes). In those cases the matching between simulated and observed averages is not that great for soil moisture there are systematic errors in all cases (Figure 9A-D). By comparing average-simulated soil moisture for the whole hillslope to the average-observed soil moisture for the whole catchment, you're failing to rigorously test the spatially-explicit predictions made possible by the model. You should try to show that the model actually properly represents spatial variability in soil moisture, saturated-zone expansion, hydraulic gradients, etc. If not, then it's hard to argue that the distributed model teaches us anything more than we would learn from a lumped model.*

**RL:** We agree that this needs to be further specified. In fact we "only" showed that the model works well for runoff. The setup can obviously be improved to better reproduce soil moisture dynamics, by perturbing for instance porosity to match the observed variability of soil moisture. We will revise the conclusions accordingly and carry out a more spatially distributed comparison of simulated and observed soil moisture.

*Reviewer 48) Lines 812-816: I fully agree with this statement. You don't need a high-dimensional, spatially-distributed model if all you want to do is predict runoff at an annual timescale, or even at shorter time scales. Use a transfer function, maybe even a time variable transfer function you will still have vastly fewer degrees of freedom than in the spatially-distributed Richards equation model. But doesn't this statement contradict the overall message of your paper, that those more complex models are needed for learning about catchment functioning?*

**RL:** No, again we believe that a successful simulation of the catchment functioning does not end with successful simulation of the runoff. It is quite difficult to estimate soil moisture and evapotranspiration with a transfer function. The second point is that we use an a priori model setup that was not calibrated automatically, but based on several observations which are independent of discharge. Particularly the latter is not possible with a transfer function approach, which can only be calibrated using discharge data. But again you are right our language is not precise and we need to clarify this in the revised manuscript.

*Reviewer 49): The spatial variability of soil-hydraulic properties may be quite important, in fact, for properly simulating all those runoff peaks in the summer, where your model does quite poorly (Figure 8B,D, and Figure 12B)*

**RL:** This might be the case. But it could also be that steep runoff peaks are generated by forest roads and paved areas in the catchment. Furthermore, most of the runoff is produced in winter. That means, around 90% of the overall runoff within a hydrological year is not primarily controlled by soil heterogeneity of soil-hydraulic properties since our model does well with respect to runoff in winter. Please note that we do not consider spatial variability of soil properties unimportant. We simply want to stress that the proposed approach to define representative soil hydraulic functions works acceptable. We will rewrite this in a revised manuscript.

*Reviewer 50) Line 823-844: I would suggest you delete all of this. It seems wildly speculative and I have no idea how, based on the analyses performed in this paper, you conclude that equifinality and the concept of a representative hillslope is rather more a blessing than a curse since there is an infinite number of possible macropore setups which yield the same runoff characteristics. If this were not the case, we could not transfer macropore setups from the literature across system borders and successfully simulate two distinct runoff regimes which are strongly influenced by preferential flow." An infinite*

*number of macropore setups that yield the same runoff characteristic. What are you talking about here? You tested 2 such scenarios (Figure 3C, D). Your model does a fairly poor job at matching runoff peaks at many times of year. Those peaks are the hydrological attribute most likely to be influenced by the activation or latency of preferential flow paths. When you say you that you "successfully simulate two runoff regimes" I presume you are talking about the double-mass curves, because your simulated hydrographs show significant errors at many times of the years.*

**RL:** The reviewer is right that the section is not entirely supported by the evidence provided in this paper and will be removed. However, the above mentioned studies by Klaus and Zehe (2010) and Wienhöfer and Zehe (2014) show that more than a single macropore network architecture is capable to yield the same simulate runoff (Equifinality). This is because the total amount of fast subsurface flow is jointly determined by the macropore density and their hydraulic conductance. Note that Klaus and Zehe (2010) varied both macropore density and their conductance within the range of observed values, and found 13 out of 420 setups that simulated tile train discharge in the same manner. This is of course not an infinite number.

The points we wanted to stress here is that the macroporous medium and the setup Wienhöfer and Zehe (2014) used in their study in Austria also improved the model performance significantly in the Wollefsbach and in fact also in the Colpach catchment. We think that the fact that several parameterizations of the macropore network work equally well is maybe an advantage rather than a problem. Simply because we cannot measure the real macropore network in the catchment nor could we map the real setup in our model or any model. That's what we mean when we write that equifinality is rather a blessing than a curse in the case of physically-based models.

*Reviewer 51) Line 864-866:* *Do you mean, "below which", instead of "above which"? I wouldn't spend much time on that. By changing slope and nothing else, you're vastly oversimplifying how soils, geology, and geomorphology affect streamflow, and how all those variables are related to topography in naturally evolving landscapes.*

**RL:** In the virtual experiment we doubled the gradient, in fact we picked on of the steepest slopes, with no effect. When changing the gradient to slopes much less steep, we expect that this will have an effect, because water flows slower through the fast lateral flow path. Hence we expect a threshold below which a reduction of the gradient starts to matter. This will not be part of a revised manuscript.

*Reviewer 52) Lines 888-889: How do you justify this statement? Did all of your virtual experiments where you manipulate the bedrock topography have an equal volume of depressions? And if so, how do you go about quantifying the volume of depressions in an undulating rock surface? What constitutes a depression or a high point, versus a portion of the rock surface that is part of a datum plane?*

**RL:** The reference slope, the slope without bedrock interface and the slope with the riparian zone (VE 2.3) had the same volume of depressions. The number of depressions is equal to the number of local

maxima in the bedrock topography. The depression volume is then simply counting the number of grid cells upslope a local minimum with an elevation above the local bedrock interface but below the next downslope maximum. As the reference hillslope and the VE2.3 have the storage volume (Figure 9), but differ with respect to its distribution, we think this statement is supported. This part will be removed in a revised manuscript.

[Figure]

**Figure 9 Sketch of the reference hillslope and of the conceptualized hillslope with the barrier (no flow area) at the left hillslope border.**

*Reviewer 53) Lines 867 and 895: You're using the questions as section headers, but the subsequent content does not answer the questions. First of all, define what you mean by "first order control". Do you just mean that the response variable (annual runoff ratio, or other?) is a linear function of the independent variable (bedrock topography, or vegetation)? If so, do your data corroborate such a linear relationship? I'm not sure you can say, based on the limited scenarios of bedrock topography you tried. You would have to come up with some quantitative metric distinguishing one bedrock scenario from another. In terms of vegetation, all you did was try 2 different times for bud break. Can you discern a linear relationship between "vegetation" and some response variable based on these tests?*

**RL:** The term first-order control is often used to specify the most sensitive parameter or information source – or which of the parameter or information sources contributes most to the explained variance and thus to an error function. Simulations without macropores and the fast bedrock interface (using the same bedrock topographie) reduces the KGE by 0.12, while an additional removal of the bedrock topography reduces the KGE to 0.59. This supports that bedrock topography is a first-order control in

respect to the runoff formation. We will leave this out in a revised manuscript since it belongs to the virtual experiment.

*Reviewer: 54) Lines 912-923: Of course you can! You can setup heterogeneous rainfall inputs at the soil surface in your model domain. You can setup different scenarios of incoming solar radiation along the hillslope domain to emulate aspect-related differences in the radiation balance, water budget, and possibly soil hydraulic/or geological characteristics. I'm really struggling to understand how you continually advocate for spatially distributed models, but continually state that one can't expect them to accurately represent spatially-explicit hydrological processes. If you don't expect a spatially distributed model to accurately represent spatially-explicit hydrological processes, then why use it, instead of a lumped model?*

**RL:** Here we discuss the limitations of using a single hillslope for a catchment which consists of numerous hillslopes. Much confusion arises from our fuzzy use of the term "distributed" and "single hillslope". The representative hillslope is distributed along the slope line. Of course we can add distributed rainfall to the hillslope and allow for perturbations of soil parameters. However, the characteristic length of rainfall variability in the Colpach is larger than the total extent of our hillslope, similarly the variability of radiation depends not only on slope but also on aspect and landuse. A representation of the full sources of variability requires a model setup consisting of all the hillslopes in the catchment and their interconnecting river network. Catflow allows for this, as for instance shown in Zehe et al., (2001), but this was not the goal in our study here. We will better explain this part in the revised manuscript.

The reviewer is generally right that we waste too much effort on discussing the limitations of the Richards approach. We will change this tenor in the revised manuscript.

*Rewiewer 55) Lines 937-969: I would suggest deleting this to shorten and focus your discussion. There is not much reference to your analysis in this section, it's a little bit of a ramble, and you don't say anything about the Richards equations that hasn't already been said many times before over the last 70-80 years.*

**RL:** Good point, we will remove most of this part, but keep the part on how to deal with emergent soil structures.

*Reviewer 56) Lines 971-993: I recommend you remove this from the manuscript. You're pontificating about all the ways the land surface models must be fundamentally improved for hydrological modeling, and in doing so you're demonstrating that you have no awareness of the related disciplines of hydrometeorology, plant physiology, and biophysics. All the phenomena that you imply are important, for example, "implies that phenology evolves in response to climate and hydrological controls, thereby creating feedbacks" are in fact known to be important by people in those fields, and others. The upgrades to our model representations that you suggest should happen, have in fact happened, and continue to be upgraded, for example, models that link plant metabolism and water use, or that utilize spatially- and*

*temporally-dynamic root uptake schemes. You conclude by saying that the literature is full of more realistic models for parameterizing stomatal conductance, but you still use a fairly standard version of the PM model with non-local parameters. So I don't think your analyses are very relevant to the state of practice in evapotranspiration modeling. I think you should stay focused on the representation of runoff processes.*

**RL:** We will remove this passage. The reviewer is right, that there are much more sophisticated approaches for ET available.

**Reference**

Angermann, L., Jackisch, C., Allroggen, N., Sprenger, M., Zehe, E., Tronicke, J., Weiler, M., Blume, T., 2016. In situ investigation of rapid subsurface flow: Temporal dynamics and catchment-scale implication. Hydrol. Earth Syst. Sci. Discuss. 2016, 1–34. doi:10.5194/hess-2016-189

Bos, R. van den, Hoffmann, L., Juilleret, J., Matgen, P., Pfister, L., 1996. Conceptual modelling of individual HRU ' s as a trade-off between bottom-up and top-down modelling , a case study ., in: Conf. Environmental Modelling and Software. Proc. 3rd Biennal Meeting of the International Environmental Modelling and Software Society. Vermont, USA.

Breuer, L., Eckhardt, K., Frede, H.-G., 2003. Plant parameter values for models in temperate climates. Ecol. Modell. 169, 237–293. doi:10.1016/S0304-3800(03)00274-6

Dooge, J., 1986. Looking for hydrologic laws. Water Resour. Res. 22, 46S–58S. doi:10.1029/WR022i09Sp0046S

Gupta, H. V., Clark, M.P., Vrugt, J. a., Abramowitz, G., Ye, M., 2012. Towards a comprehensive assessment of model structural adequacy. Water Resour. Res. 48, 1–16. doi:10.1029/2011WR011044

Jackisch, C., Angermann, L., Allroggen, N., Sprenger, M., Blume, T., Weiler, M., Tronicke, J., Zehe, E., 2016. In situ investigation of rapid subsurface flow: Identification of relevant spatial structures beyond heterogeneity. Hydrol. Earth Syst. Sci. Discuss. 2016, 1–32. doi:10.5194/hess-2016-190

Klaus, J., Zehe, E., 2011. A novel explicit approach to model bromide and pesticide transport in connected soil structures. Hydrol. Earth Syst. Sci. 15, 2127–2144. doi:10.5194/hess-15-2127-2011

Klaus, J., Zehe, E., 2010. Modelling rapid flow response of a tile-drained field site using a 2D physically based model: assessment of "equifinal" model setups. Hydrol. Process. 24, 1595–1609. doi:10.1002/hyp.7687

Kling, H., Gupta, H., 2009. On the development of regionalization relationships for lumped watershed models: The impact of ignoring sub-basin scale variability. J. Hydrol. 373, 337–351. doi:10.1016/j.jhydrol.2009.04.031

Martínez-Carreras, N., Krein, A., Gallart, F., Iffly, J.-F., Hissler, C., Pfister, L., Hoffmann, L., Owens, P.N., 2012. The Influence of Sediment Sources and Hydrologic Events on the Nutrient and Metal Content of Fine-Grained Sediments (Attert River Basin, Luxembourg). Water, Air, Soil Pollut. 223, 5685–5705. doi:10.1007/s11270-012-1307-1

Menzel, A., Jakobi, G., Ahas, R., Scheifinger, H., Estrella, N., 2003. Variations of the climatological growing season (1951-2000) in Germany compared with other countries. Int. J. Climatol. 23, 793–812. doi:10.1002/joc.915

Schaefli, B., Gupta, H. V, 2007. Do Nash values have value? Hydrol. Process. 21, 2075–2080.

Wienhöfer, J., Zehe, E., 2014. Predicting subsurface stormflow response of a forested hillslope – the role of connected flow paths. Hydrol. Earth Syst. Sci. 18, 121–138. doi:10.5194/hess-18-121-2014

Wrede, S., Fenicia, F., Martínez-Carreras, N., Juilleret, J., Hissler, C., Krein, A., Savenije, H.H.G., Uhlenbrook, S., Kavetski, D., Pfister, L., 2015. Towards more systematic perceptual model development: a case study using 3 Luxembourgish catchments. Hydrol. Process. 29, 2731–2750. doi:10.1002/hyp.10393

Zehe, E., Blöschl, G., 2004. Predictability of hydrologic response at the plot and catchment scales: Role of initial conditions. Water Resour. Res. 40, 1–21. doi:10.1029/2003WR002869

Zehe, E., Ehret, U., Pfister, L., Blume, T., Schröder, B., Westhoff, M., Jackisch, C., Schymanski, S.J., Weiler, M., Schulz, K., Allroggen, N., Tronicke, J., Dietrich, P., Scherer, U., Eccard, J., Wulfmeyer, V., Kleidon, A., 2014. HESS Opinions: Functional units: a novel framework to explore the link between spatial organization and hydrological functioning of intermediate scale catchments. Hydrol. Earth Syst. Sci. Discuss. 11, 3249–3313. doi:10.5194/hessd-11-3249-2014

Zehe, E., Graeff, T., Morgner, M., Bauer, A., Bronstert, A., 2010. Plot and field scale soil moisture dynamics and subsurface wetness control on runoff generation in a headwater in the Ore Mountains. Hydrol. Earth Syst. Sci. 14, 873–889. doi:10.5194/hess-14-873-2010

Zehe, E., Maurer, T., Ihringer, J., Plate, E., 2001. Modeling water flow and mass transport in a loess catchment. Phys. Chem. Earth, Part B Hydrol. Ocean. Atmos. 26, 487–507. doi:10.1016/S1464-1909(01)00041-7

---

## Author Comment (AC3) · 2 Sep 2016

Reply to Anonymous Referee #2:

**Ralf Loritz (RL):** We would like to thank the anonymous reviewer #2 for his comments and the time he invested in assessing our manuscript. She/he has addressed many issues which will greatly improve the readability of our manuscript.

***Reviewer: Summary and Recommendation:***

***Reviewer:*** *The manuscript presents 2D distributed physically-based modeling of water flow dynamics at two representative hillslopes differing in geology, soil, and vegetation characteristics. Authors proposed several scenarios with different conceptualizations of hillslope to analyze first-order controls on soil water dynamics and limitations in current modeling approaches. The manuscript conveys an interesting topic, potentially attracting readers of wide hydrological community. However, several points deserve further attention and need to be clarified/improved.*

***General comments***

***Reviewer:*** *1. Rather than a rigorous study on hillslope modeling with detailed data-model comparison, authors set up representative hillslopes built on two perceptual models. As a result, no comparison of spatially dependent variables (although measured) was presented. Authors avoided intentionally the first step in modeling. The idea of using the state-of-the-art distributed model for analyzing soil water dynamics in a simplistic representative hillslope segment seems awkward as full potential of the model is not exploited.*

**RL**: We are thankful for this comment and will better explain our point of view in the revised manuscript. We are aware that distributed model studies often do a point to point comparison between observed and simulated fluxes and state variables in a catchment (Ebel et al., 2008; Scudeler et al., 2016; VanderKwaak and Loague, 2001). While this might be appropriate in less heterogeneous systems or in environmental system simulators as Biosphere 2 LEO, we regard this as rather difficult in more heterogeneous environments. In fact we think we can only be successful in doing point to point comparisons of simulated and observed state variables if the model is parameterized on highly resolved exhaustive observations of soil hydraulic parameters, rainfall and much more. As such observations are not at hand, we prefer comparison of statistical moments, based on the hypothesis that for instance observed and simulated soil moisture belong to the same ensemble and that they share similar dynamics. Also we started with what you claim the "*first step in modeling*" and did a comparison between soil moisture observations and soil moisture simulations at similar positions at a hillslope. Unfortunately this was not too conclusive. As can be seen in Figure 1 and 2 the variability of our measurements and observations cannot be grouped simply by their position in the landscape.

We do agree with reviewer #1 and #2 that we need to step beyond comparing medians of a single layer in our hillslope model to observations in the same depth. In the revised manuscript we will compare spatial variance of observed and simulated soil moisture observations using virtual observations. This virtual observations are located at different lateral positions on the hillslope in the same vertical depth (10 and 50cm) as the observations (For details see reply to reviewer 1).

[Figure]

**Figure 1 Soil mositure observation from downslope positions (left) and from midslope positions (right) at the hillslope.**

[Figure]

**Figure 2 Fitted soil water retention curves (orange) and measured soil water retention relationships with a color key for their measurement depth (two upper plots) and for their distance to the stream (to lower plots) for the Colpach and Wollefsbach catchment.**

*Reviewer: 2. The hydraulic functioning of the simulated hillslope segment is questioned. Given the vertical height of the 2D flow domain (2 m), maximum soil depth of 1.8 m, and thickness of the soil-bedrock interface (0.2 m), bedrock was excluded at some locations. This hydraulic setting affects deep percolation fluxes across the interface. Furthermore, saturated hydraulic conductivities are too high for*

*soil layer, soil-bedrock interface, and drainage system and too low for bedrock. Can still be laminar flow assumed for near saturated conditions (Richards eq.)? Is there any measurement indicating such high Ks value of the soil-bedrock interface? Can Ks value of the soil-bedrock interface be higher than upper soil? Similarly, bedrock porosity values seem too high, is there any (experimental) justification? Grid size of 1 m used for macropores seems unrealistic.*

**RL:** Thank you for this comment. In the following section we will explain the unclear parts of our parametrization in more detail. In a revised manuscript we will rephrase the parts which reviewer #1 and #2 found not conclusive. It is our goal to make our modelling as transparent as possible.

*Bedrock parametrization:*

Maximum soil depth with the bedrock interface was 1.8 m. Hence bedrock was not excluded at any location in our model domain. We apologize and we will stress this in a revised manuscript.

In both catchments we expect no major groundwater body beneath the hillslopes (Wrede et al., 2015). In the Wollefsbach catchment you can find redoximorphic features in deeper soil layers indicating a local episodic groundwater body above an impermeable bedrock. Also the schist bedrock of the Colpach catchment is besides of cracks and fractures described as quasi impermeable. We therefore think a Ksat of $10^{-9}$ m/s can be justified in both catchments.

*Hydraulic conductivity of the soils:*

Yes, the hydraulic conductivities of the young silty soils are high. These values are corroborated based on a large set of observation using both undisturbed soil samples and constant head permeameter measurements (Jackisch et al., 2016). Furthermore Wienhöfer et al. (2009) reports partly similar high values for a clay loam in a forested slope in the Austrian Alps. These high values of Ks and porosity might arise in case the fine silty material is aggregated with large inter-aggregate pores.

Bedrock interface velocities:

Yes, subsurface lateral flow can exceed Ksat values of the topsoil. Wienhöfer et al. (2009) derived these conductance estimates particularly for the lateral pipes based on a series for two tracer experiments in a forested site. He observed breakthrough velocities of $2 \times 10^{-2}$ m/s and average travel velocities of $10^{-3}$ m/s for a lateral transport distance of order of 30 m. These high velocities coincided with a Peclet number of 2, which indicates that these fast reactions were mainly due to lateral preferential flow. Top soil Ks values in this area where on average $10^{-5}$ m/s. Furthermore Angermann et al. (2016) report breakthrough velocities of $10^{-3}$ m/s in the periglacial deposits which are on top of the schist bedrock in the Colpach.

*Laminar flow assumption:*

We agree with the reviewer that, with a Ks value of $5 \times 10^{-3}$ m/s, a vertical grid size of 0.1 and a porosity of 0.25 we have a Reynolds number of 50. This is slightly above the threshold for strict laminar flow, which is 10 according to Bear (1972). This implies that strong fluctuations might cause turbulence, which, implies that we need a different flux law. However, we are still below the threshold of 1000, were flow is

entirely turbulent. Nevertheless in a revised manuscript we will reduce Ks to 1x10$^{-3}$m/s, to stay below the critical Reynolds number of 10.

Macropores:

We agree that a grid size of 1 m seem too coarse for the macropores. For a grid size of 1 m the Poisson process does often generate several macropores in the same grid cell, the generation process is carried out at a much smaller grid (1 -2cm). All contribute to the enhanced conductance of the model element, the enlarged conductance is hence an effective representation of the subscale macropores. This infiltrability is sufficient to allow a fast flow to the bedrock (in the Colpach) anddrainage (in the Wollefsbach) and then subsequent fast lateral flow. This is shown to be sufficient for runoff generation, but it has certainly deficiencies when dealing with solute transport. To improve this we usually work with an adaptive grid size of a few centimeters, generate the macropores, and reduce the grid resolution in areas without macropores. This was done in the Wollefsbach. We already employed the same approach in the Colpach, and simulated the hydrological year of 2014, the result is the same as with the coarser grid, as shown in Figure 3 and 4 in the reply to the first reviewer. In a revised manuscript we will reduce the grid size of the macropores in the Colpach to 0.1 m similar to the grid size in the Wollefsbach catchment.

*Reviewer: 3. Runoff processes (subsurface lateral flow, overland flow, and deep percolation) and their partitioning mostly depend on hydraulic setup of the hillslope (see point 2). Observed catchment streamflow was compared with the sum of the three runoff component. It would be interesting to see individual contribution of each process to streamflow.*

**RL:** We agree with the reviewer and will analyze and display the three components separately in a revised manuscript.

*Reviewer: 4. Some details of the model were not unveiled in the manuscript. For instance, algorithm of the root water uptake module remains unclear as well as the parameters (not sufficient to refer to previous study). As one hydrological year was considered in the simulations and most of runoff occurred during the winter season (Figure 8), no information was given on snowmelt runoff. More comments are appended below.*

**RL:** We apologize that this was not clearly written in our discussion paper. We will make our ET simulations in a revised manuscript as transparent as possible and rework the description. For further details we refer to our reply to the first review.

*Reviewer: 5. The manuscript is too long. Some parts can be shorten and condensed (see Detailed comments). Please focus more on hydrology and less on philosophy. Conclusions part resembles Discussion.*

*6. Discussion should be condensed, there are too many points which are discussed. The results obtained in this study are not related to previous literature, i.e. no deep discussion is provided. Focus on the main aspects of the results. Some discussion parts are too vague and superficial, adding limited value to*

*objectives ("to identify limits in our theories and related physically-based models") and overall knowledge.*

**RL:** We will streamline our revised manuscript by removing most of the virtual experiments, shorten our introduction and be more precise in our discussion. Our goal was to show that perceptual models can be used to constrain physically based hillslope models in a qualitative manner. Moreover, it was our objective to show that such a modeling approach can be based primarily on field measurements and literature values and therefore we intentionally avoided tuning of our model parameters against a single hydrological response. Finally we wanted to show which is the most important information or data source that is needed for setting up our hillslope models.

We apologize that we lost track at some of the sections of our discussion paper and think that with the comments and references of reviewer #1 and reviewer #2 we are able to be more concise and improve our manuscript.

***Detailed comments***

***Reviewer:*** *Lines 14-27: This is too long introduction in the abstract. Please shorten.*

**RL:** We will consider shortening the abstract.

***Reviewer:*** *L16-20: Conceptual models can also be physically-based. The use of conceptual models is incorrect.*

**RL:** We agree with the reviewer. In fact we adopted the term physically-based, as it is widely used for models based on the Richards equation. We will add a note to the introduction to clarify this.

***Reviewer:*** *L34-6: Not true, internal water storage was not simulated well (see Figure 9).*

**RL:** We did not use the term "well" in line 34 but "some success" and we think this is true. We agree that the model performance could be improved by further tuning, but we do not think this is a real surprise neither was this our goal. Our objective was to set up our model based on field measurements where it is possible and on literature where it isn't. Relating to our issue we think it is legitimate to write we had some success in simulating soil moisture observations as well as sap flow dynamics.

***Reviewer:*** *L79-85: The literature body of recent hillslope 1D&2D modeling applications is somehow limited to a narrow window (mostly of the author's group). For instance, predictions of 2D Richards-based hillslope model with a provision for preferential flow were compared with field data (hillslope discharge and spatially distributed pressure heads) in recent studies.*

**RL:** Thank you for that advice, we did not leave out studies on purpose. We will carefully check the literature once more to refer to other relevant studies.

***Reviewer:*** *L110-3: Yes, hillslopes are indeed important in some headwater catchments. However, wetlands/riparian zones may control the runoff generation in other headwater catchments.*

**RL:** You are right, with our model we are not able to check how important wetlands are for the runoff generation. But we would like to highlight that there is also a saturated area at hillslope foot as a result of our bedrock topography. We further think that the riparian zone is still is a part of a hillslope.

***Reviewer:*** *L181-4: A reference is needed here to support the statement on water balance. Introduction section seems to be quite long, it can be effectively shorten without losing the central messages.*

**RL:** Thank you, we will shorten the introduction. Especially the first passage and the part dealing with ET.

***Reviewer:*** *L311-4: Double mass curve is not well suited for studying annual water balance since it relates runoff to precipitation. There is no provision for e.g. storage changes.*

**RL:** We think it is a suitable measure for detecting difference in seasonal runoff behavior. This is corroborated in the work of Seibert et al. (2016) and Jackisch (2015). We agree that strong inter-annual variations of runoff coefficient can be either a result of storage changes or due to changes in ET. The double mass curve cannot be used to discriminate these reasons. However, it is well suited to detect the shift between the summer and the winter regime. Furthermore do we not expect large groundwater bodies in both catchments. Hence inter annual storage changes shouldn't be of high relevance. We will explain this in the revised manuscript.

*Reviewer: L439-42: Any experimental justification for arrangement of the structures? Taken from previous studies on different hillslopes?*

**RL:** Several field experiments related to macropore flow took place in the Colpach as well as in the Wollefsbach (Angermann et al., 2016; Jackisch et al., 2016). We apologize as we did not do a good job in describing how we parametrized our model with respect to the macropore setup. We will make the parametrization of the macropores simpler and more transparent in a revised manuscript (for details see reply to reviewer 1).

*Reviewer: L453-4: Grid size of 1 m seems as a crude approximation for macropores.*

**RL:** We agree and will change this to 10 cm (See reply to reviewer 1 and above).

*Reviewer: L472-3: Ohm\*m. In Figure 6B, contour line of 1500 ohm\*m is situated in depths ranging from 1.0 m to 3.2 m for 100 m hillslope length. How such spatial configuration could be simulated in 2 m high hillslope segment? Please use the same hillslope segment in Figure 3 as ERT cross-section shown in Figure 6B.*

**RL:** Good point this was not explained well. Rather than directly transferring the ERT bedrock topography into our hillslope model we divided the deepest bedrock position and transferred it relatively to our 2 m deep and 350 m long hillslope. We chose a 2 m deep hillslope after analyzing the mean depth distribution of the 1500 ohm\*m bedrock surface of 7 ERT measurements in the catchment. Our goal was not to have an exact image of the ERT measurement we show in Figure 6B but to generate a bedrock topography which follows the image of the perceptual model. We will better explain this in the revised manuscript.

*Reviewer: L474-7: Model of van Genuchten assumes zero air-entry value. Thus, alpha parameter is not reciprocal to air entry value. Not clear explanation of the soil hydraulic parameters - list also the parameters of macropores in Table 1. Furthermore, show the (variable) depths of the soil structures in Table 1.*

**RL:** We apologize for this. We will correct this in the revised manuscript.

*Reviewer: L501-3: Be more specific, what kind of data?*

**RL:** We will add a table with the vegetation parameters in a revised manuscript.

*Reviewer: L508-9: Pressure-water content relationship was measured in 0-3 pF range in detail, resulting in large spread of data points. This is not true for smaller pressure head values. These aspects may invalidate the fitted representative curves used in modeling.*

**RL:** It is correct that the soil water retention was measured with two different methods for 0-3 pF and above respectively. The Hyprop apparatus (for the low tension range) allows for a continuous measurement resulting in a large number of data points. The WP4C apparatus can only measure matric potential at discrete states and resulted in 2-5 points for each sample. For the calculation of the representative curve (as also for the fitting of individual retention curves to the data), the pF range is binned into steps of 0.05 pF. The resulting curve is fitted to the respective mean of each bin (when data is available).

This method reduces the effect of different data point densities. As we use all records of the grouped soil samples, there is a very strong basis for such a fitting also in the higher tension range. Hence the concern of the reviewer may not justify. Notwithstanding, there may be alternative means to derive such a representative retention curve.

To clarify this, we will change the used formulation L364ff:

"For both geological settings we estimated a mean soil retention curve by fitting a van Genuchten-Mualem model to all recorded retention data points of all soil samples in each group (51 and 28, respectively). This was done as maximum likelihood method accounting for the different data point density coming from the two measurement techniques. The tension axis was binned in 0.05 pF increments. The resulting curve is fitted to the respective mean soil water value of each bin (Table 1 and Figure 7)."

*Reviewer: L585-7: Does this mean that value of the saturated hydraulic conductivity was increased 75 times compared to reference scenario? Reference scenario used already high Ks value.*

**RL:** The point behind this virtual experiment was to show that emergent structures observed in the catchment have a strong influence on the runoff generation. With our static parametrization we unnecessarily constrain the agility of our model (Mendoza et al., 2015) and are not able to simulate the runoff generation in the Wollefsbach in summer. Instead of increasing the Ks value we should have added extra vertical macropores to our hillslope model. But as you point out above the strong increase of Ks actually violates the laminar flow assumption of the Darcy Richards equation. We admit that this virtual experiment was not done in a smart way and we apologize. We will remove this from the revised manuscript.

*Reviewer: L602-5: This can't be concluded, so far only runoff component of hillslope balance was shown. No comparison on hillslope storage was made.*

**RL:** We will revise this and state that the hillslope models closely portray the seasonal pattern of the catchment's runoff production, also because simulated and observed annual runoff coefficients match very well.

*Reviewer: L607-9: I would not say that in case of Wollefsbach, please see the winter period in Figure 8D. To support this statement, provide efficiency coefficients for both winter and summer period.*

**RL:** Thank you for this good comment we will do this in the revised manuscript.

*Reviewer: L610-2: Is hydrological year 2013 meant here? I miss the point. A1 scenario for Colpach has NSE = 0.84, so why discuss smaller NSE? Instead, scenario A2 for Wollefsbach requires discussion (NSE = 0.26).*

**RL:** Yes, we also tested the model for the hydrological year 2013. Since we only have runoff and rainfall for the year 2013 a more detailed test is not possible. Our goal was to show that our model also work in a different time period (Split sample test (Klemeš, 1986)).

*Reviewer: L613-5: Not clear what is discussed, Weirbach simulation?*

**RL:** We are sorry. We meant the Weierbach catchment which is a headwater catchment in the Attert basin. Our goal was to show that our model also works in a different catchment in the same hydrological landscape (Proxy-basin test (Klemeš, 1986)). We will clarify this in the revised manuscript.

*Reviewer: L624-5: Infiltration-excess overland flow was most likely not simulated due to extremely high Ks value of (top)soil. Was the extent of overland flow decreased by further increasing Ks value of bulk soil (emergent structures)? Such short high flow events also could not produce saturation-excess overland flow. Please check.*

**RL:** It is not the Ks value that is of relevance here but Ku(theta). The latter is in the order of $10^{-11}$ m/s in summer as specified in our reply to the reviewer 1. In Catflow we usually treat infiltration and runoff generation in a mixed or Cauchy boundary condition in combination with a fine discretization of the top soil layer (< 5cm). Rainfall is treated as flux boundary condition, until the upper element gets saturated within a time step. We then switch to a Dirichlet boundary condition while using the overland flow depth as pressure boundary. Infiltrating water flux is calculated according to the Darcy law: the product of the finite difference approximation of the potential gradient and the averaged hydraulic conductivity $k_{1+1/2}$ between the two upper nodes. When using the geometric mean sqrt(ksat*k($\theta_2$)) and a ksat of 1e-4 m/s and k($\theta_2$)= 1e-11 m/s the averaged conductivity is $k_{1+1/2}$ = 0.5 e-7 m/s. In fact it is the conductivity of the second node which acts as bottle neck and may produce infiltration excess.

*Reviewer: L666-8.: This suggests misrepresentation of the soil profile (e.g., soil layering) as well as misparametrization of the soil hydraulic properties.*

**RL:** Since the model works acceptable in winter we think it is because we do not account for emergent structures like worm borrows and cracks. If we would have misparametrized our model why should it work in winter for soil moisture?

*Reviewer: L670: This paragraph needs a reference to Figure 10A. Evapotranspiration module is treated as a black-box for readers. We are left unaware what parameters were changed. Was there a difference between potential and actual ET fluxes? Did any water stress occur? Add information on Wollefsbach hillslope.*

**RL:** We are again sorry. We will make our ET simulation more transparent.

*Reviewer: L690-2: Do not write "it may be", this must be exhaustively explained by the model. Less storage above the interface for steeper hillslope setting would lead to increased storage in bedrock. Note that deep percolation was considered when comparison with observed streamflow was performed. This would also cause a delay in simulated runoff compared to reference scenario.*

**RL:** Sorry for using the term "it may be". We will rephrase this. The bedrock is quasi impermeable and why should a steeper hillslope increase the storage in the bedrock? Deep percolation is close to zero.

*Reviewer: L788-90: Satisfying match was obtained due to large measurement variability. Many scenarios with different parameter sets would fall within measured soil water content range (even different modeling approaches can provide similar match).*

**RL:** We refer here to the 12-hour rolling median not to the ensemble of all observations.

*Reviewer: L802-4: Given the comments above, I would hesitate to make such statements.*

**RL:** We think that the statement is well justified in respect to the runoff simulation. But not in the context of the simulation of the soil moisture.

*Reviewer: L810-2: The values contradict the previous statement.*

**RL:** You are right. We are sorry we meant 51.

*Reviewer: L812-5: I do not agree with this statement, see Figure 9.*

**RL:** Our model uses a representative soil water retention curve and does well with respect to discharge simulation in winter. If we now combine this with the fact that 90% of the overall runoff within a hydrological year is produced in winter one might come to the conclusion that heterogeneity of soil-hydraulic properties is not the major control on runoff generation in a hydrological year. Please note that we do not consider spatial variability of soil properties as unimportant. But since three to four parameter models able to simulate the runoff of catchments we think the heterogeneity of the soil-hydraulic properties cannot be of such importance for the runoff generation in a catchment.

*Reviewer: L886-9: It may be also due to location of a large depression (considered at the hillfoot region in this scenario). Therefore, these statements are not fully justified by the results.*

**RL:** You are right. We were too fast with our conclusion. Since this belongs to a virtual experiment we will remove this from the revised manuscript.

*Reviewer: L911: The message of this section remains unclear. What is suggested here? The need to use distributed model of representative hillslope with spatially uniform rainfall and pET fluxes?*

**RL:** We are sorry. The message was that a single hillslope won't be able to explain patterns which are a result of phenomena which have a larger spatial extent than the hillslope itself. To account for them you need a fully spatially distributed model.

*Reviewer:* L936: This section is too vague.

**RL:** This section will be removed from the revised manuscript.

*Reviewer:* L995-7: This can't be concluded in such a general way. Beside hillslopes, riparian zone may play an important role in runoff generation in some catchments.

**RL:** We think a riparian zone is still part of the hillslope. You could easily add a riparian zone to a perceptual model if you think that the riparian zone is a major control in your catchment.

The following comments are rephrased or corrected.  We apologize for these mistakes.

*L166-8: Is there any source to justify this statement?*

*L172-4: Add a few references here.*

*L223-5: Delete "concentration".*

*L268-70: ". . .perceptual. . .".*

*L421-4: Delete "a function of".*

*L424-6: This is not clear, please explain.*

*L428-30: The algorithm of RWU is not clear.*

*L450-3: Define slope angle. Was variable hillslope width considered (total area remains unclear)? The same pertains to Wollefsbach hillslope.*

*L464-6: Boundary conditions are not clearly explained. Does free outflow refer to free drainage BC (with unit hydraulic gradient condition)? Is gravitational flow boundary condition seepage face BC?*

*L486-9: I did not find any band generator in Zehe et al. (2010a).*

*L518-22: Boundary and initial conditions are identical for the two hillslopes. Do not repeat the information.*

*L523: "Model scenarios"*

*L527-31: Not left boundary (Figure 3CD)? What is the added value of log NSE compared to NSE criterion?*

*L551: "In VE2.1 scenario, . . ."*

*L558-9: Instead of "Last not least" use "Finally".*

*L565-8: Need to say what parameters were changed and what were kept unchanged. Otherwise it is a black-box.*

*L581-2: Delete "until the onset of the summer period".*

*L582-3: "Summer period was started with ...".*

*L593-4: This a harsh break from runoff to 2D saturation distribution, I suggest discussing runoff first and then move to saturation.*

*L622-4: Please provide the volume proportions (overland and subsurface flow, deep percolation) of runoff for both hillslopes.*

*L631-5: Not clear ". . . terrestrial filter properties . . .".*

*L708-10: Please make a reference to VE2.3.*

*L732-5: Please reword.*

*L737-41: Delete "revealed and" and "matching".*

*L773-6: Instead of "parallel" use "lateral".*

*L778-80: Instead of "in concert with" use "and".*

*L796-9: Instead of "benchmarking" use "comparison".*

*L828-30: Check KGE values (see Table 2 and Figure 11).*

*L884-6: Is VE2.3 scenario discussed here?*

*L908-9: Not clear what are SVAT modules.*

*L912-3: Not clear ". . . models encountered to capture flashy . . .".*

*L1171-3: Cite this study in HESS.*

*L1272-5: Improve the reference.*

*Figure 3: Add dimensions instead of "small section".*

*Figure 4: Dimensions are necessary.*

*Figure 9: No green color found. Stick to Colpach and Wollefsbach.*

*Figure 10: Relative saturation > 1?*

*Figure 12: Add hillslope location.*

*Table 2: Check KGE values shown in Table 2 and Figure 11.*

Angermann, L., Jackisch, C., Allroggen, N., Sprenger, M., Zehe, E., Tronicke, J., Weiler, M., Blume, T., 2016. In situ investigation of rapid subsurface flow: Temporal dynamics and catchment-scale implication. Hydrol. Earth Syst. Sci. Discuss. 2016, 1–34. doi:10.5194/hess-2016-189

Bear, J., 1972. Dynamics of Fluids in Porous Media.

Ebel, B. a., Loague, K., Montgomery, D.R., Dietrich, W.E., 2008. Physics-based continuous simulation of long-term near-surface hydrologic response for the Coos Bay experimental catchment. Water Resour. Res. 44, 1–23. doi:10.1029/2007WR006442

Jackisch, C., Angermann, L., Allroggen, N., Sprenger, M., Blume, T., Weiler, M., Tronicke, J., Zehe, E., 2016. In situ investigation of rapid subsurface flow: Identification of relevant spatial structures beyond heterogeneity. Hydrol. Earth Syst. Sci. Discuss. 2016, 1–32. doi:10.5194/hess-2016-190

Klemeš, V., 1986. Operational testing of hydrological simulation models. Hydrol. Sci. J. 31, 13–24. doi:10.1080/02626668609491024

Mendoza, P.A., Clark, M.P., Barlage, M., Rajagopalan, B., Samaniego, L., Abramowitz, G., Gupta, H., 2015. Are we unnecessarily constraining the agility of complex process-based models? Water Resour. Res. 51, 716–728. doi:10.1002/2014WR015820

Scudeler, C., Pangle, L., Pasetto, D., Niu, G.-Y., Volkmann, T., Paniconi, C., Putti, M., Troch, P., 2016. Multiresponse modeling of an unsaturated zone isotope tracer experiment at the Landscape Evolution Observatory. Hydrol. Earth Syst. Sci. Discuss. 1–29. doi:10.5194/hess-2016-228

VanderKwaak, J.E., Loague, K., 2001. Hydrologic-Response simulations for the R-5 catchment with a comprehensive physics-based model. Water Resour. Res. 37, 999–1013. doi:10.1029/2000WR900272

Wienhöfer, J., Germer, K., Lindenmaier, F., Färber, A., Zehe, E., 2009. Applied tracers for the observation of subsurface stormflow at the hillslope scale. Hydrol. Earth Syst. Sci. 13, 1145–1161. doi:10.5194/hess-13-1145-2009

Wrede, S., Fenicia, F., Martínez-Carreras, N., Juilleret, J., Hissler, C., Krein, A., Savenije, H.H.G., Uhlenbrook, S., Kavetski, D., Pfister, L., 2015. Towards more systematic perceptual model development: a case study using 3 Luxembourgish catchments. Hydrol. Process. 29, 2731–2750. doi:10.1002/hyp.10393

---

## Referee Report (RR1)

Manuscript HESS-2016-307: Picturing and modelling catchments by representative hillslopes by Loritz et al.

The authors carefully revised the manuscript according to comments and suggestions raised by the two reviewers as well as the editor. The revised version of the manuscript improved significantly compared with the original submission. Nevertheless, I ask the authors to consider the technical details/comments listed below when they elaborate the revision.

Detailed comments

Lines 12-3: Justify the need of this approach at the beginning of the abstract.

L66-70: Diffusion equation for lateral flow has been recently coupled with 1D vertical dual-continuum Richards equation for prediction of preferential hillslope stormflow.

L131-3: Add a few references on the previous applications of this idea.

L193-4: Delete areas in the parentheses, these are presented once again below.

L202-3: Refer to Figure 2A.

L216-7: "The lack of significant observations of base flow …" – this is awkward, rephrase.

L273-4: Was ERT surveying performed also in marl area? Perhaps not, please explain the reason. It would be good to have ERT representation from both areas (hillslopes).

L298-300: Show the stations in Figure 1.

L325-6: Improve English.

L343-5: Delete "along the soil surface".

L374-5: Description of seepage face boundary condition is not clear. I assume the outflow only under saturated and no flow under unsaturated conditions.

L383-6: Compare Figure 5A and 5B.

L483-4: Can you provide individual flow contribution (overland and subsurface) to hydrograph?

L556-8: "18 mm min$^{-1}$".

L561-3: Maybe I missed it, but how were the infiltration fluxes treated during the winter periods on the soil surface boundary (atmospheric boundary condition)? How was snowmelt determined?

L569-72: Value of saturated hydraulic conductivity was most likely increased instead.

L608-10: Close the parenthesis.

L622-5: This might be also explained by the calibration curve used to obtain soil moisture data. No information on this was given. It is very unlikely that permanent wilting point was reached during the winter period.

L632-4: Use " … the simulated transpiration fluxes …" rather than "the simulations".

L638-40: Delete "presented model".

L643-5: Delete "partly".

L669-72: Delete comma after the reference.

L681-4: I would also mention hillslope and catchment water storage.

L705-6: Need to show units.

L709-12: Delete the first "that".

L725-7: Delete "it".

L752-3: "… we are aware of the fact that …"

L802-5: Improve English.

L819-21: Delete comma after the reference.

L979-80: Technical report?

L1074-5: Improve the reference.

L1232-3: Instead of "measured" use "determined".

L1233-5: "Though" – typo.

L1278-80: Delete the units.

L1288-90: Sink term represents mass change per volume. How the water stress function is incorporated in the model (Feddes type)?

Figure 1: The boundary of Attert catchment is shown as dashed line on map, while solid line is used in the legend.

Briefly indicate what information is obtained from the cluster stations.

Figure 3: Color of the vertical and lateral macropores at Wollefsbach catchment (D) is not similar to red color shown in the legend.

Figure 4: Add the dimension of the square(s).

Figure 7: Please check the units of soil water content.

Table 3: Please check the horizontal alignment of logNSE values.

---

## Referee Report (RR2)

**Synopsis:**

In this paper the authors address two basic questions:

1) If you have a lot of spatially-distributed information about the geology and soil-hydraulic properties in a catchment, can you parameterize a high-dimensional, spatially-distributed model (without any calibration or inverse optimization) to accurately represent water flow within a single 2-d hillslope, based on that existing knowledge?

2) If your knowledge-based (not optimized) model domain and parameterization prove reasonably representative, can you then extrapolate this representative 2-d hillslope across the 3-d volume of the entire catchment, to simulate hydrograph dynamics and the annual water balance for the entire catchment? If so, this supports the notion that a "representative hillslope" is a sufficiently good representation of an *entire* catchment.

To address these questions the authors employ a Richards-equation-based model with evapotranspiration module and overland flow routing modules. They apply the model to simulate hillslope-scale soil moisture dynamics and water-balance partitioning from two catchments in Luxembourg with varying geology, topography, soil, and vegetation. In addition to the modeling, their analysis includes extensive, and impressive data sets representing spatially distributed soil-hydraulic properties, geologic features, plant transpiration, and topography. These questions, the observations, and the methodological approach adopted here are of interest in scientific hydrology and would be received with interest by readers of *HESS*. The authors did a nice job of revising this paper from its original version. The organization of the paper now seems much more coherent, and it was a pleasure to read.

**I recommend that this paper could be accepted for publication in *HESS* with only minor remaining revisions.** I urge the authors to reconsider the phrasing of their hypothesis, and some instances where they discuss the hypothesis later in the paper. They claim to test whether a representative hillslope is "the most parsimonious" representation of a catchment. I think they have gone on to show that a representative hillslope can be used to reasonably represent streamflow generation processes for a whole catchment. But, to say that the hillslope is the most parsimonious representation would require testing the representative hillslope against some other alternatives. That was not done here. I think it is important to clarify this point, but I don't think it will require major revision—just some changes in phrasing (see comments 3 and 13). My other comments are mostly technical details. I urge the authors to consider them all carefully in their revision.

**Specific Comments:**

1.  Page 4; line 118: Can you indicate what land areas are associated with the "lower mesoscale" designation?
2.  Page 4; lines 124-128: Maybe consider rephrasing this. It seems the main point you're trying to make is that it's not really possible to develop a model that accurately represents a hillslope, or an average of hillslopes, since we'll always lack sufficient knowledge of the subsurface. As such, we have to develop a simplified conceptual model. As it reads

now, you say "cannot be a simple copy of a real hillslope", which is a little off, because a copy of a real hillslope wouldn't be simple at all.

3.  Page 5; line 142: Do you really test this hypothesis in this paper? If so, please indicate to the reader here how you will determine if the hillslope is the _most_ parsimonious representation of the catchment, rather than any other conceptual representation, or analog measurement, such as bedrock topography or porosity, soil permeability, catchment area, net radiation (i.e. Budyko model), etc. Please state clearly to the reader what will be the basis for accepting or rejecting the hypothesis. I would argue that the stated objectives that follow—while worthwhile and interesting—do not provide a rigorous test of this stated hypothesis. It seems the work is more objective driven than hypothesis driven. I don't think this is an impediment to publication of your work (most hydrological science is objective driven, rather than by hypothesis), but whether or not you're testing a hypothesis, and on what basis, should be crystal clear up front.

4.  Page 9; line 285-287: I think you should put appendix A2 in the main text. From Figure 7 I notice that many of your soil samples have porosity > 0.6. That's quite high porosity. Are the soils high in clay content, and very low in bulk density?

5.  Page 10; lines 314-317: Why do you use this proxy instead of actual sapflow [$L^3 T^{-1}$]? You have the radially-distributed measurements of heat dissipation across the xylem tissue, so you apparently have what you need to integrate and get reasonable volumetric flows. So why do you use this proxy? Also, you say you use a 12-h daily mean. Mean of what exactly, hourly-maximum velocities, or something else? It's not clear what you're averaging. I encourage you to put these calculations in equation form and report the dimensions, so this is all completely clear to the reader.

6.  Page 11; lines 345-351: Excellent, thank you! This is clearly explained and much easier to understand than the previous version.

7.  Page12; lines 374-375: Usually a seepage-face boundary is a no-flow boundary except at nodes where pressure head is greater than 0 (i.e. greater than local atmospheric pressure). Is that what you mean here?

8.  Page 15; line 476: Can you explain further what exactly was done here? It seems one of the key points of your paper is that you don't use inverse optimization procedures to defined best parameters for your model. Rather, you develop all your parameter sets based on measurement or perception. When you say that both hillslopes were setup in a few "test simulations", does this mean that you actually did optimize the parameters based on iterative simulation results and your error metric?

9.  Page 16; lines 499-500: Maybe another sentence or two here to explain a little further what this means?

10. Page 17; lines 513-514: "12-hour-rolling median of _daily_ sap flow velocities" is not clear. Can you put this into an equation so it is explicit? I'm still not entirely clear why you don't use your field measurements to quantify _sap flow_ [$L^3 T^{-1}$] across time, rather than looking at maximum values of _sap flux_ [$L T^{-1}$]. If you quantify _sap flow_ using the field measurements then you can compare it more directly to simulated transpiration. I use italics for emphasis here. You may want to adopt the sap flow and sap flux phrases for volumetric flows and fluxes, respectively. For example, you refer to "oberved sap flow", but from the observations you're only using fluxes, no? And comparing those to simulated transpiration?

11. Figure 11: It seems the model systematically overestimates soil moisture—being above the range of all observations in some cases. The measured moisture-retention curves you present in a figure are notable in that the porosities seem very high (>0.6). I'm wondering if, under the wettest observed conditions, you ever see volumetric water content measurements as high as that in the field? Figure 11 suggests your field observations of volumetric water content almost never exceed 0.4. Any chance there is some bias in those lab-measured moisture retention curves toward excessive porosity? If so, and if you reduced the porosity parameter in your model (which is based on those lab measurements), your simulated time series of soil moisture should come down closer into the range of what you observe in the field.

12. Page 20; Section 4.4 and Figure 12: I would recommend that you just omit this section and Figure 12. From the double mass curves it is clear that your model is reasonably approximating the catchment water balance at daily to seasonal time scales. Implicitly this must mean that your transpiration routine is approximately correct. Looking at these time series doesn't add anything for me. For one, even though you have normalized both variables, the comparison is still possibly confounded. The total water storage within tree stems can change (tree physiologists regularly measure this). You could have similar measured flux on two days, but slightly different volumetric flow out of the leaves (transpiration), if on one of those days part of the flux contributed to storage change wtihin the stem, rather than direct discharge from leaf to atmosphere. Also, it remains unclear exactly how you derive those daily flux estimates. Providing equations might help clarify, as noted in previous comments. Omitting this section and figure would also shorten the paper a bit, which I suspect will increase your readership. You have good depth of content even without this section and figure—to me it's just a distraction. Your decision though.

13. Page 21; lines 638-640: Here, as well as where you discuss your hypothesis in the introduction, you're referring to a "representative hillslope" as the "most parsimonious representation" of a catchment. I think you should reconsider the phrasing. Your model is not too parsimonious—it's fairly complex. A transfer function in an instantaneous-unit hydrograph approach would be much more parsimonious, and might predict time series of streamflow, and the double-mass curve, equally well. As I noted regarding your stated hypothesis, you haven't really examined other null or alternative hypotheses. I think it is more appropriate to say that you "tested the hypothesis that a representative, 2-d hillslope—conceived and parameterized based solely on observations—*can reasonably embody* all the pedological and geomorphic complexities that control streamflow generation at the whole catchment scale." To support your claim that the hillslope is "the *most parsimonious* representation of a catchment", you would have needed to compare this representation to some others, but that wasn't done here.

14. Page 22; lines 700-702: I agree, and your sensitivity analysis seems to support this.

15. Page 23; lines 730-733: Maybe clarify what you mean by "groundwater". You do have unconfined aquifers in the catchment, no, with full saturation and positive pore-water pressures, even if they are transient? I presume these exist in the riparian areas and extend upslope during large storms? It will help to more specifically distinguish "groundwater" and "subsurface storm flow".

16. Page 25; lines 767-770: Again, regarding your moisture retention curves, in the model I see you parameterized them with n values of near 1 within the Mualem-vanGenuchten

hydraulic model.  That is an exceptionally low n value, and will result in very strong suction pressures even under relatively small declines in volumetric-water content. Possibly worth looking back at that parameter selection (also see comment 11 above).

17. Page 25; line 771:  It's hard to image this being a problem consistently for so many probes.  If you wet the soil and compress it around the sensor before installation, you normally have sufficiently good contact to avoid major errors due to air space.  Also, the soil always shrinks and swells somewhat during wetting and drying, so voids that might exist due to disturbance during installation should become compressed in fairly short time.